# Mitigating Social Desirability Bias in Random Silicon Sampling

## Abstract

Large Language Models (LLMs) are increasingly used to simulate population responses, a method known as "Silicon Sampling". However, responses to socially sensitive questions frequently exhibit social desirability bias, diverging from real human data toward socially desirable answers. Existing studies on social desirability bias in LLM-based sampling remain limited. In this work, we investigate whether minimal, psychologically grounded prompt wording can mitigate this bias and improve alignment between silicon and human samples. We conduct a study using data from the American National Election Studies (ANES) on three LLMs spanning the Llama and GPT model families. We first replicate a baseline silicon sampling study, confirming persistent social desirability bias. We then test four prompt-based mitigation methods: *reformulation* (neutral, third-person phrasing), *reverse-coding* (semantic inversion), and two meta-instructions, *priming* and *preamble*, respectively encouraging analytical reasoning and sincerity. We evaluate alignment with ANES using Jensen-Shannon divergence, and introduce a signed desirability gap that measures the direction of the bias, distinguishing genuine reduction of social desirability bias from generic distributional changes.

Results show that reformulated prompts most effectively improve alignment, reducing concentration on socially desirable answers and yielding distributions closer to ANES. Reverse-coding produced mixed results across eligible items, while Priming and Preamble showed no systematic benefit for bias mitigation. We further show that reformulation remains effective across survey waves and transfers to a different survey instrument covering Western populations outside the U.S. Our findings support the use of prompt-based framing controls to mitigate social desirability bias in LLMs, providing a practical path toward more representative silicon samples.

## 1 Introduction

Large Language Models (LLMs) (Radford et al., 2018; Kojima et al., 2022) can simulate human emotions and opinions, from subjective labeling of Twitter posts (Törnberg, 2023; Yang et al., 2024) and participation in psychological studies (Aher et al., 2023; Qiu & Lan, 2024) to producing behavior changes consistent with personality frameworks (Serapio-García et al., 2023; Besta et al., 2025). These non-trivial capabilities naturally led researchers to explore whether LLMs can be used to simulate entire populations for social sciences (Argyle et al., 2023; Yang et al., 2024; Gao et al., 2023), polling (Yu et al., 2024; Zhang et al., 2024), or marketing research studies (Sarstedt et al., 2024; Arora et al., 2025). Using LLM-simulated respondents in these setups could help tackle the limitations of large sample sizes, high costs, and long execution times associated with human respondents. This led to the idea of *silicon sampling*, which refers to the use of LLM-generated agents with demographic conditioning to simulate population-level survey responses (Argyle et al., 2023; Sun et al., 2024).

Prior work found remarkable alignment between silicon and human samples on some topics; however, they diverged more sharply when sensitive topics or groups were involved (Sun et al., 2024). This divergence likely reflects *Social Desirability Bias (SDB)*, i.e., the tendency of LLMs to generate socially approved rather than demographically representative answers (Salecha et al., 2024) (see Section 2).

To make silicon sampling a viable method, reliability across topics is of key importance, which makes investigating this divergence an important research direction. We believe that the social desirability bias may largely stem from the otherwise benign decision of model developers to suppress humans' harmful biases and stereotypes learned by the model, a finding that is repeatedly confirmed by research that finds almost no explicit stereotypes in responses of different models (Liang et al., 2021; Lin et al., 2024; Bai et al., 2025; Zhao et al., 2025; Li et al., 2025). When performing silicon sampling, however, it is desirable to elicit a model's knowledge about stereotypes held by different groups of people to yield more accurate results. Although models have been successfully trained to avoid displaying any *explicit* stereotypes, researchers still find substantial *implicit* stereotypes when models are queried in a more indirect manner that does not provoke "defensive" behavior (Bai et al., 2025; Zhao et al., 2025). This suggests that careful prompt design (White, 2023) may reduce the social desirability bias and achieve more representative responses. In a silicon sampling setting, relying entirely on implicit querying methods is not feasible. However, we hypothesize that it is possible to reduce the social desirability bias by using other methods, drawn from previous LLM and psychological research.

The main goal of this study lies in the systematic exploration of prompt engineering techniques for reducing the social desirability bias in large population silicon sampling (Sun et al., 2024). We test four prompt design strategies to evaluate their effectiveness in mitigating bias and bringing a model's responses closer to human populations. This way, we hope to provide future silicon sampling studies with a method of achieving closer alignment with human samples, particularly on sensitive topics. The main contributions of this work are:

- We present a systematic evaluation of prompt-based mitigation strategies for social desirability bias in large-scale silicon sampling with LLMs. We combine Jensen–Shannon divergence and a novel signed desirability gap to quantify both distributional alignment and desirability bias.
- We establish a controlled experimental benchmark by replicating prior silicon-sampling results and extending them with four prompt manipulation conditions, enabling a comparative analysis against human survey distributions.
- We demonstrate that item reformulation, a combined intervention of neutralized wording, third-person framing, and, for some items, relabeled options, generally improves distributional alignment and response diversity, while other strategies (e.g., reverse-coding, priming, and preamble) yield limited or inconsistent benefits.
- We evaluate effectiveness under demographic stratification, across survey waves (ANES 2020 vs. 2024), and on a different survey instrument with non-U.S. Western populations (WVS Wave 7), showing that the effects are not confined to a single dataset.

## 2 Related Work

**Silicon sampling and synthetic populations.** Researchers have examined population-level simulations using LLMs, an approach known as *silicon sampling* (Argyle et al., 2023; Lee et al., 2024; Sarstedt et al., 2024). Lee et al. (2024) investigated the algorithmic fidelity and biases of LLMs in simulating public opinions regarding global warming. Instead, Sun et al. (2024) used demographic distributions from the American National Election Studies (ANES) to construct synthetic respondents and compared their political survey responses with those of real participants. While there was significant alignment between silicon and human samples on objective or politically neutral items, a larger disagreement was observed for socially sensitive questions, especially about racial diversity, gender equality, or identity politics (Sun et al., 2024). The results indicate that LLMs may tend to produce responses that align with socially desirable or politically neutral positions rather than the human opinions in real populations.

**Bias and social desirability in LLMs.** There is a broader body of research on bias in modern LLMs (Schramowski et al., 2022; Gallegos et al., 2024) that provides insight on diverging answers in silicon sampling when dealing with sensitive subjects. Many LLMs undergo the process of alignment training, in which the models are fine-tuned to follow human values and avoid harmful or biased outputs (Bai et al., 2025; Zhao et al., 2025; Li et al., 2025). While this process reduces explicit bias, implicit biases can remain embedded in internal representations and emerge only under indirect or carefully crafted queries. Prior work shows that prompting models with demographic personas can elicit stereotyped, essentializing por-

trayals rather than faithful representations of a group, an effect most pronounced for marginalized and intersectional groups (Cheng et al., 2023). Conversely, models can also move toward the opposite extreme, exhibiting social desirability bias by producing socially approved rather than statistically representative answers (Salecha et al., 2024). It is further revealed that the extent of social desirability bias depends on the context. In particular, models show stronger social desirability bias in the context of being "judged", mirroring findings from human psychology, where respondents offer more socially desirable answers when under the impression of being evaluated (Crowne & Marlowe, 1960; Paulhus & Reid, 1991).

**Mitigation strategies and cognitive framing.** Research in Psychology and Computer Science suggests approaches to reduce social desirability bias through changes in framing and context. In human respondents, bias can be mitigated by rephrasing evaluative questions into neutral or third-person formats, and by assuring anonymity and non-judgment (Fisher, 1993). Similarly, in LLMs, prompt conditioning and framing can influence the response mode of the model (Park et al., 2023; Besta et al., 2025). For example, by pre-conditioning models to adopt a more analytical persona, researchers have been able to produce more consistency in models' strategies in a game setting, with less adaptation to social circumstances. The results were also more representative of the real-world answers and less constrained by social expectations (Besta et al., 2025). Beyond prompt framing, human survey methodology offers further corrections (measurement-based control via desirability scales and post-hoc statistical adjustment against population benchmarks) that do not transfer cleanly to LLMs. Treating the LLM as an unsupervised respondent without such calibration signals, we focus on prompt-level interventions and detail these alternatives, alongside the broader bias-mitigation landscape, in Appendix A.

**Summary and research gap.** Prior work shows that LLMs can simulate human-like personas and that silicon sampling can approximate population-level survey responses. However, responses to socially sensitive questions often diverge from empirical data due to social desirability bias. Although alignment training helps reduce explicit bias, it may also suppress implicit biases that contribute to realistic human behavior in simulations. Despite this, little is known about how prompt framing, evaluation context, or cognitive conditioning influences SDB in LLM-based population sampling. We address this gap by systematically evaluating psychologically grounded prompt interventions, such as neutral phrasing, third-person framing, and rational-mode prompting, and their effects on alignment between silicon and human responses.

## 3 Methodology

Our methodology follows a four–stage pipeline: (1) extracting demographic distributions from the ANES 2020 dataset; (2) generating a synthetic (silicon) population by conditioning LLMs on these demographic profiles; (3) collecting survey responses using zero-shot prompting under five prompt conditions; and (4) evaluating alignment between silicon and human responses using divergence-based metrics. The experimental pipeline is illustrated in Figure 1. Data, code, and results are at `https://anonymous.4open.science/r/mitigate-social-desirability-bias-B092/`.

### 3.1 Data and sampling

**Survey data.** We mainly use the American National Election Studies (ANES) 2020 pre-election survey dataset, which includes 5,441 respondents.[1] The dataset contains both demographic information and responses to a wide range of political and social questions. We focus on ten multiple-choice questions/items covering different social and political topics, including *Racial Diversity*, *Gender Role*, *Current Economy*, *Drug Addiction*, *Climate Change*, *Gay Marriage*, *Refugee Allowing*, *Health Insurance*, *Gun Regulation*, and *Income Inequality*. We additionally use the ANES 2024 pre-election survey for temporal robustness analysis and World Values Survey (WVS) Wave 7 data for European countries to test generalization beyond ANES; details are given in the following sections.

**Answer-option categorization.** For each question, we classify its discrete response options into three types according to social-desirability valence, following the survey-methodology view that response options differ in their perceived social desirability (Stocké, 2004; Tourangeau & Yan, 2007). A *socially desirable*

---

[1] `https://electionstudies.org/data-center/2020-time-series-study/`

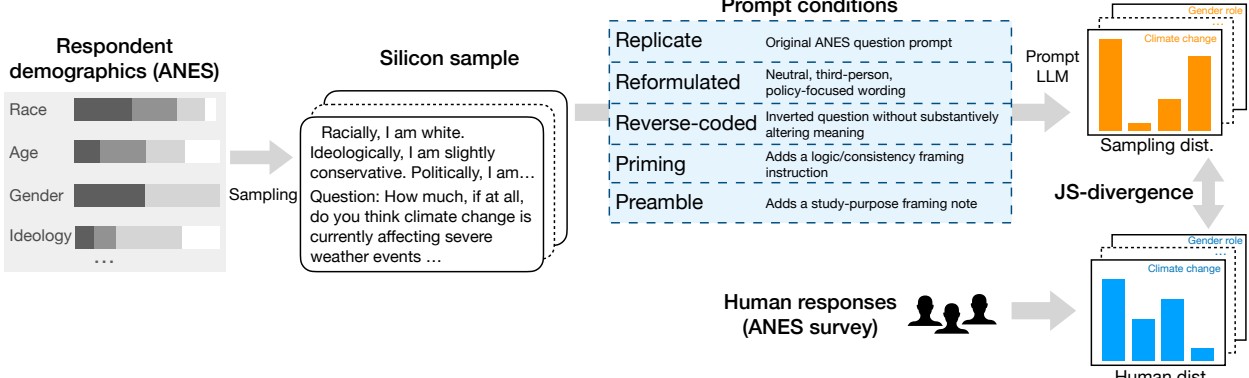

Figure 1: Overview of the experimental pipeline. We draw a random "silicon sample" of personas from the ANES respondent demographic distribution and query an LLM under five prompt conditions, then compare the resulting model response distribution (orange) against real human ANES responses (blue) via per-item Jensen–Shannon divergence and the signed desirability gap. Item wording follows Sun et al. (2024).

*(acceptable)* option expresses support for a position generally framed as socially inclusive or supportive in the survey context; a *socially undesirable* option expresses the more restrictive or opposing position; and a *safe (neutral or moderate)* option avoids strong judgment or policy change, typically a status-quo or middle response. For instance, the Health Insurance item asks:

> Do you favor an increase, decrease, or no change in government spending to help people pay for health insurance when people cannot pay for it all themselves?
>
> 1. Increase 2. Decrease 3. No change

Here, *Increase* is the socially desirable option, *Decrease* the socially undesirable option, and *No change* the safe option. The full set of items with their response options is given in Appendix (E.2 and E.3), where the most socially desirable option for each item is underlined. Desirability labels were assigned independently by the authors and two annotators external to the project, who agreed on every directional item. Disagreements occurred only for items lacking a clear social approval direction; these items were classified as non-directional.[2]

**Silicon sampling.** To generate synthetic respondents, we follow the random silicon sampling procedure in Sun et al. (2024). Let $\mathcal{D}_{\text{ANES}}$ denote the empirical ANES population. From $\mathcal{D}_{\text{ANES}}$, we estimate empirical marginal distributions over $K = 8$ demographic variables $\{D_1, D_2, \ldots, D_K\}$, corresponding to *race*, *age*, *gender*, *ideology*, *party identification*, *church attendance*, *political interest*, and *political discussion frequency*. We then construct a silicon sample $\{R_i\}_{i=1}^N$ with $N = 5,441$, matching the size of the human dataset. Each synthetic respondent $R_i$ is defined by a demographic profile sampled independently from these empirical marginals:

$$R_i = \{d_k^{(i)} \sim D_k\}_{k=1}^K. \tag{1}$$

The demographic attributes of each $R_i$ are rendered in natural language and provided to the language model as conditioning context, together with a survey question. Answer options are presented in a fixed order across all experiments. Because option positions are identical for matched items, any position bias (Zheng et al., 2024) is largely constant across conditions and is unlikely to confound our comparisons. The language model then generates a response for that synthetic individual. Repeating this procedure across all $R_i$ yields an aggregate response distribution from random silicon subjects.[3]

---

[2]These labels do not reflect a universal moral standard. We interpret them as indicators of population-level social pressure rather than as claims about the values of every subgroup.

[3]Sampling each $D_k$ independently does not preserve empirical dependencies among attributes (e.g., age–party–ideology), so some profiles may be rare or implausible. We adopt this design to obtain broad, systematic coverage of the demographic space for probing model responses in a controlled manner. Sun et al. (2024) further show that random silicon sampling reproduces

All comparisons in this study are made at the level of the aggregate distribution rather than the individual respondent $R_i$: we evaluate how closely the aggregate silicon distribution matches the *marginal* human distribution, and do not claim that any single profile predicts a real person's response. Because profiles are sampled independently from the empirical marginals, the same profile may be drawn more than once. We find that such duplication is rare and does not meaningfully affect the aggregate results (Appendix D.7). Details of demographic variables are in Appendix E.1.

## 3.2 Model selection

Our analysis employs three instruction-tuned LLMs to compare performance across scale and licensing: the open-source Llama-3.1-8B-Instruct (Llama-8B) and Llama-3.1-70B-Instruct (Llama-70B) (AI, 2024), and the closed-source GPT-4.1-mini (released on 2025-04-14). All three are alignment- and instruction-tuned, consistent with our focus on the deployed systems practitioners use for silicon sampling rather than raw pretraining checkpoints; we discuss this scope decision and the role of pre-alignment base models in Limitations (Section 5). The closed-source model is a substantial advance over the GPT-3.5-turbo model (released on 2023-06-13) used in prior silicon sampling study (Sun et al., 2024). We selected GPT-4.1-mini over the GPT-5 family (released on 2025-08-07) for its optimal performance-cost balance and faster inference speed, making it better suited for high-throughput survey simulations. Together, the selected models allow us to evaluate the trade-off among model scales, capability, computational cost, and reproducibility. All models were configured under stochastic decoding. More implementation details are provided in Appendix B.

## 3.3 Prompt-based mitigation strategies

The core of silicon sampling involves generating responses by conditioning the LLM on unique demographic profiles derived from the ANES dataset. To mitigate SDB, we design five experimental conditions by systematically varying the question structure, language, and contextual instructions.

**Replicate Condition (0)**   We replicate the results of prior work Sun et al. (2024). We keep the prompts unchanged, and ask questions directly copied from the ANES dataset. This condition is used as a baseline to compare other conditions, as well as a measure of comparative performance of three LLMs on this task.

**Reformulated Condition (1)**   We attempt to minimize SDB by reducing the perception of being evaluated or asked for an explicit opinion as determined by the LLM. This follows prior work showing that LLMs tend to produce more socially desirable answers, particularly when evaluated, and that alignment training controls explicit but not implicit bias (Salecha et al., 2024; Bai et al., 2025). We apply several modifications to the phrasing of the survey questions while ensuring that their meaning remains unchanged. For several items this also entails relabeling answer options, so the condition is a combined intervention over question framing and option wording (see Limitatisons):

- Following psychological research, we neutralize the questions by using less evaluative language (e.g., avoiding evaluative words like "good", "positive", "bad").
- We re-formulate the questions to avoid direct "what do you think" phrasing, using a more neutral third-person formulation (e.g., "what would this respondent think").
- Where possible, questions are re-formulated to ask about one's opinion about a policy rather than a social phenomenon or a group.

An example of a typical prompt under Reformulated Condition:

> *Racially, the respondent is black. The respondent doesn't like to discuss politics with their family and friends. Ideologically, the respondent is slightly liberal. Politically, the respondent is an independent. The respondent does not attend church. The respondent is 43 years old. The respondent is a man. The respondent is somewhat interested in politics.*

---

the aggregate ANES distribution more closely than full joint replication (their Table 1). Nevertheless, sampling that better preserves correlations is a promising direction we consider as a future work.

> *How would this respondent assess if there should be an increase, decrease, or no change in government spending to help people pay for health insurance when people can't pay for it all themselves?*
>
> *1. Increase 2. Decrease 3. No change*

**Reverse-coded Condition (2)** Prior work found that reverse-coding of questions can reduce SDB (Salecha et al., 2024). To test whether it can alleviate social desirability on sensitive topics, we include reverse-coded versions *where this does not substantively alter the semantic meaning of the item.* An example is given below:

> Original: How much, if at all, do you think climate change is currently affecting severe weather events or temperature patterns in the United States?
>
> *Reverse-coded: How much do you agree with the statement that climate change is unrelated to severe weather events or temperature patterns in the United States?*

For reversed-coded questions, the resulting answer options were inverted numerically to ensure consistency in the final distribution analysis. Items where reverse-coding alters the construct are omitted.

**Priming Condition (3)** This condition is grounded in studies showing that psychological priming of LLMs can influence behaviour (Besta et al., 2025). We use the conditioning prompts to induce more of a "Thinking" agent, which is added before the survey question. The verbatim instructions are:

> You value logic, objectivity, and internal consistency. When responding, you prioritize reasoning over emotion, and aim to base your answers on evidence, structure, and rational analysis. Your goal is to provide clear, well-reasoned, and intellectually honest answers that reflect careful thought. The best answer is that which reflects your reasoning process.

**Preamble Condition (4)** Adding a preamble encouraging sincere answers and promising no judgment is standard in studies on sensitive topics as it reduces pressure to conform among human respondents. We use the following verbatim preamble:

> In this study, we are exploring how people naturally respond to various questions. To ensure meaningful results, it is important that responses reflect your genuine thoughts and feelings. There are no correct or desirable answers, and your responses will not be evaluated or judged. Please answer honestly and without concern for how your answers might be perceived. Your sincerity helps us better understand authentic human responses.

For each condition, the same distribution of demographic variables was used, and each new response was produced with a fresh and isolated LLM session to preclude any memory or carry-over of context. To standardize data collection, all user prompts listed the discrete answer options and explicitly instructed the model to *respond with a single number only.* All questions and prompt conditions, including reformulated and reverse-coded variants are in Appendix E.2. LLM prompting examples are in Appendix E.4.

### 3.4 Evaluation

**Jensen-Shannon divergence (JS-divergence).** To quantify alignment between LLM-generated ("silicon") and human survey responses, we compute the JS-divergence between the empirical human distribution from ANES and the corresponding silicon distribution. JS-divergence is derived from the Kullback–Leibler divergence (KL-divergence), defined as:

$$\mathrm{JSD}(P\|Q) = \frac{1}{2}\mathrm{KLD}(P\|M) + \frac{1}{2}\mathrm{KLD}(Q\|M), \tag{2}$$

$$\text{KLD}(P\|M) = \sum_{o \in \mathcal{O}} P(o) \log \frac{P(o)}{M(o)}, \tag{3}$$

where $P$ and $Q$ are the human and silicon response distributions over answer options $o \in \mathcal{O}$ and $M = \frac{1}{2}(P + Q)$. Lower values indicate closer alignment. We adopt JS-divergence over KL-divergence or a Chi-squared test for three reasons. It is *symmetric*, suiting a comparison of two samples rather than a comparison against a "true" reference. It is *bounded* on $[0, 1]$ (base-2 logarithms) and finite under non-overlapping support, whereas $\text{KLD}(P\|Q)$ diverges whenever the silicon distribution assigns zero probability to a human-selected option, common for categorical items with finite samples. Unlike a Chi-squared statistic (Sun et al., 2024), which scales with sample size, it yields a normalized distance comparable across items and conditions.

For each question $X$, we evaluate five experimental conditions $c \in \{0, 1, 2, 3, 4\}$, with $c_0$ the Replicate baseline. Let $P_X$ denote the human response distribution for question $X$, and $Q_{c,X}$ the silicon response distribution under condition $c$. This yields five JS-divergence values per question:

$$\text{JSD}(X, c) = \text{JSD}(P_X\|Q_{c,X}).$$

**Signed desirability gap.** JS-divergence only indicates that the model's choices differ from human choices. It does not show the direction of the difference, e.g., whether the model chooses the socially desirable option more or less often than humans do, which is the key prediction of SDB. To capture the direction of the difference, we introduce a *signed desirability gap*, computed using the desirability labels from Section 3.1 and the same distributions as JS-divergence. For an item with options $\mathcal{O}$, we summarize each distribution by a desirability-weighted score $S(P) = \sum_{o \in \mathcal{O}} w_o P(o)$ and define the gap:

$$D = \frac{S(Q) - S(P)}{Z}. \tag{4}$$

The weights $w_o$ encode only the direction of social desirability. For categorical items, $w_o = 1$ for the desirable option and 0 otherwise, so $S(\cdot)$ is the probability mass on that option and $Z = 1$. For ordinal items, $w_o$ is based on the option's scale position, oriented to increase toward the desirable pole (higher values are more desirable). $S(\cdot)$ is then the mean response, and $Z = o_{\max} - o_{\min}$ normalizes by the scale range. In both cases $D \in [-1, 1]$. A positive gap means the silicon distribution is tilted more toward the desirable pole than the human one. Under SDB, we therefore expect a positive gap at baseline, $D > 0$; successful mitigation should then reduce it toward zero. Items with no desirable direction are excluded here but retained in the JS-divergence comparison.

We read each JS-divergence reduction together with the change in the signed gap. For item $X$ and condition $c$ relative to the baseline $c_0$, we consider two changes jointly: the change in divergence, $\Delta\text{JSD}_{c,X} = \text{JSD}(X, c) - \text{JSD}(X, c_0)$, and the change in the absolute gap, $\Delta|D|_{c,X} = |D(X, c)| - |D(X, c_0)|$, each defined so that negative values denote improvement. When a divergence reduction coincides with a narrowing gap ($\Delta\text{JSD}_{c,X} < 0$ and $\Delta|D|_{c,X} < 0$), the model is choosing the socially desirable answer less often, rather than its answers changing for some unrelated reason that improves alignment with human data. We refer to this as the joint criterion.

**Uncertainty estimation and statistical comparison.** To quantify uncertainty and test whether differences in JS-divergence between conditions are statistically meaningful, we use a paired non-parametric bootstrap that resamples both the silicon and the human (ANES) responses. For each question $X$ and replicate $j = 1, \ldots, n$:

1. we resample the human responses with replacement to form $P_X^{(j)}$, shared across all conditions at replicate $j$;
2. for each condition $c$, resample its silicon responses to form $Q_{c,X}^{(j)}$ and compute $\text{JSD}^{(j)}(X, c) = \text{JSD}(P_X^{(j)}\|Q_{c,X}^{(j)})$.

Resampling the human side incorporates ANES sampling error rather than treating the reference distribution as fixed; sharing $P_X^{(j)}$ within a replicate correlates the paired divergence terms and reduces the variance

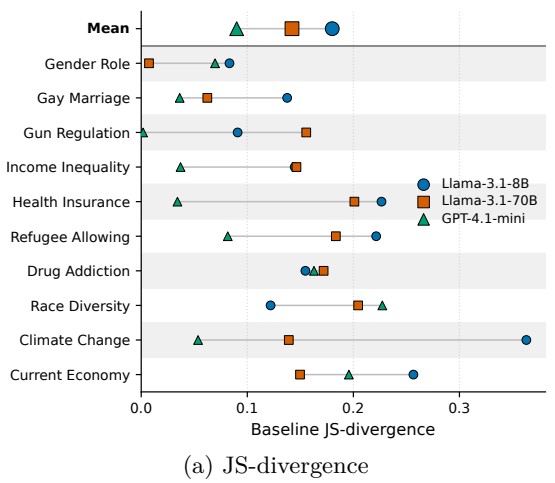 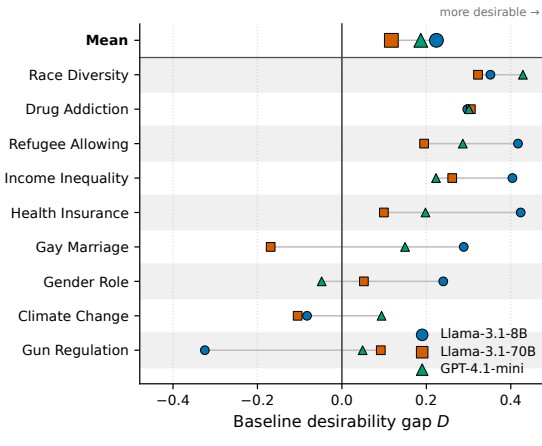

(a) JS-divergence              (b) Signed desirability gap $D$

Figure 2: Baseline (*Replicate*) alignment with the human ANES 2020 distribution, per item and per model, under stochastic decoding. **(a)** JS-divergence over all ten items; lower is closer. **(b)** Signed desirability gap over the nine directional items (economy excluded); $D > 0$ marks a shift toward the socially desirable pole, and all three models show a positive mean gap, consistent with social desirability bias. Items are ordered by cross-model mean; the top row is that mean.

Table 1: Baseline (Replicate) alignment and desirability, averaged over the nine directional items for each model. $\overline{\mathrm{JSD}}$ is the mean JS-divergence from the human distribution. $\overline{D}$ and $\overline{|D|}$ are the mean signed and absolute desirability gaps, where $D > 0$ indicates over-selection of the socially desirable option. Averaged over all ten items (adding the non-directional economy item), $\overline{\mathrm{JSD}}$ is 0.180, 0.142, and 0.090 respectively.

| Model | $\overline{\mathrm{JSD}}$ | $\overline{D}$ | $\overline{|D|}$ |
|---|---|---|---|
| Llama-3.1-8B | 0.172 | 0.224 | 0.314 |
| Llama-3.1-70B | 0.141 | 0.117 | 0.178 |
| GPT-4.1-mini | 0.078 | 0.187 | 0.198 |

of their difference. We report the 95% percentile interval of each condition's bootstrap distribution for description only. Significance is assessed using a two-sided bootstrap test on the paired differences relative to the baseline $c_0$. We compute the corresponding bootstrap $p$-values based on the empirical distribution of $\Delta\mathrm{JSD}^{(j)}_{c,X} = \mathrm{JSD}^{(j)}(X,c) - \mathrm{JSD}^{(j)}(X,c_0)$. To account for multiple comparisons (e.g., 10 items × 4 conditions per model), we control the false discovery rate at $q = 0.05$ using the Benjamini–Hochberg procedure (Benjamini & Hochberg, 1995), yielding adjusted $p$-values $p^{\mathrm{BH}}$. A comparison is significant when $p^{\mathrm{BH}} \leq q$. The family is defined as all item-condition comparisons within a given model and dataset. All bootstrap analyses use $n = 10{,}000$ replicates.

## 4 Experimental Results

### 4.1 Results across experimental conditions

**Replicate Condition (0)** We replicated the base silicon-sampling pipeline of Sun et al. (2024) using our selected LLMs, conditioning each sample on standard demographic attributes and the original survey question. Figure 2 reports per-item and per-model alignment with ANES 2020: JS-divergence (panel a) and the signed desirability gap $D$ (panel b). Table 1 gives the model-level averages, and the full response distributions are in Appendix Figure 6.

All three models show a positive mean desirability gap ($\overline{D}$) at baseline, placing more mass on the socially desirable pole than human respondents. This shift toward the desirable option is the defining feature of

Table 2: Effect of each prompt condition relative to Replicate, per model (ANES 2020). Values report changes in the mean JS-divergence $\overline{\Delta\text{JSD}}$ and mean absolute desirability gap $\overline{\Delta|D|}$ across items (condition − baseline); negative denotes improvement. Reformulated, Priming, and Preamble are averaged over the nine directional items; Reverse-coded over the six eligible items against its own six-item baseline, so its values are not directly comparable to the other columns.

| Model | Base $\overline{\text{JSD}} / \overline{|D|}$ | Reformulated | | Reverse-coded | | Priming | | Preamble | |
| --- | --- | --- | --- | --- | --- | --- | --- | --- | --- |
| | | $\overline{\Delta\text{JSD}}$ | $\overline{\Delta|D|}$ | $\overline{\Delta\text{JSD}}$ | $\overline{\Delta|D|}$ | $\overline{\Delta\text{JSD}}$ | $\overline{\Delta|D|}$ | $\overline{\Delta\text{JSD}}$ | $\overline{\Delta|D|}$ |
| Llama-3.1-8B | 0.172 / 0.314 | −0.027 | −0.065 | +0.026 | +0.012 | +0.008 | −0.051 | −0.004 | −0.043 |
| Llama-3.1-70B | 0.141 / 0.178 | +0.005 | +0.043 | +0.121 | +0.157 | +0.015 | −0.018 | +0.007 | +0.020 |
| GPT-4.1-mini | 0.078 / 0.198 | −0.019 | −0.050 | +0.146 | +0.114 | +0.042 | +0.042 | +0.043 | +0.065 |

social desirability bias, which the gap is built to measure. It holds across models despite their differing alignment: GPT-4.1-mini is closest to ANES in JS-divergence ($\overline{\text{JSD}} = 0.090$ over all ten items), followed by Llama-70B (0.142) and Llama-8B (0.180).

The models differ in how the bias appears. Llama-8B combines the largest gap ($\overline{D} = 0.224$) with the highest divergence, and on sensitive items such as *Refugee Allowing*, *Health Insurance*, and *Gay Marriage* its distributions concentrate heavily on the desirable option relative to the more dispersed human responses. Llama-70B has the smallest *signed* gap ($\overline{D} = 0.117$), but this reflects cancellation rather than closer agreement: its items fall on both sides of zero (Figure 2b), so positive and negative deviations offset in the mean. The absolute gap is larger ($\overline{|D|} = 0.178$): strongly positive on *Race Diversity* and *Drug Addiction*, negative on *Gay Marriage*, near zero on *Gender Role*, *Climate Change*, and *Gun Regulation*. This mix of opposing deviations suggests a polarized response pattern, with mass on the extremes rather than the moderate options humans more often choose. GPT-4.1-mini has the tightest alignment overall but the second-largest gap, with signed and absolute means nearly equal ($\overline{D} = 0.187$, $\overline{|D|} = 0.198$). Its shift toward the desirable pole is consistent in direction across items, producing distributions that are narrower and more concentrated on the desirable option than ANES.

Across all three models the baseline gap is positive, indicating that each over-represents the socially desirable response relative to humans. Llama-8B and GPT-4.1-mini concentrate on the desirable option, while Llama-70B splits mass across both the desirable and undesirable options. Model architecture, parameter scale, and instruction-tuning can all drive these behavioral variations, stemming from the different models' underlying representation of population knowledge and reasoning capabilities. by model capacity and underlying population knowledge. Appendix D.3 provides a more detailed per-model discussion with the full response distributions. These patterns motivate the mitigation strategies evaluated in the following sections.

**Reformulated Condition (1)** In this condition, items are neutralized and expressed in a third-person perspective. Table 2 reports the mean change in JS-divergence and the absolute desirability gap relative to the Replicate baseline, and Figure 3a relates the two per item. Table 2 shows that Reformulation reduces both quantities for Llama-8B and GPT-4.1-mini, and by the largest margins of any condition. The mean JS-divergence falls and the mean gap shrinks, indicating that improved alignment is accompanied by reduced over-selection of the socially desirable option rather than a generic response shift.

In Figure 3a, most Llama-8B and GPT-4.1-mini items fall in the lower-left quadrant, where both divergence and gap decrease: the items that improve most on JS-divergence are those whose desirability gap shrinks most. The effect is clearest on socially sensitive items, such as *Race Diversity* and *Refugee Allowing* for GPT-4.1-mini (both $\Delta|D| < −0.15$), where neutral third-person phrasing moves probability mass off the socially desirable option toward the human distribution. *Gender Role* is the exception for Llama-8B ($\Delta|D| = +0.060$). Here the human distribution concentrates on the neutral "makes no difference" option (yellow), and reformulation moves it *away* from human data (Figure 7a), increasing the absolute desirability bias. This may imply a general limitation of bias mitigation: When an model's internal representations diverge from empirical reality, reducing SDB can push the simulated distribution further from human data.

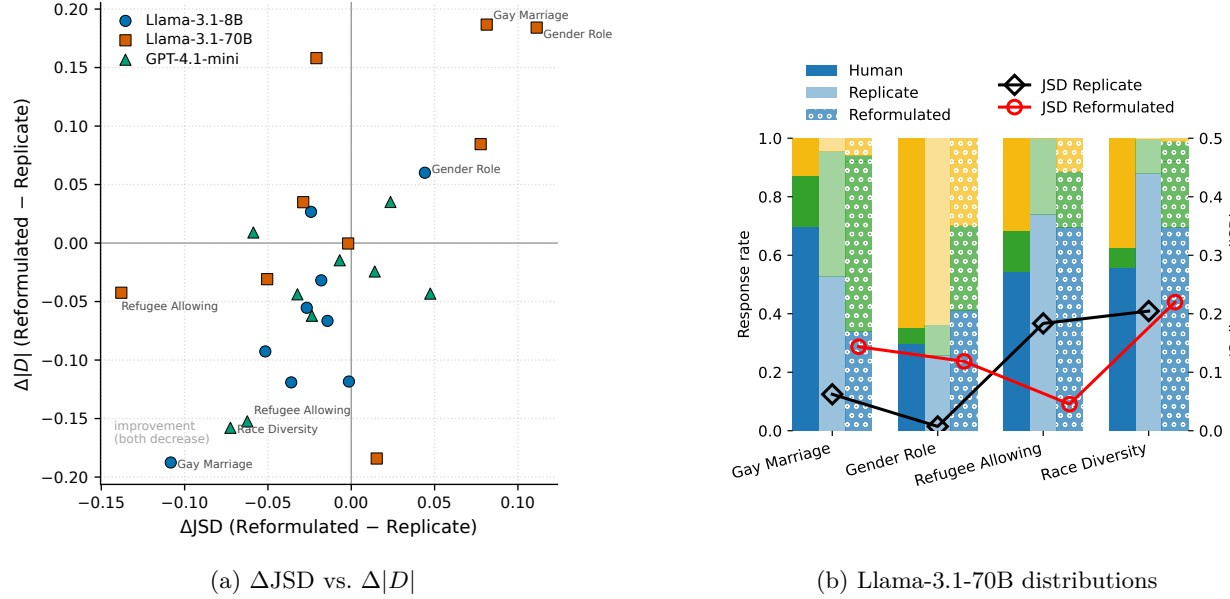

(a) ΔJSD vs. Δ|D|             (b) Llama-3.1-70B distributions

Figure 3: Reformulated condition. **(a)** Per-item change (Reformulated − Replicate) in JS-divergence and absolute desirability gap $|D|$; the lower-left quadrant denotes improvement on both. **(b)** Human, Replicate, and Reformulated response distributions for Llama-3.1-70B on four selected items, with per-item JS-divergence.

Llama-70B behaves differently. Its mean gap *widens* under reformulation ($\overline{\Delta|D|} = +0.043$) with almost no change in JS-divergence ($\overline{\Delta\mathrm{JSD}} = +0.005$), and its items scatter toward the upper-right of Figure 3a. The distributions in Figure 3b illustrate this in more detail. On *Gay Marriage* and *Gender Role*, where the gap grows most ($\Delta|D| = +0.187$ and $+0.184$, respectively), reformulation moves mass toward both extremes rather than toward the moderate human responses, increasing response diversity without reducing the gap. On *Gay Marriage* it shifts mass away from the socially desirable "Favor" (blue) option toward the socially undesirable "Oppose" (green). On *Gender Role* it shifts responses from the neutral "Makes no difference" (yellow) toward "Worse" (green). This reflects the polarized baseline documented in the Replicate condition. *Race Diversity* shows a similar dispersion in its option distribution, and although its gap shrinks, its JS-divergence still worsens ($\Delta\mathrm{JSD} = +0.015$), as reducing the gap does not move it toward the human distribution. *Refugee Allowing* is one of the items where both JS-divergence and the gap improve ($\Delta\mathrm{JSD} = -0.138$, $\Delta|D| = -0.042$).

Overall, reformulation reduces both divergence and desirability bias on the two models where it improves alignment, and the two reductions co-occur at the item level, indicating that the mechanism is reduced over-selection of the socially desirable option rather than arbitrary redistribution. On the polarized Llama-70B the same intervention sometimes increases diversity without reducing the gap (Appendix 7b).

**Reverse-coded Condition (2)** The Reverse-coded condition applies to only the six items for which reverse-coding was judged feasible, and is evaluated against its own six-item Replicate baseline. Its results are unstable, and reverse-coding provides limited value as an SDB mitigation strategy. On average it increases both the JS-divergence and the absolute desirability gap for all three models (Table 2), and the item-level improvements it does produce do not generalize across models. Figure 8 (in Appendix) shows that both Llama models exhibit improvements on *Climate Change*, with response distributions more closely resembling the human data. However, only Llama-70B achieves a reduction in the desirability gap (Table 10).

A consistent pattern of degradation appears for *Gender Role*, where all models worsen substantially. This likely reflects the difficulty of preserving semantic equivalence when implementing reverse-coding. The original question evaluates attitudes toward a situation, "*the man works outside the home and the woman takes care of the home and family.*" In this condition, it was modified to "*both the man and the woman share*

Table 3: Change relative to the Replicate baseline in JS-divergence $\Delta$JSD (top, shaded) and in the absolute desirability gap $\Delta|D|$ (bottom, gray) for GPT-4.1-mini (ANES 2020), across stratified groups and items. Each value is the per-subgroup change averaged within the group; negative denotes improvement for both metrics. Cells are shaded by $\Delta$JSD (blue $< 0$, red $> 0$, intensity scaling with magnitude). A group-item entry indicates a reduction in SDB relative to human distribution only when both values are negative.

| Group | Reformulated | | | Reverse-coded | | | Priming | | | Preamble | | |
|---|---|---|---|---|---|---|---|---|---|---|---|---|
| | Race Diversity | Refugee Allowing | Income Inequality | Race Diversity | Refugee Allowing | Income Inequality | Race Diversity | Refugee Allowing | Income Inequality | Race Diversity | Refugee Allowing | Income Inequality |
| Race | -0.036 | -0.069 | 0.069 | -0.151 | -0.057 | -0.027 | 0.037 | 0.028 | 0.028 | -0.001 | 0.145 | 0.021 |
| | -0.124 | -0.151 | -0.043 | -0.142 | -0.108 | -0.19 | 0.014 | 0.041 | 0.067 | 0 | 0.133 | 0.043 |
| Discuss Politics | -0.13 | -0.122 | 0.036 | -0.193 | -0.081 | -0.061 | 0.042 | 0.005 | 0.032 | -0.033 | 0.147 | 0.013 |
| | -0.166 | -0.212 | -0.078 | -0.185 | -0.111 | -0.276 | 0.014 | 0.018 | 0.057 | -0.018 | 0.124 | 0.024 |
| Ideology | -0.06 | -0.046 | 0.041 | -0.14 | -0.042 | 0.106 | 0.046 | 0.025 | 0.065 | -0.017 | 0.133 | 0.065 |
| | -0.172 | -0.122 | -0.04 | -0.146 | -0.086 | 0.145 | 0.023 | 0.027 | 0.092 | -0.011 | 0.145 | 0.085 |
| Party | -0.074 | -0.073 | 0.035 | -0.146 | -0.059 | 0.064 | 0.036 | 0.009 | 0.05 | -0.009 | 0.118 | 0.057 |
| | -0.147 | -0.117 | -0.016 | -0.132 | -0.109 | 0.078 | 0.013 | 0.026 | 0.078 | -0.005 | 0.125 | 0.08 |
| Church | -0.077 | -0.064 | 0.045 | -0.15 | -0.045 | -0.035 | 0.035 | 0.013 | 0.031 | -0.013 | 0.133 | 0.035 |
| | -0.159 | -0.152 | -0.044 | -0.137 | -0.085 | -0.146 | 0.013 | 0.033 | 0.072 | -0.005 | 0.142 | 0.069 |
| Gender | -0.075 | -0.061 | 0.047 | -0.146 | -0.045 | -0.031 | 0.037 | 0.013 | 0.035 | -0.013 | 0.134 | 0.038 |
| | -0.159 | -0.151 | -0.041 | -0.14 | -0.085 | -0.165 | 0.013 | 0.033 | 0.076 | -0.006 | 0.143 | 0.072 |
| Political Interest | -0.085 | -0.052 | 0.034 | -0.151 | -0.053 | -0.04 | 0.057 | 0.015 | 0.034 | 0.005 | 0.157 | 0.048 |
| | -0.153 | -0.169 | -0.07 | -0.145 | -0.124 | -0.2 | 0.024 | 0.032 | 0.068 | 0.005 | 0.156 | 0.08 |

*work outside the home and take care of the home and family.*" We consider this the closest plausible reverse-coded variant, yet it arguably alters the underlying construct rather than merely reversing its polarity. Such semantic shifts likely account for the pronounced misalignment observed for this question.

**Priming (3) and Preamble (4) Conditions** Priming and Preamble produce mixed mean effects across models (Table 2). Priming increases mean JS-divergence for all three models, while reducing the absolute desirability gap for both Llama models and increasing it for GPT-4.1-mini. Preamble slightly reduces both metrics for Llama-8B, but increases them for Llama-70B and GPT-4.1-mini. At the item level, Llama-8B shows JS-divergence reductions on a few individual items, most visibly on *Race Diversity*, where its responses align more closely with the human data with the absolute gap drop largely. However, these gains are inconsistent and often reversed on other items. For Llama-70B and GPT-4.1-mini, both Priming and Preamble increase JS-divergence, and lower response diversity in many cases compared to the Replicate condition. Per-item response distributions are shown in Figure 9 and Figure 10, and complete per-item absolute desirability gap results are reported in Appendix D.2.

The degradation in Priming and Preamble is often driven by increased response homogeneity, as model outputs concentrate more heavily on socially desirable options. We hypothesize that explicitly instructing models to be "truthful" or "sincere" may inadvertently activate a perception of evaluation in the model, reinforcing socially safe patterns, as LLMs notice they are being monitored and adjust their behavior accordingly. This observation is consistent with recent evidence on LLMs' agentic behavior where models made more conforming choices under evaluation compared to when they perceived the task as real deployment (Lynch et al., 2025). Rather than reducing SDB, such conditions appear to amplify it, resulting in less representative distributions. Full distributions for all models and four conditions are provided in Appendix D.4.

## 4.2 Demographic-stratified results

Table 3 summarizes how the four mitigation strategies, Reformulated, Reverse-coded, Priming, and Preamble, affect simulation alignment across seven *aggregated* demographic groups using GPT-4.1-mini.[4] For each demographic group (e.g., Race) we compute, per subgroup (e.g., White, Black, Asian), both the JS-divergence and the absolute desirability gap $|D|$, and report the average change in each relative to the Replicate setting. We focus on three representative items with a relatively high baseline signed desirability gap across the three models (Figure 2b): *Race Diversity*, *Refugee Allowing*, and *Income Inequality*.

The Reformulated condition reduces JS-divergence for all seven groups on the social and political items, *Race Diversity* and *Refugee Allowing*, with the desirability gap narrowing in every case. The economic item *Income*

---

[4] We exclude *age* from the stratified analysis because it is a continuous variable that requires arbitrary discretization into ranges, which could introduce confounding design choices and obscure subgroup effects.

Table 4: Difference in JS-divergence ($\Delta$JSD) and absolute desirability gap ($\Delta|D|$) between the Reformulated and Replicate conditions across the eight overlapping items in ANES 2020 and 2024, using GPT-4.1-mini. Negative values indicate improvement for both metrics. All $\Delta$JSD are significant except those marked [ns].

|  |  | Climate Change | Health Insurance | Income Inequality | Race Diversity | Drug Addiction | Gay Marriage | Gun Regulation | Gender Role |
|---|---|---|---|---|---|---|---|---|---|
| 2020 | $\Delta$JSD | **-0.032** | 0.024 | 0.047 | **-0.073** | **-0.024** | **-0.007** | 0.014 | -0.059 |
|  | $\Delta|D|$ | **-0.044** | 0.035 | -0.043 | **-0.158** | **-0.062** | **-0.015** | -0.024 | 0.009 |
| 2024 | $\Delta$JSD | **-0.051** | 0.005 | 0.074 | **-0.102** | **-0.010** | **-0.016** | 0.022 | $-0.007$[ns] |
|  | $\Delta|D|$ | **-0.142** | 0.017 | -0.010 | **-0.186** | **-0.092** | **-0.082** | 0.056 | 0.030 |

*Inequality* behaves differently: JS-divergence rises for every group, yet the desirability gap still narrows. The reformulation thus continues to reduce the desirability bias, while the overall distribution moves away from the human one for a reason unrelated to desirability. A plausible cause is the slight semantic shift introduced by paraphrasing. Economic items may be more susceptible to this shift given their lower political sensitivity.

The Reverse-coded condition produces the largest improvements on *Race Diversity*, lowering divergence for every group (roughly $-0.14$ to $-0.193$) with a narrowing gap throughout, and is the only condition that also improves *Income Inequality* on both metrics, doing so for most groups. Its gap reductions are the largest of any condition, reaching $\Delta|D| = -0.276$ for the Discuss Politics groups. For Ideology and Party, the two groups defined by political orientation, the economic gains are reversed. Why Reverse-coding reverses specifically for these groups is unclear and left to future work. On *Refugee Allowing* it improves every group but generally by less than Reformulated. Overall, it was shown to effectively reduce SDB on the three items, but these effects do not generalize to other eligible items or models.

Priming and Preamble shift all subgroups in a consistent direction for JS-divergence and the absolute desirability gap, but they mostly worsen alignment on both metrics. Priming degrades every item for every group on both divergence and gap. Preamble sharply worsens *Refugee Allowing*, yet leaves *Race Diversity* near baseline. The overall results imply that subgroup sensitivity is not driven by any single demographic axis but instead reflects broader structural properties of how LLMs emulate population-conditioned responses. Fine-grained subgroups (e.g., White vs. Black vs. Asian) broadly reflect the group-level patterns across all four conditions; per-subgroup breakdowns for both metrics are in Appendix D.9.

### 4.3 Consistency across temporal and demographic variation

A key question for silicon sampling is whether prompt-based mitigation strategies remain effective under temporal shifts in survey populations and response patterns. To evaluate this, we compare the Reformulated condition on eight overlapping items between ANES 2020 and the newly released ANES 2024 survey.[5]

As shown in Table 4, reformulation significantly lowers JS-divergence for four of the eight overlapping items in ANES 2024: *Climate Change*, *Race Diversity*, *Drug Addiction*, and *Gay Marriage*. For these items the reduction is accompanied by a narrower desirability gap, and the same holds in ANES 2020. The exceptions are also stable across waves: reformulation does not lower divergence for *Income Inequality*, *Gun Regulation*, or *Health Insurance*. For *Gender Role* the desirability gap widens in both years even though divergence falls in 2020 and is essentially unchanged in 2024. Reformulation helps and fails on the same items in both waves, which suggests the effect is not tied to a single survey year, though the evidence comes from only eight items and one model. The demographic composition of the two waves differs only modestly (Appendix C.1). This means the comparison only reflects a small shift in the target population, so it primarily tests whether the effect is stable over time rather than robust to large demographic differences.

---

[5]https://electionstudies.org/data-center/2024-time-series-study/

Table 5: Difference in JS-divergence ($\Delta$JSD) and absolute desirability gap ($\Delta|D|$) between the Reformulated and Replicate conditions across the selected items, per country (Netherlands, Germany, and Great Britain) in **WVS Wave 7**, using GPT-4.1-mini. Negative values indicate improvement for both metrics. All $\Delta$JSD are significant except those marked [ns]. *Government Responsibility* and *Income Equality* are non-directional items for which no desirability gap is defined, so only $\Delta$JSD is reported. $N$ is the human sample size.

| | Environment vs Economy | Refugee Asylum | Government Responsibility | Income Equality | Immigration Impact | Gay Parenting | Maternal Employment | Immigration Policy | Government Surveillance | Strong Leader |
|---|---|---|---|---|---|---|---|---|---|---|
| **Netherlands** ($N = 2145$) | | | | | | | | | | |
| $\Delta$JSD | **-0.008** | **-0.010** | -0.017 | -0.028 | 0.048 | -0.002[ns] | -0.032 | **-0.077** | **-0.081** | **-0.156** |
| $\Delta|D|$ | **-0.056** | **-0.005** | N/A | N/A | 0.023 | 0.027 | 0.054 | **-0.030** | **-0.006** | **-0.046** |
| **Germany** ($N = 1528$) | | | | | | | | | | |
| $\Delta$JSD | **-0.017** | -0.001[ns] | -0.019[ns] | -0.031 | 0.073 | 0.004[ns] | -0.001[ns] | **-0.021** | **-0.137** | **-0.091** |
| $\Delta|D|$ | **-0.026** | -0.001 | N/A | N/A | 0.054 | 0.003 | 0.061 | **-0.010** | **-0.044** | **-0.022** |
| **Great Britain** ($N = 2609$) | | | | | | | | | | |
| $\Delta$JSD | **-0.020** | **-0.009** | -0.026 | -0.039 | **-0.032** | 0.006[ns] | -0.011 | **-0.010** | **-0.102** | **-0.141** |
| $\Delta|D|$ | **-0.050** | **-0.004** | N/A | N/A | **-0.006** | 0.021 | 0.055 | **-0.013** | **-0.037** | **-0.051** |

### 4.4 Generalization to a new survey instrument and population

We further test whether the Reformulated condition remains effective under a change of survey instrument and population. We apply it to ten World Values Survey (WVS) Wave 7 items in three Western European countries (Netherlands, Germany, and Great Britain)[6] using GPT-4.1-mini. The items were selected to span comparable social, economic, and governance topics with differing levels of social-desirability pressure, and the demographic conditioning variables parallel those used for ANES, with the addition of education and household income (Table 35 in Appendix). We report the change in JS-divergence for each item and the change in the absolute desirability gap $|D|$ for items that have a directional social-desirability pole.

As shown in Table 5, four items satisfy the joint criterion in all three countries: *Environment vs Economy*, *Immigration Policy*, *Government Surveillance*, and *Strong Leader*, where reformulation significantly lowers JS-divergence and narrows the absolute desirability gap in every country. *Government Surveillance* and *Strong Leader* carry the highest baseline divergence and show the largest reductions, and neither has a counterpart in the ANES set, which indicates the effect is not tied to those specific items. *Maternal Employment* and *Gay Parenting* show conflicting metric directions or worsen both across countries; hence, we do not classify them as a reduction in SDB. *Immigration Impact* is the only item whose direction differs across populations: reformulation worsens both metrics in the Netherlands and Germany, while satisfying the joint criterion only in Great Britain. Meanwhile, *Refugee Asylum* produces more apparent improvements in the Netherlands and Great Britain, while yielding a non-significant change in JS-divergence and a negligible absolute gap shift in Germany. We report these patterns as observations rather than explaining them.

This cross country comparison is partly confounded: *Race* (Q290) is not used for Germany in WVS Wave 7, so the German personas condition on one fewer attribute than the other two. Full response distributions and per-item divergences for the three countries are in Appendix Figure 11. These results indicate that the Reformulated condition transfers to a different survey instrument and to Western populations outside the U.S. on similar items, while also reproducing failure mode that the joint criterion is designed to detect. Because the three are culturally proximate Western European countries, and because the transfer holds for only some items and a single model, broad cross-cultural generalization, in particular to populations whose majority views differ in direction, remains future work.

## 5 Conclusion

This work presents a systematic study of prompt-level mitigation strategies for SDB in LLM-based survey simulation. Building on random silicon sampling, we evaluate four strategies, Reformulated, Reverse-coded, Priming, and Preamble, against a Replicate baseline on three instruction-tuned LLMs. We pair JS-divergence with a signed desirability gap, counting as SDB mitigation only a joint decrease in divergence and the

---

[6] https://www.worldvaluessurvey.org/WVSDocumentationWV7.jsp

absolute gap. At baseline, all models assign more probability mass to socially desirable responses than human respondents, consistent with SDB in instruction-tuned models. The form of this bias differs across models: concentration on the desirable option in Llama-8B and GPT-4.1-mini, and a more polarized response pattern in Llama-70B.

Reformulation is the most effective strategy among all tested conditions. Neutral, third-person rephrasing reduces evaluative pressure in item wording and, in most cases, yields both lower divergence from human data and less concentration on socially desirable options. However, when the baseline is already polarized, it can shift the distribution further away from the more moderate human distribution. These effects hold across survey waves on overlapping items, transfer to a different survey instrument and non-U.S. Western populations on a subset of items. Results are stable across decoding strategies (Appendix D.6), indicating a property of the prompt design itself. Demographic-stratified analyses further suggest that SDB is largely structural rather than tied to specific subgroups, with exceptions in groups defined by political orientation.

Reformulation does not fully eliminate SDB. Its effectiveness varies with item sensitivity, is bounded by the model's underlying knowledge of subgroup opinion, and reflects a combined intervention over question framing and, for some items, option wording. The remaining strategies produce limited or inconsistent benefits. Reverse-coding is highly sensitive to semantic fidelity. Priming and preamble tend to increase response uniformity, suggesting that simple instructions are insufficient to mitigate SDB and may even exacerbate it.

These findings underscore the importance of careful prompt design for LLM-based population simulation and of evaluating mitigation with a direction-aware measure alongside distributional distance. Future work should isolate framing from option relabeling, validate desirability labels through expert annotators, examine how mitigation effectiveness varies with item sensitivity, extend the comparison across model families, sample demographic profiles that preserve attribute correlations, combine prompt-level strategies with post-hoc calibration where human benchmarks are available, and broaden evaluation to populations whose majority views differ in direction from those studied here.

## Limitations

**Model dependence.** We observe variation in baseline silicon-sampling performance across LLMs. These differences likely reflect model-specific training data, architectures, and bias mechanisms. Although item reformulation reduces SDB in most cases, its effectiveness may vary in other models. Thus, our conclusions should not be assumed to generalize to all LLMs, and broader evaluation across model families is needed.

**Alignment scope.** Our analysis is restricted to aligned, instruction-tuned models, the systems practitioners usually deploy for silicon sampling. Because alignment and preference tuning likely shape how models respond to socially sensitive items, we cannot rule out that the divergence we document is driven substantially by these post-training stages rather than by pretraining alone. We do not compare against pre-alignment base checkpoints, which would help isolate this effect but are often unavailable for closed models and exhibit weaker instruction following (e.g., less stable output formatting). We thus view SDB as a property of the deployed systems silicon sampling emulates rather than an artifact to be avoided by sampling from base models, and leave a systematic study across the pretraining–alignment pipeline to future work.

**Population coverage.** Our analysis focuses on the U.S. population represented in the ANES data, with a generalization study on three Western European populations (WVS Wave 7). These are culturally proximate to the U.S., so the effectiveness of mitigation strategies like item reformulation may not extend to more distant cultural contexts. Because we sample attributes independently from their empirical marginals (Equation 1), some profiles may be implausible. Since all comparisons are aggregate-level, this is unlikely to invalidate our conclusions, though we cannot rule out residual artifacts from profiles with conflicting attributes. Future research should validate these findings across broader demographic dimensions and global survey datasets.

**Bias, knowledge gaps, and stereotypes.** Even when SDB is present, attributing a given response to it alone remains challenging. A model's tendency to default to a "safe" response may reflect normative pressure, lack of group-specific knowledge, stereotyped associations learned about demographic groups (Cheng et al., 2023), or some combination of these. Because these sources are not separable from the prompt alone, our

aggregate distributions may conflate genuine population-level regularities with stereotypical priors. Moreover, where the model lacks accurate internal representations of subgroup preferences, bias mitigation alone may be ineffective or even misleading. This limits our ability to precisely quantify the magnitude of SDB and the upper bound of mitigation performance.

**Other sources of response bias.** SDB is one of several biases that may shape silicon-sampling outputs. LLMs exhibit a mix of human- and non-human-like biases, reproducing some human tendencies but omitting or reversing others (Chen et al., 2025). They can also favor answer options by placement or ID rather than content (position bias) (Zheng et al., 2024); we use a fixed option ordering across all conditions, holding this constant for matched comparisons, though absolute distributions may retain residual effects. Models may also homogenize, collapsing toward similar answers even under stochastic decoding (Jiang et al., 2026); since faithful sampling must capture the variance of human attitudes, this flattens within-group diversity regardless of SDB. Sycophancy may further interact with SDB: models defer to perceived user views on subjective items but resist on objective ones (Ranaldi & Pucci, 2023). We focus on SDB because it is measurable and theoretically grounded, and leave the interaction of these mechanisms to future work.

**Contextual independence.** Our study uses single-item prompting to isolate the effects of specific mitigation strategies and items, maintaining a controlled environment. However, real-world respondents do not provide answers in isolation; they navigate a sequence of questions that are susceptible to Question Order Bias (McFarland, 1981). Thus, while our study establishes the efficacy of reformulation for individual items, it does not account for the cumulative contextual biases that may emerge in full-scale sequential survey simulations.

**Reformulation as a combined intervention.** Our reformulation alters both question framing (neutral, third-person wording) and, for several items, the answer-option labels (e.g., "Favor/Oppose" → "Should be allowed/Should not be allowed," or "good/bad" → "strong/weak"). The measured effect therefore reflects this combined intervention rather than framing in isolation, and we cannot fully separate the contribution of relabeling options from that of reframing questions.

## Ethical Considerations

Silicon sampling raises ethical complications that deserve careful attention, and should support human perspectives rather than replace them.

**Synthetic data.** While LLMs efficiently simulate population-level responses, their outputs are model-based, not human-based, and should not be treated as empirical facts. Researchers should clearly disclose the synthetic nature of these data and *label synthetic outputs wherever they appear.*

**Bias.** Even carefully framed prompts cannot eliminate the effects of societal inequalities, blind spots, or misinformation embedded in training data. This is critical when simulating perspectives from marginalized groups, where biased outputs can reinforce stigma. Transparency in prompt design, model parameters, and demographic conditioning, as well as a brief *harm review* for sensitive items, is essential.

**Risks of mitigation.** Mitigating social desirability bias can mask rather than resolve a deeper limitation. A closer match to a population benchmark may reflect either grounded subgroup knowledge or merely outputs shifted toward a plausible target. Where the model lacks genuine subgroup knowledge, removing surface bias is not merely ineffective but misleading. Reframing should therefore be reported as the reduction of a specific, measurable bias, not as evidence that the resulting sample is representative.

**Privacy.** Participant privacy must also be protected. Research should rely on aggregate data, avoid reconstructing individual responses or using outputs to characterize or make decisions about individuals, comply with dataset licenses, and share only what is needed for replication.

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

# A    Additional Related Work

**Survey methodology corrections beyond question design.** Beyond framing, human survey methodology offers corrections that do not transfer cleanly to LLMs. *Procedural* techniques such as anonymity assurances, the bogus-pipeline paradigm (Jones & Sigall, 1971), and randomized response (Warner, 1965), reduce desirable responding by altering perceived detectability; for an LLM these are delivered through the prompt and so reduce to the framing interventions. Two distinct approaches remain. The first *measures and controls* for desirable responding via validated scales such as Marlowe–Crowne (Crowne & Marlowe, 1960) or the BIDR (Paulhus & Reid, 1991), using a respondent's desirability score to adjust their substantive answers. These scales were developed and validated for human respondents, and it remains unclear whether they apply to LLMs, so this is a non-trivial future direction rather than a readily available correction. The second, *post-hoc statistical adjustment* through calibration weighting (Deville & Särndal, 1992) and post-stratification (Royal, 2019), and human-LLM rectification such as prediction-powered inference (Angelopoulos et al., 2023), corrects responses against population benchmarks or held-out human labels. As we treat the LLM as an unsupervised respondent without such calibration signals, we focus on prompt-level interventions and regard these corrections as complementary directions where validation data are available.

**Bias measurement and mitigation in LLMs.** A growing body of work measures and mitigates bias in LLMs (Gallegos et al., 2024), and the techniques are commonly grouped by the stage at which they intervene. Pre-processing methods modify inputs or training data, as in counterfactual data augmentation (Lu et al., 2020). In-training methods edit parameters or the optimization objective, ranging from alignment fine-tuning (Bai et al., 2025) to parameter-efficient (Ranaldi et al., 2024) and projection-based (Ravfogel et al., 2020) approaches. Intra-processing methods alter inference behaviour without retraining, such as decoding-time self-debiasing (Schick et al., 2021). Post-processing methods rewrite or filter generated outputs (Tokpo & Calders, 2022). Parameter-level methods reach a model's internal representations but require model access and retraining, while intra-processing methods (which act at inference time, including prompt-level conditioning) are cheap and model-agnostic yet reach only elicited behaviour, not the underlying representations. Our approach is an intra-processing method, but departs from prior work in both target and goal: we target *social desirability bias* rather than stereotyping, and our goal is population representativeness in silicon sampling, not harm reduction.

# B    Implementation Details

## B.1    Model configuration

We use Llama-3.1-8B-Instruct (Llama-8B), Llama-3.1-70B-Instruct (Llama-70B), and GPT-4.1-mini in our study. All three models use stochastic sampling. For each demographic profile, we sample from the response distribution rather than selecting the model's most likely answer. For Llama-8B and Llama-70B, sampling was enabled (`do_sample=True`) with temperature $= 0.6$, top-$p = 0.9$, and top-$k = 50$; the maximum number of new tokens was set to two, sufficient for the single-word numerical answer options. GPT-4.1-mini followed the same elicitation procedure via the OpenAI API but with temperature $= 1.0$ and top-$p = 1.0$. Llama models were run on a high-performance computing cluster with nodes equipped with NVIDIA H100 GPUs.

## B.2    Non-response handling

Across all items and in both the ANES and WVS data, we compute human response distributions over the substantive answer options only, i.e., the same options presented to the model. Non-substantive responses (e.g., "Don't know", "Refused", "Inapplicable", and equivalent negative survey codes) are excluded, and the remaining option probabilities are renormalized to sum to one before computing JS-divergence. Exclusion is applied per item rather than per respondent: an answer is dropped only for the item on which it is non-substantive, and the respondent is kept for every item they did answer. We use this available-case treatment rather than listwise deletion, following prior work (Sun et al., 2024). This keeps the human and silicon distributions defined over an identical support for each item and applies uniformly across datasets. It also ensures that comparisons between ANES and WVS rest on the same treatment of missing responses.

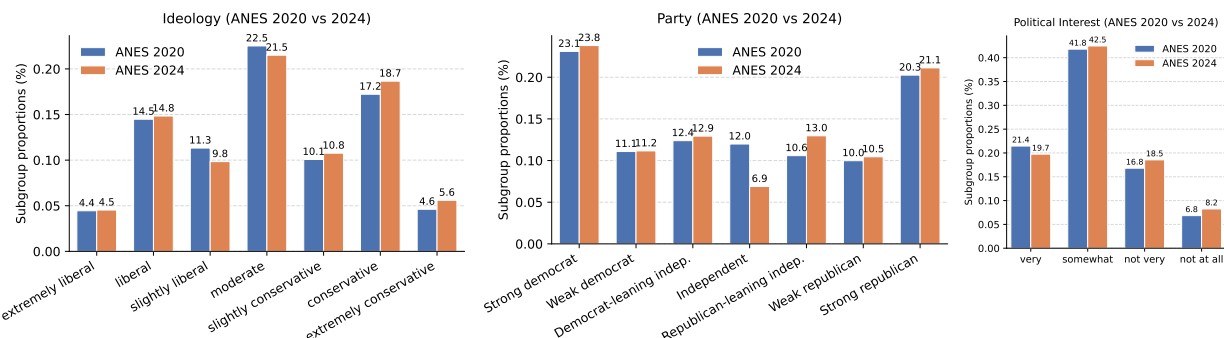

Figure 4: Subgroup proportions for Ideology, Party, and Political Interest, ANES 2020 vs. 2024.

Non-substantive codes on the demographic conditioning variables are likewise excluded, and the sampling marginals are computed over the substantive categories only. We consider silicon sampling with an explicit abstention option as future work.

## C   Demographic Background

### C.1   Demographic shifts between ANES 2020 and 2024

Figure 4 illustrates detailed subgroup proportions for *Ideology*, *Party*, and *Political Interest*. We found for *Ideology*, a small portion of respondents became more conservative. For *Party*, more respondents tend to be less concentrated on the most neutral option, "Independent", and were more polarized. For *Political Interest*, relatively more respondents became less interested in 2024 compared to 2020.

### C.2   Demographic distributions of the WVS populations

Figure 5 shows the distributions of the demographic conditioning variables for the three WVS Wave 7 populations. The personas used in our silicon sampling are drawn from these marginals, so the figure characterizes the inputs to the generalization experiment rather than any model output. The three profiles are broadly similar. Respondents cluster around the ideological center, most report middle household income and upper-secondary education, and the largest share rarely or never attend religious services, with gender close to balanced.

## D   Additional Results

### D.1   Full JS-divergence results

Table 6, Table 7, and Table 8 show per-item changes in JS-divergence under each condition across the three models. Full per-comparison statistics (bootstrap CIs) are in subsection D.10.

### D.2   Full absolute desirability gap results

Table 9, Table 10, and Table 11 show per-item changes in absolute desirability gap under each condition across the three models.

### D.3   Full response distributions for the Replicate condition

The full response distributions for the Replicate condition are shown in Figure 6. The smallest model, Llama-8B, shows the largest departure from the human distributions. On sensitive items such as *Refugee Allowing*, *Health Insurance*, *Drug Addiction*, and *Gay Marriage*, its responses concentrate on the socially

Table 6: Per-item change in JS divergence $\Delta$JSD (condition $-$ Replicate) under each condition, Llama-3.1-8B (ANES 2020). Negative values indicate improvement. Reverse-coded applies to six eligible items. All differences are significant except those marked [ns].

| Question | Reformulated | Reverse-Coded | Priming | Preamble |
|---|---|---|---|---|
| Climate Change | $-0.0242$ | $-0.2489$ | $0.0510$ | $0.0062^{\text{ns}}$ |
| Current Economy | $-0.1343$ | N/A | $0.0392$ | $-0.0541$ |
| Refugee Allowing | $-0.0267$ | $0.0512$ | $-0.0176$ | $-0.0375$ |
| Health Insurance | $-0.0516$ | N/A | $-0.0175$ | $-0.0132$ |
| Income Inequality | $-0.0014^{\text{ns}}$ | $-0.0826$ | $0.0311$ | $-0.0142$ |
| Race Diversity | $-0.0143$ | $0.0474$ | $-0.1198$ | $-0.0973$ |
| Drug Addiction | $-0.0360$ | N/A | $-0.0119$ | $-0.0549$ |
| Gay Marriage | $-0.1083$ | $0.0345$ | $0.0326$ | $0.0345$ |
| Gun Regulation | $-0.0180$ | N/A | $0.1014$ | $0.0908$ |
| Gender Role | $0.0442$ | $0.3546$ | $0.0298$ | $0.0551$ |

Table 7: Per-item change in JS divergence $\Delta$JSD (condition $-$ Replicate) under each condition, Llama-3.1-70B (ANES 2020). Negative values indicate improvement. Reverse-coded applies to six eligible items. All differences are significant except those marked [ns].

| Question | Reformulated | Reverse-Coded | Priming | Preamble |
|---|---|---|---|---|
| Climate Change | $0.0776$ | $-0.0865$ | $0.0385$ | $0.0149$ |
| Current Economy | $0.0697$ | N/A | $0.0510$ | $0.0183$ |
| Refugee Allowing | $-0.1380$ | $0.0156$ | $0.0077$ | $-0.0015$ |
| Health Insurance | $-0.0209$ | N/A | $-0.0031^{\text{ns}}$ | $-0.0175$ |
| Income Inequality | $-0.0019^{\text{ns}}$ | $-0.0163$ | $-0.0092$ | $0.0144$ |
| Race Diversity | $0.0153$ | $0.0553$ | $-0.0264$ | $0.0122$ |
| Drug Addiction | $-0.0505$ | N/A | $0.0008^{\text{ns}}$ | $0.0008^{\text{ns}}$ |
| Gay Marriage | $0.0813$ | $0.0427$ | $-0.0031^{\text{ns}}$ | $-0.0029^{\text{ns}}$ |
| Gun Regulation | $-0.0288$ | N/A | $0.0035^{\text{ns}}$ | $0.0408$ |
| Gender Role | $0.1112$ | $0.7133$ | $0.1253$ | $0.0034$ |

desirable option (lighter bars), where human responses are more dispersed. This concentration on a single desirable option, rather than the spread humans show, is consistent with a strong desirability effect in this model.

Llama-70B has generally lower JS-divergence than Llama-8B (Table 1), but its distributions depart from humans in a different way. On several items (e.g., *Refugee Allowing*, *Health Insurance*), it places substantial mass on both the socially desirable (lighter blue) and the socially undesirable (lighter green) options, where human respondents more often choose the moderate (yellow) response (e.g., "No change" or "Neither favor nor oppose"). This bimodal pattern, rather than concentration on a single option, distinguishes 70B from 8B: the model places mass on both poles instead of the moderate response, a polarized rather than one-sided departure from the human distribution.

GPT-4.1-mini shifts toward the socially desirable option on most items, producing distributions that are narrower and more concentrated on the desirable option than those in ANES 2020. On sensitive items such as *Race Diversity* and *Drug Addiction*, its simulated respondents concentrate on a single option almost entirely, whereas the human responses are spread across several options.

### D.4 Full response distributions for Reformulated, Reverse-coded, Priming and Preamble conditions

The full response distributions across the Reformulated, Reverse-coded, Priming, and Preamble conditions are shown in Figure 7, Figure 8, Figure 9, and Figure 10.

Table 8: Per-item change in JS divergence $\Delta$JSD (condition $-$ Replicate) under each condition, GPT-4.1-mini (ANES 2020). Negative values indicate improvement. Reverse-coded applies to six eligible items. All differences are significant except those marked [ns].

| Question | Reformulated | Reverse-Coded | Priming | Preamble |
|---|---|---|---|---|
| Climate Change | −0.0323 | 0.2666 | 0.1460 | 0.0224 |
| Current Economy | −0.0494 | N/A | 0.0679 | −0.0112 |
| Refugee Allowing | −0.0624 | −0.0459 | 0.0120 | 0.1331 |
| Health Insurance | 0.0235 | N/A | 0.0314 | 0.1237 |
| Income Inequality | 0.0474 | −0.0323 | 0.0344 | 0.0382 |
| Race Diversity | −0.0725 | −0.1486 | 0.0390 | −0.0106 |
| Drug Addiction | −0.0236 | N/A | 0.0099 | 0.0099 |
| Gay Marriage | −0.0069 | 0.0590 | −0.0126 | 0.0107 |
| Gun Regulation | 0.0141 | N/A | 0.0067 | 0.0099 |
| Gender Role | −0.0587 | 0.7763 | 0.1118 | 0.0460 |

Table 9: Per-item change in absolute desirability gap $\Delta|D|$ (condition $-$ Replicate) under each condition, Llama-3.1-8B (ANES 2020). Negative values indicate improvement. Reverse-coded applies to six eligible items. *Current Economy* is non-directional and excluded.

| Item | Replicate $|D|$ | Reformulated | Reverse-coded | Priming | Preamble |
|---|---|---|---|---|---|
| Gay Marriage | 0.289 | −0.188 | +0.016 | +0.016 | +0.016 |
| Refugee Allowing | 0.417 | −0.055 | −0.220 | −0.029 | −0.085 |
| Income Inequality | 0.404 | −0.118 | −0.272 | +0.009 | −0.028 |
| Gender Role | 0.240 | +0.060 | +0.443 | −0.092 | +0.031 |
| Climate Change | 0.083 | +0.027 | +0.066 | −0.027 | −0.014 |
| Gun Regulation | 0.325 | −0.032 | N/A | +0.127 | +0.118 |
| Drug Addiction | 0.297 | −0.119 | N/A | −0.097 | −0.074 |
| Race Diversity | 0.352 | −0.067 | +0.039 | −0.336 | −0.341 |
| Health Insurance | 0.423 | −0.093 | N/A | −0.025 | −0.009 |
| Mean | 0.0.314 | −0.065 | +0.012 | −0.051 | −0.043 |

### D.5   Full response distributions for WVS Wave 7

Figure 11 shows the human, Replicate, and Reformulated response distributions for the ten WVS Wave 7 items, with the per-item JS-divergence under each condition, for the Netherlands, Germany, and Great Britain, complementing the analysis in the main text (subsection 4.4).

### D.6   Decoding robustness

Our main experiments use stochastic decoding, but we also evaluate the methods under deterministic (greedy) decoding, which provides greater reproducibility. Table 12 compares the two decodings for GPT-4.1-mini on ANES 2020, reporting the change in mean JS-divergence and mean absolute desirability gap relative to Replicate under each setting. The two decodings yield nearly identical results. Baseline alignment is slightly worse under deterministic decoding (higher $\overline{\text{JSD}}$ and $\overline{|D|}$), because always selecting the single most likely answer discards the response variation that better matches the spread of human responses. The effect of each prompt condition is nonetheless stable across settings. Reformulated improves alignment on both metrics under both decodings, while Reverse-coded, Priming, and Preamble do not. The reformulation effect therefore appears to be a property of the prompt design rather than a consequence of the decoding strategy.

Table 10: Per-item change in absolute desirability gap $\Delta|D|$ (condition − Replicate) under each condition, Llama-3.1-70B (ANES 2020). Negative values indicate improvement. Reverse-coded applies to six eligible items. *Current Economy* is non-directional and excluded.

| Item | Replicate $|D|$ | Reformulated | Reverse-coded | Priming | Preamble |
|------|------|------|------|------|------|
| Gay Marriage | 0.169 | +0.187 | −0.123 | −0.037 | −0.069 |
| Refugee Allowing | 0.195 | −0.042 | +0.182 | −0.106 | +0.020 |
| Income Inequality | 0.261 | −0.000 | −0.005 | −0.062 | +0.066 |
| Gender Role | 0.052 | +0.184 | +0.825 | +0.001 | −0.011 |
| Climate Change | 0.105 | +0.085 | −0.058 | −0.048 | −0.012 |
| Gun Regulation | 0.092 | +0.035 | N/A | +0.032 | +0.048 |
| Drug Addiction | 0.305 | −0.031 | N/A | +0.000 | +0.000 |
| Race Diversity | 0.322 | −0.184 | +0.118 | +0.044 | +0.027 |
| Health Insurance | 0.099 | +0.158 | N/A | +0.014 | +0.110 |
| Mean | 0.178 | +0.043 | +0.157 | −0.018 | +0.020 |

Table 11: Per-item change in absolute desirability gap $\Delta|D|$ (condition − Replicate) under each condition, GPT-4.1-mini (ANES 2020). Negative values indicate improvement. Reverse-coded applies to six eligible items. *Current Economy* is non-directional and excluded.

| Item | Replicate $|D|$ | Reformulated | Reverse-coded | Priming | Preamble |
|------|------|------|------|------|------|
| Gay Marriage | 0.150 | −0.015 | −0.027 | +0.001 | +0.048 |
| Refugee Allowing | 0.286 | −0.152 | −0.085 | +0.033 | +0.143 |
| Income Inequality | 0.223 | −0.043 | −0.168 | +0.074 | +0.071 |
| Gender Role | 0.048 | +0.009 | +0.898 | +0.006 | +0.004 |
| Climate Change | 0.094 | −0.044 | +0.205 | +0.142 | +0.063 |
| Gun Regulation | 0.049 | −0.024 | N/A | +0.029 | +0.070 |
| Drug Addiction | 0.301 | −0.062 | N/A | +0.004 | +0.004 |
| Race Diversity | 0.429 | −0.158 | −0.137 | +0.014 | −0.005 |
| Health Insurance | 0.198 | +0.035 | N/A | +0.075 | +0.185 |
| Mean | 0.198 | −0.050 | +0.114 | +0.042 | +0.065 |

Table 12: Effect of each prompt condition relative to Replicate for GPT-4.1-mini under stochastic (T=1) and deterministic (T=0) decoding (ANES 2020). Values report changes in the mean JS-divergence $\overline{\Delta \text{JSD}}$ and mean absolute desirability gap $\overline{\Delta|D|}$ across items (condition − baseline); negative denotes improvement. Reformulated, Priming, and Preamble are averaged over the nine directional items; Reverse-coded over the six eligible items against its own six-item baseline, so its values are not directly comparable to the other columns.

| | | Reformulated | | Reverse-coded | | Priming | | Preamble | |
|------|------|------|------|------|------|------|------|------|------|
| Temperature | Base $\overline{\text{JSD}}$ / $\overline{|D|}$ | $\overline{\Delta \text{JSD}}$ | $\overline{\Delta|D|}$ | $\overline{\Delta \text{JSD}}$ | $\overline{\Delta|D|}$ | $\overline{\Delta \text{JSD}}$ | $\overline{\Delta|D|}$ | $\overline{\Delta \text{JSD}}$ | $\overline{\Delta|D|}$ |
| $T = 1$ | 0.078 / 0.198 | −0.019 | −0.050 | +0.146 | +0.114 | +0.042 | +0.042 | +0.043 | +0.065 |
| $T = 0$ | 0.089 / 0.208 | −0.021 | −0.051 | +0.144 | +0.119 | +0.039 | +0.043 | +0.051 | +0.067 |

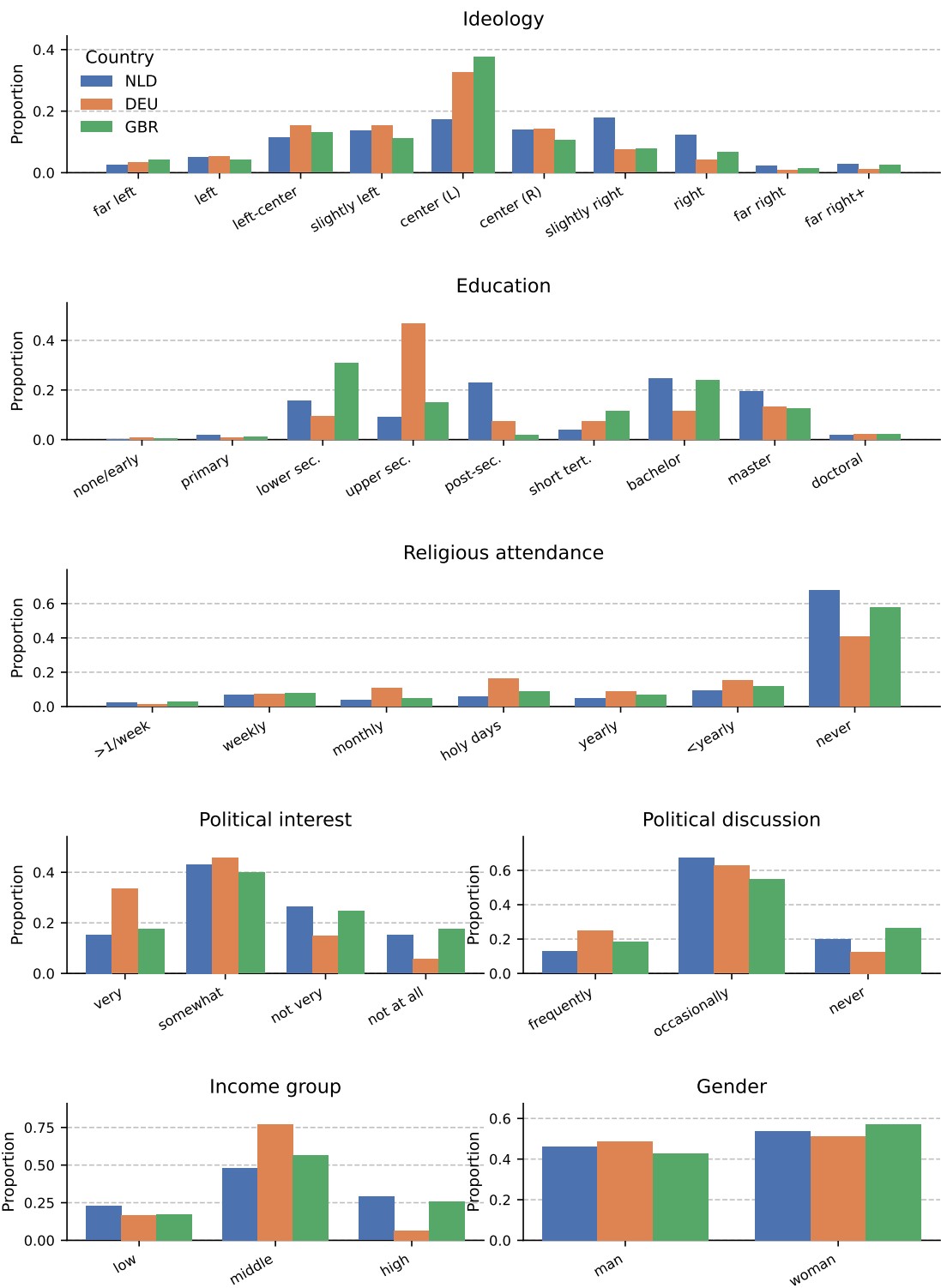

Figure 5: Distributions of the demographic conditioning variables across the three WVS Wave 7 countries (Netherlands, Germany and Great Britain). Bars show the proportion of respondents in each category per country. Race (Q290) is omitted, as it uses country-specific, non-comparable category schemes, and not used in Germany.

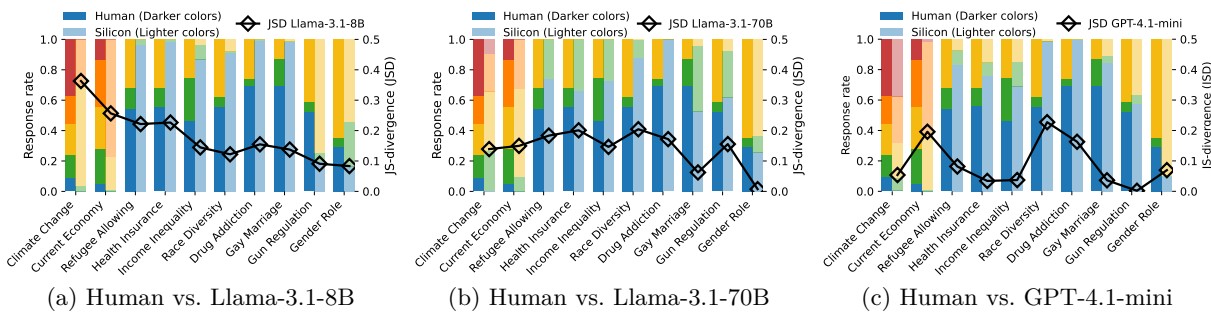

Figure 6: Baseline (Replicate) response distributions for the ten items, ANES 2020. For each item, bar color denotes the answer option in its fixed survey order and does not encode social desirability; darker and lighter shades of the same color denote human and silicon respondents, respectively. The black markers (right axis) give the JS-divergence between the human and silicon distributions.

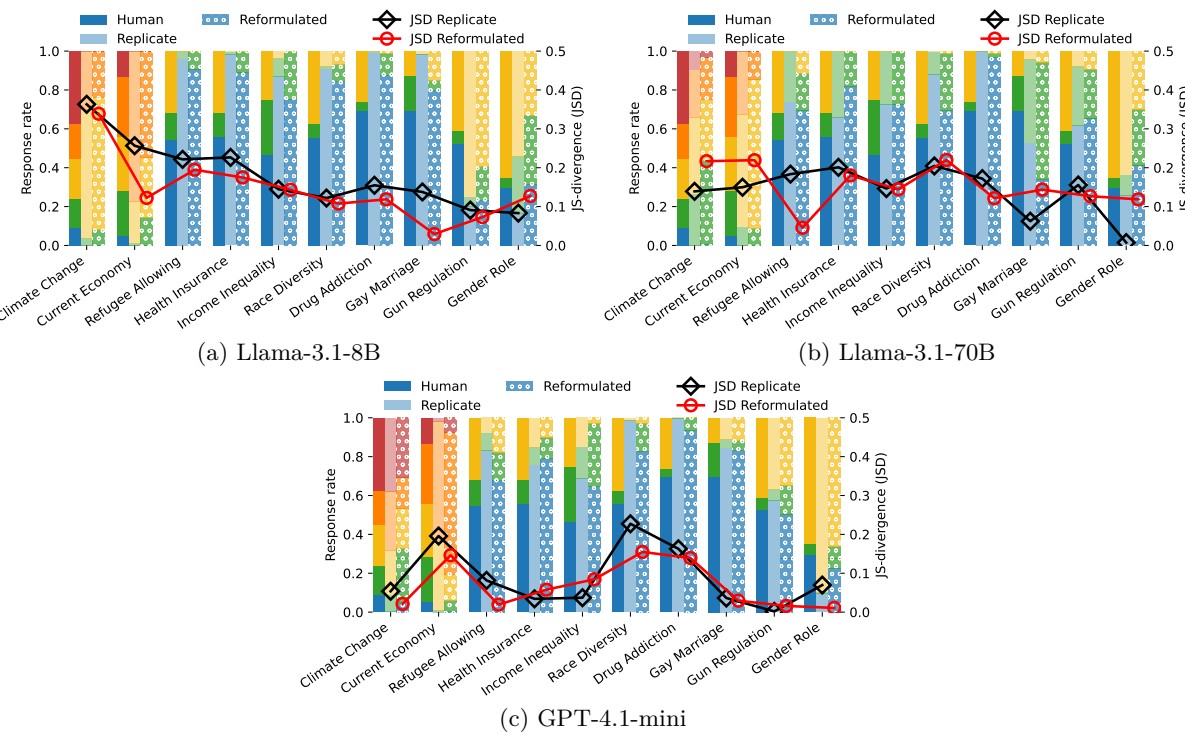

Figure 7: Human, Replicate, and **Reformulated** responses for all items on three models.

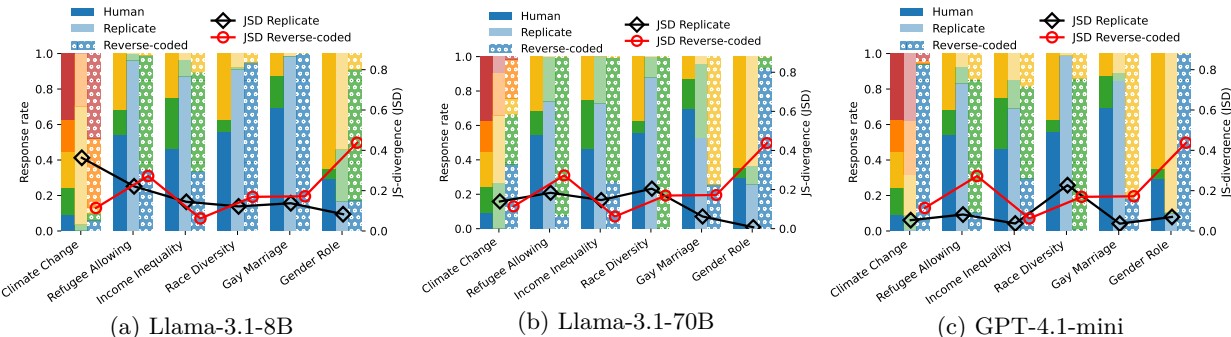

Figure 8: Human, Replicate, and **Reverse-coded** responses for all items on three models.

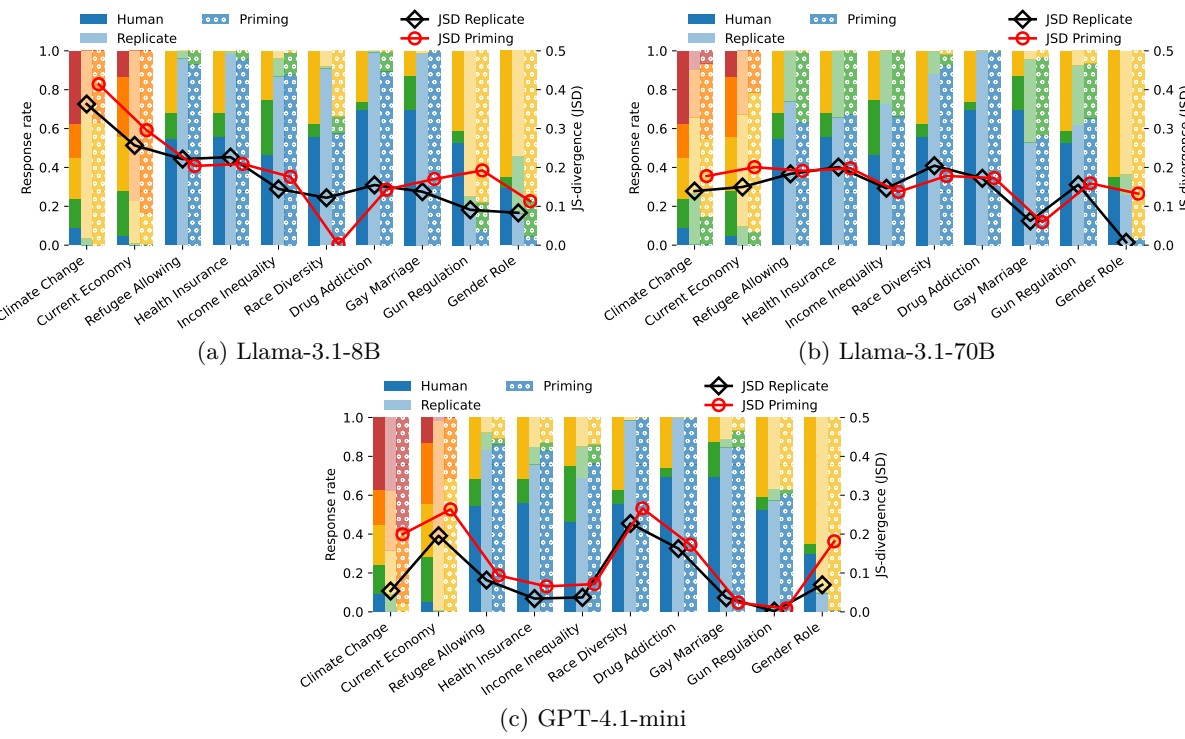

Figure 9: Human, Replicate, and **Priming** responses for all items on three models.

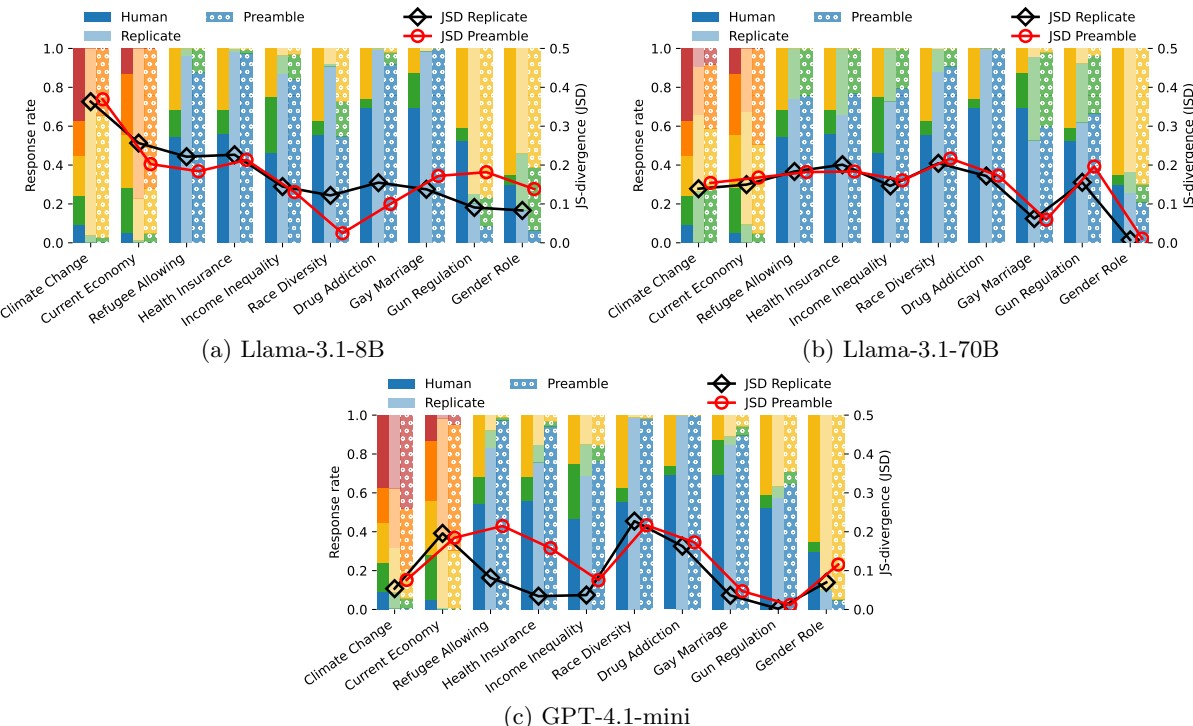

Figure 10: Human, Replicate, and **Preamble** responses for all items on three models.

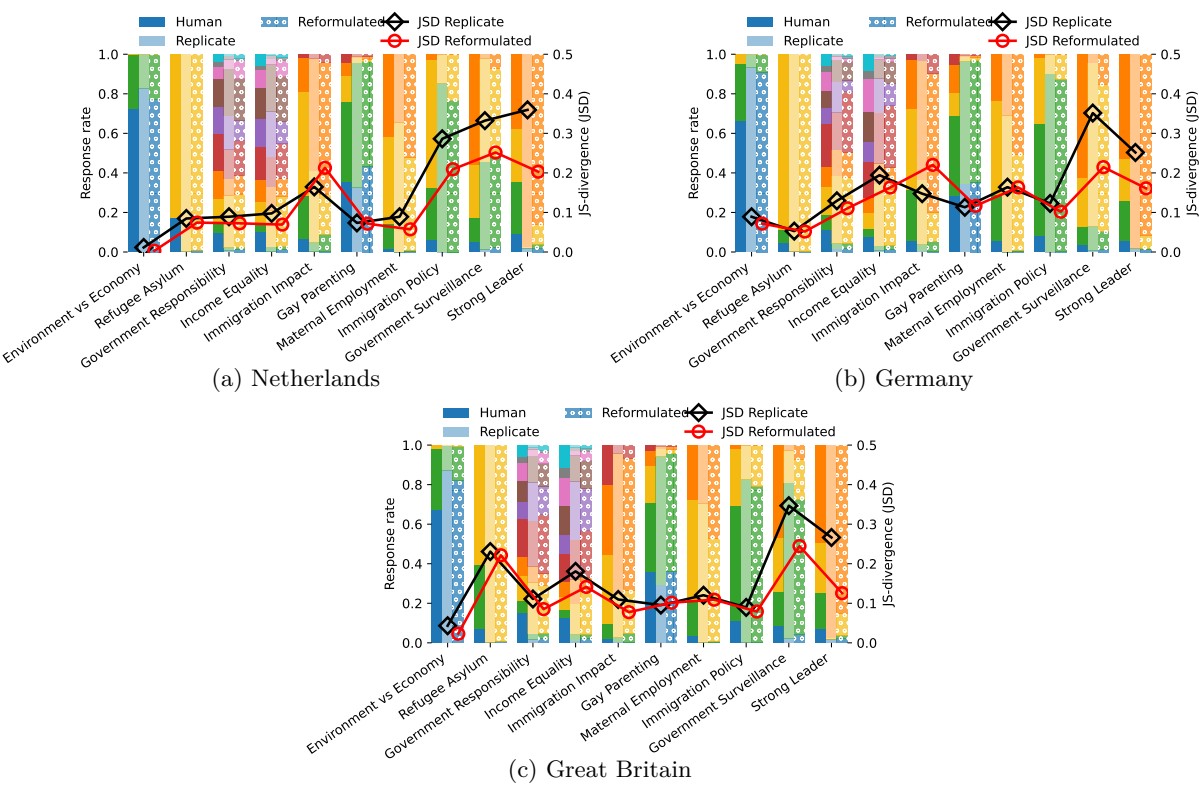

Figure 11: Human, **Replicate**, and **Reformulated** response distributions for the ten WVS Wave 7 items, using GPT-4.1-mini, for the Netherlands, Germany, and Great Britain.

### D.7 Demographic profile duplication

In silicon sampling, each synthetic respondent is assigned a demographic profile drawn from the target population distribution. Because profiles are sampled, the same combination of demographic attributes may be drawn more than once, producing duplicate profiles. This duplication is a property of the sampling process itself, independent of the decoding strategy. It becomes a concern under greedy (deterministic) decoding, where duplicate profiles are queried with identical prompts and therefore yield identical responses, effectively sampling the same individual multiple times. To assess how often this occurs, we measured the rate of duplicated demographic profiles produced by our sampling procedure.

We define two respondents as duplicates if they are identical across all demographic variables used to construct the profile (subsection E.1) We sampled 5,441 synthetic respondents (matching the size of the ANES 2020 sample), using GPT-4.1-mini, and computed the proportion of respondents whose profile coincided with that of at least one other respondent. Table 13 reports these rates across the ten survey questions under both the Replicate and Reformulated conditions.

Table 13: Percentage of duplicated demographic profiles across survey items, for 5,441 sampled synthetic respondents under the Replicate and Reformulated conditions.

| | Current Economy | Gay Marriage | Refugee Allowing | Income Inequality | Gender Role | Climate Change | Gun Regulation | Drug Addiction | Race Diversity | Health Insurance | **Average** |
|---|---|---|---|---|---|---|---|---|---|---|---|
| Replicate | 2.55 | 2.30 | 2.57 | 2.57 | 2.33 | 2.26 | 2.41 | 2.22 | 2.33 | 2.72 | **2.43** |
| Reformulated | 2.54 | 2.78 | 2.46 | 2.44 | 3.18 | 2.61 | 2.83 | 2.87 | 2.68 | 2.50 | **2.69** |

Duplication rates are low and stable across questions, averaging 2.43% under Replicate and 2.69% under Reformulated, corresponding to roughly 132 and 146 of 5,441 respondents, respectively. Comparable rates were observed in the remaining experimental conditions. The vast majority of sampled profiles are therefore unique, and the small number of duplicates is unlikely to meaningfully affect our results, even under greedy decoding. Our main results rely on stochastic decoding, where duplicate profiles need not yield identical responses, which addresses this concern directly. We further note that some degree of duplication is expected and not inherently problematic: identical profiles arise when a demographic combination is genuinely common in the target population, so their repetition reflects realistic population frequencies rather than a sampling artifact.

### D.8 Non-response rates and their effects

Here, we examine whether the non-response handling in subsection B.2 affects our results. Of the 5,441 ANES 2020 respondents, 83.8% answer all ten items substantively, and non-response is concentrated in the remaining 16.2% rather than spread across the sample. Because non-response is concentrated in this subset rather than uniform across respondents, listwise deletion would remove a non-random group and could bias the composition of the retained sample, which available-case handling avoids.

As shown in Table 14, item non-response is moderate and similar across the eight policy items, between 12.8% and 13.7%, and near zero on *Gay Marriage* (1.3%) and *Current Economy* (0.3%), items on which respondents rarely decline to answer. These rates tell us how much is missing from each item, but not whether it is missing evenly across groups. Excluding non-response could distort demographic comparisons if some groups skipped questions more often than others. To check this, for each item we classify every respondent as substantive or non-substantive and test with a Chi-Square ($\chi^2$) test whether the non-response rate differs across the levels of each conditioning variable. Because the sample is large, significance is easy to reach, so alongside the count of items with $p < 0.05$ we report Cramér's $V$, which measures how much the non-response rate differs across groups, from 0 (no difference) to 1 (a strong difference). Table 15 shows these differences are small. Non-response is essentially uniform across Gender, and statistically detectable across Party and Race on nine and six of the ten items, but weak throughout. The largest $V$ across all 70 variable-by-item tests is 0.078, below 0.1, the conventional threshold for a small effect. Several of the remaining significant results come from sparse tables and are discounted, as the $\chi^2$ test is unreliable when

Table 14: Item non-response in ANES 2020, as a percentage of the 5,441 respondents.

| Item | Non-response (%) |
|---|---|
| Drug Addiction | 13.7 |
| Health Insurance | 13.7 |
| Race Diversity | 13.4 |
| Gender Role | 13.3 |
| Gun Regulation | 13.2 |
| Climate Change | 13.1 |
| Income Inequality | 13.0 |
| Refugee Allowing | 12.8 |
| Gay Marriage | 1.3 |
| Current Economy | 0.3 |

Table 15: Differential item non-response across the levels of each conditioning variable (ANES 2020). "Items differential" counts the items (of ten) with $p < 0.05$; Max $V$ is the largest Cramér's $V$ over the ten items. [†]For these variables the differential items come mostly from sparse tables (minimum expected count below five) and are not treated as reliable.

| Conditioning variable | Items differential | Max $V$ |
|---|---|---|
| Party | 9/10 | 0.078 |
| Race | 6/10 | 0.051 |
| Discuss Politics[†] | 3/10 | 0.049 |
| Political Interest[†] | 2/10 | 0.042 |
| Church | 1/10 | 0.044 |
| Ideology | 1/10 | 0.055 |
| Gender | 0/10 | 0.016 |

expected counts are very low. The excluded non-response is thus only mildly patterned across groups, and the contrasts are preserved up to small shifts on the party and racial dimensions.

As a further check, we recompute the JS-divergence with human non-response reintroduced as one additional category on both distributions. The silicon distribution then contains zero non-response, whereas the human distribution retains its non-responses. Given this asymmetric design, we compare only how the two treatments rank the group-item pairs by divergence. That ordering is stable: the rank correlation between the substantive-only and augmented divergences across the 290 group-item pairs is 0.90 (Spearman), so the groups and items the model reproduces relatively well or poorly are the same whether or not non-response is included. The substantive-only results therefore do not depend on the exclusion of non-response.

### D.9 Full stratified analysis by demographic subgroup

We present the full demographic subgroup analysis for GPT-4.1-mini, reporting both the change in JS-divergence ($\Delta$JSD) and the change in the absolute desirability gap ($\Delta|D|$) for each mitigation condition relative to the Replicate condition. For every condition the two metrics are shown as a pair of tables sharing the same column order, so a given subgroup and item can be read across both. In each table, items are ordered left to right by their overall average in JS-divergence, from low to high. *Current Economy* is omitted from the $\Delta|D|$ tables, as it has no desirability direction and $|D|$ is therefore undefined.

Table 16: Difference in JS-divergence (ΔJSD) between **Reformulated** and Replicate conditions on stratified subgroups using **GPT-4.1-mini**. Darker blue signifies greater improvement in alignment (ΔJSD < 0), while darker red signifies greater worsening (ΔJSD > 0).

| | | Race Diversity | Refugee Allowing | Gender Role | Climate Change | Drug Addiction | Gay Marriage | Gun Regulation | Health Insurance | Income Inequality | Average |
|---|---|---|---|---|---|---|---|---|---|---|---|
| Race | White | -0.082 | -0.06 | -0.048 | -0.042 | -0.024 | -0.009 | 0.012 | 0.023 | 0.043 | -0.021 |
| | Black | -0.083 | -0.053 | -0.07 | 0.008 | -0.011 | -0.003 | 0.015 | 0.015 | 0.08 | -0.011 |
| | Asian | -0.005 | -0.053 | -0.08 | 0.029 | -0.019 | 0.003 | 0.052 | 0.03 | 0.092 | 0.006 |
| | Native American | -0.006 | -0.105 | -0.07 | -0.108 | -0.035 | -0.03 | -0.053 | -0.035 | 0.08 | -0.04 |
| | Hispanic | -0.007 | -0.072 | -0.084 | 0.001 | -0.024 | 0.0 | 0.018 | 0.034 | 0.048 | -0.01 |
| Discuss Politics | Like to discuss politics | -0.059 | -0.053 | -0.065 | -0.032 | -0.022 | -0.005 | 0.018 | 0.035 | 0.045 | -0.015 |
| | Never discuss politics | -0.201 | -0.19 | -0.103 | -0.048 | -0.024 | -0.033 | -0.021 | -0.024 | 0.026 | -0.069 |
| Ideology | Extremely liberal | -0.021 | -0.011 | 0.251 | -0.005 | 0.0 | -0.016 | 0.02 | 0.014 | 0.037 | 0.03 |
| | Liberal | -0.016 | -0.029 | 0.0 | 0.026 | -0.005 | 0.0 | 0.027 | 0.014 | 0.049 | 0.007 |
| | Slightly liberal | -0.046 | -0.01 | -0.062 | 0.06 | -0.015 | -0.005 | 0.028 | 0.013 | 0.065 | 0.003 |
| | Moderate | -0.064 | -0.097 | -0.098 | -0.013 | -0.015 | -0.022 | 0.011 | 0.032 | 0.027 | -0.026 |
| | Slightly conservative | -0.125 | -0.103 | -0.082 | -0.109 | -0.04 | -0.012 | 0.005 | 0.058 | 0.001 | -0.045 |
| | Conservative | -0.154 | -0.063 | -0.07 | -0.1 | -0.04 | 0.007 | 0.027 | 0.016 | 0.051 | -0.036 |
| | Extremely conservative | 0.004 | -0.01 | -0.032 | -0.14 | 0.022 | 0.024 | 0.139 | 0.092 | 0.057 | 0.018 |
| Party | Strong democrat | -0.004 | -0.038 | -0.013 | -0.009 | 0.0 | 0.007 | 0.035 | 0.021 | 0.031 | 0.003 |
| | Weak democrat | -0.034 | -0.043 | -0.065 | 0.049 | -0.004 | 0.019 | -0.005 | 0.036 | 0.042 | -0.001 |
| | Democrat-leaning indep. | 0.0 | -0.052 | -0.039 | 0.011 | 0.0 | -0.005 | -0.032 | 0.021 | 0.031 | -0.007 |
| | Independent | -0.142 | -0.159 | -0.101 | -0.079 | -0.032 | 0.014 | -0.007 | 0.001 | 0.046 | -0.051 |
| | Republican-leaning indep. | -0.096 | -0.152 | -0.135 | -0.123 | -0.04 | -0.005 | 0.019 | 0.032 | -0.054 | -0.062 |
| | Weak republican | -0.129 | -0.099 | -0.059 | -0.076 | -0.019 | 0.013 | -0.007 | 0.015 | 0.002 | -0.04 |
| | Strong republican | -0.115 | 0.029 | -0.001 | 0.005 | 0.001 | -0.024 | 0.174 | 0.045 | 0.145 | 0.029 |
| Church | Attend church | -0.102 | -0.078 | -0.074 | -0.052 | -0.027 | -0.004 | 0.008 | 0.021 | 0.023 | -0.032 |
| | Does not attend church | -0.052 | -0.049 | -0.046 | -0.011 | -0.021 | -0.008 | 0.018 | 0.025 | 0.067 | -0.008 |
| Gender | Man | -0.089 | -0.057 | -0.072 | -0.049 | -0.023 | -0.008 | 0.009 | 0.019 | 0.043 | -0.025 |
| | Woman | -0.061 | -0.066 | -0.046 | -0.019 | -0.024 | -0.005 | 0.018 | 0.027 | 0.052 | -0.014 |
| Political Interest | Very | -0.043 | -0.021 | -0.043 | -0.048 | -0.004 | -0.008 | 0.035 | 0.026 | 0.045 | -0.007 |
| | Somewhat | -0.067 | -0.04 | -0.072 | -0.041 | -0.021 | -0.011 | 0.018 | 0.03 | 0.058 | -0.016 |
| | Not very | -0.082 | -0.1 | -0.084 | -0.015 | -0.03 | -0.011 | -0.009 | -0.001 | 0.022 | -0.035 |
| | Not at all | -0.146 | -0.047 | -0.063 | 0.018 | -0.044 | 0.003 | -0.006 | 0.021 | 0.011 | -0.028 |

Table 17: Change in the absolute desirability gap (Δ|D|) between **Reformulated** and Replicate conditions on stratified subgroups using **GPT-4.1-mini**. Darker blue signifies a reduced gap (Δ|D| < 0), darker red a widened gap (Δ|D| > 0). Items follow the same order as Table 16.

| | | Race Diversity | Refugee Allowing | Gender Role | Climate Change | Drug Addiction | Gay Marriage | Gun Regulation | Health Insurance | Income Inequality | Average |
|---|---|---|---|---|---|---|---|---|---|---|---|
| Race | White | -0.173 | -0.154 | 0.039 | -0.06 | -0.067 | -0.017 | -0.059 | 0.033 | -0.044 | -0.056 |
| | Black | -0.131 | -0.118 | -0.061 | -0.005 | -0.033 | -0.001 | 0.0 | 0.063 | -0.044 | -0.037 |
| | Asian | -0.134 | -0.124 | -0.087 | 0.097 | -0.082 | -0.052 | 0.133 | 0.075 | -0.056 | -0.026 |
| | Native American | -0.075 | -0.195 | -0.033 | -0.09 | -0.045 | -0.119 | -0.217 | -0.08 | -0.024 | -0.097 |
| | Hispanic | -0.107 | -0.166 | -0.066 | 0.007 | -0.049 | 0.02 | 0.065 | 0.03 | -0.047 | -0.035 |
| Discuss Politics | Like to discuss politics | -0.152 | -0.138 | 0.005 | -0.028 | -0.062 | -0.004 | -0.004 | 0.056 | -0.042 | -0.041 |
| | Never discuss politics | -0.18 | -0.287 | -0.051 | -0.148 | -0.047 | -0.078 | -0.12 | -0.031 | -0.114 | -0.117 |
| Ideology | Extremely liberal | -0.065 | -0.074 | 0.601 | 0.001 | 0.0 | -0.011 | 0.005 | 0.022 | -0.03 | 0.05 |
| | Liberal | -0.019 | -0.077 | 0.183 | 0.042 | -0.001 | 0.0 | 0.023 | 0.023 | -0.049 | 0.014 |
| | Slightly liberal | -0.059 | -0.104 | -0.022 | 0.144 | -0.02 | -0.026 | 0.086 | 0.034 | -0.068 | -0.004 |
| | Moderate | -0.034 | -0.182 | -0.008 | 0.046 | -0.011 | -0.056 | 0.065 | 0.053 | -0.013 | -0.016 |
| | Slightly conservative | -0.24 | -0.171 | -0.009 | -0.181 | -0.058 | -0.045 | -0.089 | 0.081 | -0.152 | -0.096 |
| | Conservative | -0.394 | -0.178 | -0.009 | -0.2 | -0.166 | 0.049 | 0.028 | -0.015 | -0.014 | -0.1 |
| | Extremely conservative | -0.392 | -0.064 | -0.027 | -0.155 | -0.265 | 0.11 | 0.092 | 0.11 | 0.047 | -0.06 |
| Party | Strong democrat | -0.002 | -0.028 | 0.113 | -0.028 | 0.0 | 0.003 | 0.031 | 0.011 | 0.021 | 0.014 |
| | Weak democrat | -0.017 | -0.067 | -0.046 | 0.137 | -0.002 | 0.028 | 0.042 | 0.07 | -0.016 | 0.015 |
| | Democrat-leaning indep. | 0.0 | -0.041 | -0.005 | 0.049 | 0.0 | -0.002 | -0.051 | 0.008 | 0.025 | -0.002 |
| | Independent | -0.109 | -0.234 | -0.019 | -0.111 | -0.019 | 0.031 | -0.037 | 0.005 | -0.038 | -0.059 |
| | Republican-leaning indep. | -0.206 | -0.302 | -0.027 | -0.184 | -0.042 | -0.011 | 0.116 | 0.084 | -0.111 | -0.076 |
| | Weak republican | -0.222 | -0.175 | 0.013 | -0.14 | -0.037 | 0.035 | -0.004 | 0.032 | -0.06 | -0.062 |
| | Strong republican | -0.476 | 0.03 | -0.022 | -0.053 | -0.247 | -0.114 | -0.006 | 0.061 | 0.07 | -0.084 |
| Church | Attend church | -0.175 | -0.161 | 0.013 | -0.139 | -0.071 | -0.002 | -0.068 | 0.029 | -0.055 | -0.07 |
| | Does not attend church | -0.143 | -0.144 | 0.006 | 0.047 | -0.054 | -0.029 | 0.061 | 0.044 | -0.033 | -0.027 |
| Gender | Man | -0.161 | -0.135 | -0.032 | -0.067 | -0.067 | -0.019 | -0.072 | 0.035 | -0.021 | -0.06 |
| | Woman | -0.157 | -0.167 | 0.042 | -0.022 | -0.058 | -0.008 | 0.02 | 0.033 | -0.06 | -0.042 |
| Political Interest | Very | -0.175 | -0.102 | 0.052 | -0.111 | -0.084 | -0.037 | -0.022 | 0.032 | -0.016 | -0.051 |
| | Somewhat | -0.157 | -0.095 | 0.004 | -0.024 | -0.066 | -0.017 | -0.011 | 0.049 | -0.037 | -0.039 |
| | Not very | -0.124 | -0.238 | -0.025 | 0.017 | -0.028 | -0.036 | -0.041 | -0.001 | -0.096 | -0.063 |
| | Not at all | -0.158 | -0.243 | -0.048 | -0.043 | -0.027 | 0.024 | -0.072 | 0.057 | -0.133 | -0.071 |

Table 18: Difference in JS-divergence (ΔJSD) between **Reverse-coded** and Replicate conditions on stratified subgroups using **GPT-4.1-mini**. Darker blue signifies greater improvement in alignment (ΔJSD < 0), while darker red signifies greater worsening (ΔJSD > 0). Only the six reverse-coded items are shown.

| | | Race Diversity | Refugee Allowing | Income Inequality | Gay Marriage | Climate Change | Gender Role | Average |
|---|---|---|---|---|---|---|---|---|
| Race | White | -0.147 | -0.038 | -0.034 | 0.058 | 0.263 | 0.807 | 0.151 |
| | Black | -0.183 | -0.068 | -0.018 | 0.073 | 0.269 | 0.696 | 0.128 |
| | Asian | -0.123 | -0.048 | 0.001 | 0.079 | 0.304 | 0.665 | 0.146 |
| | Native American | -0.16 | -0.051 | -0.059 | 0.048 | 0.273 | 0.702 | 0.126 |
| | Hispanic | -0.142 | -0.08 | -0.028 | 0.063 | 0.298 | 0.7 | 0.135 |
| Discuss Politics | Like to discuss politics | -0.142 | -0.043 | -0.027 | 0.061 | 0.258 | 0.773 | 0.147 |
| | Never discuss politics | -0.243 | -0.12 | -0.095 | 0.048 | 0.381 | 0.716 | 0.115 |
| Ideology | Extremely liberal | -0.019 | -0.018 | 0.153 | 0.0 | 0.087 | 0.715 | 0.153 |
| | Liberal | -0.021 | 0.002 | 0.128 | -0.002 | 0.163 | 0.688 | 0.16 |
| | Slightly liberal | -0.072 | -0.003 | 0.062 | 0.02 | 0.209 | 0.734 | 0.158 |
| | Moderate | -0.171 | -0.096 | -0.044 | 0.036 | 0.336 | 0.744 | 0.134 |
| | Slightly conservative | -0.185 | -0.135 | -0.04 | 0.1 | 0.427 | 0.788 | 0.159 |
| | Conservative | -0.278 | -0.026 | 0.169 | 0.159 | 0.406 | 0.854 | 0.214 |
| | Extremely conservative | -0.23 | -0.02 | 0.315 | 0.073 | 0.182 | 0.836 | 0.193 |
| Party | Strong democrat | -0.059 | -0.038 | 0.051 | -0.023 | 0.162 | 0.683 | 0.129 |
| | Weak democrat | -0.09 | -0.06 | 0.017 | 0.035 | 0.237 | 0.702 | 0.14 |
| | Democrat-leaning indep. | -0.07 | -0.061 | 0.071 | -0.014 | 0.214 | 0.698 | 0.139 |
| | Independent | -0.152 | -0.159 | -0.081 | 0.053 | 0.352 | 0.714 | 0.121 |
| | Republican-leaning indep. | -0.186 | -0.098 | 0.065 | 0.078 | 0.383 | 0.752 | 0.166 |
| | Weak republican | -0.175 | -0.077 | -0.095 | 0.095 | 0.438 | 0.835 | 0.17 |
| | Strong republican | -0.293 | 0.076 | 0.421 | 0.109 | 0.395 | 0.87 | 0.263 |
| Church | Attend church | -0.167 | -0.053 | -0.062 | 0.087 | 0.313 | 0.786 | 0.151 |
| | Does not attend church | -0.133 | -0.036 | -0.007 | 0.039 | 0.229 | 0.767 | 0.143 |
| Gender | Man | -0.171 | -0.048 | -0.024 | 0.063 | 0.28 | 0.773 | 0.145 |
| | Woman | -0.128 | -0.042 | -0.037 | 0.056 | 0.257 | 0.777 | 0.147 |
| Political Interest | Very | -0.112 | -0.012 | -0.017 | 0.049 | 0.155 | 0.77 | 0.139 |
| | Somewhat | -0.15 | -0.044 | -0.028 | 0.061 | 0.256 | 0.773 | 0.144 |
| | Not very | -0.168 | -0.084 | -0.061 | 0.058 | 0.386 | 0.766 | 0.149 |
| | Not at all | -0.173 | -0.074 | -0.054 | 0.035 | 0.408 | 0.733 | 0.146 |

Table 19: Change in the absolute desirability gap ($\Delta|D|$) between **Reverse-coded** and Replicate conditions on stratified subgroups using **GPT-4.1-mini**. Darker blue signifies a reduced gap ($\Delta|D| < 0$), darker red a widened gap ($\Delta|D| > 0$). Items follow the same order as Table 18. Only the six reverse-coded items are shown.

| | | Race Diversity | Refugee Allowing | Income Inequal-ity | Gay Marriage | Climate Change | Gender Role | Average |
|---|---|---|---|---|---|---|---|---|
| Race | White | -0.137 | -0.069 | -0.151 | -0.051 | 0.211 | 0.921 | 0.121 |
| | Black | -0.135 | -0.136 | -0.18 | 0.054 | 0.135 | 0.85 | 0.098 |
| | Asian | -0.134 | -0.1 | -0.123 | -0.036 | 0.183 | 0.792 | 0.097 |
| | Native American | -0.162 | -0.072 | -0.28 | -0.082 | 0.235 | 0.847 | 0.081 |
| | Hispanic | -0.14 | -0.165 | -0.216 | 0.048 | 0.212 | 0.845 | 0.097 |
| Discuss Politics | Like to discuss politics | -0.126 | -0.082 | -0.144 | -0.029 | 0.205 | 0.901 | 0.121 |
| | Never discuss politics | -0.244 | -0.14 | -0.408 | -0.014 | 0.216 | 0.871 | 0.047 |
| Ideology | Extremely liberal | -0.008 | -0.027 | 0.208 | 0.0 | 0.038 | 0.903 | 0.186 |
| | Liberal | -0.013 | -0.039 | 0.175 | -0.003 | 0.087 | 0.83 | 0.173 |
| | Slightly liberal | -0.05 | -0.081 | 0.048 | -0.028 | 0.124 | 0.862 | 0.146 |
| | Moderate | -0.145 | -0.155 | -0.122 | -0.029 | 0.243 | 0.912 | 0.117 |
| | Slightly conservative | -0.128 | -0.212 | -0.085 | 0.073 | 0.284 | 0.921 | 0.142 |
| | Conservative | -0.276 | -0.041 | 0.305 | -0.059 | 0.319 | 0.958 | 0.201 |
| | Extremely conservative | -0.404 | -0.049 | 0.486 | -0.21 | 0.173 | 0.914 | 0.152 |
| Party | Strong democrat | -0.041 | -0.026 | 0.047 | -0.029 | 0.085 | 0.871 | 0.151 |
| | Weak democrat | -0.069 | -0.116 | 0.077 | -0.052 | 0.159 | 0.858 | 0.143 |
| | Democrat-leaning indep. | -0.047 | -0.054 | 0.141 | -0.011 | 0.134 | 0.873 | 0.173 |
| | Independent | -0.103 | -0.246 | -0.309 | -0.023 | 0.199 | 0.879 | 0.066 |
| | Republican-leaning indep. | -0.188 | -0.196 | 0.081 | -0.004 | 0.3 | 0.931 | 0.154 |
| | Weak republican | -0.156 | -0.161 | -0.205 | 0.003 | 0.289 | 0.944 | 0.119 |
| | Strong republican | -0.32 | 0.038 | 0.714 | -0.029 | 0.309 | 0.942 | 0.276 |
| Church | Attend church | -0.125 | -0.076 | -0.177 | 0.012 | 0.222 | 0.918 | 0.129 |
| | Does not attend church | -0.149 | -0.094 | -0.115 | -0.063 | 0.189 | 0.88 | 0.108 |
| Gender | Man | -0.17 | -0.089 | -0.124 | -0.031 | 0.218 | 0.899 | 0.117 |
| | Woman | -0.109 | -0.08 | -0.205 | -0.022 | 0.194 | 0.897 | 0.113 |
| Political Interest | Very | -0.084 | -0.024 | -0.125 | -0.022 | 0.131 | 0.888 | 0.127 |
| | Somewhat | -0.129 | -0.089 | -0.123 | -0.026 | 0.23 | 0.902 | 0.127 |
| | Not very | -0.192 | -0.166 | -0.289 | -0.029 | 0.277 | 0.907 | 0.085 |
| | Not at all | -0.173 | -0.218 | -0.264 | -0.044 | 0.222 | 0.87 | 0.065 |

Table 20: Difference in JS-divergence (ΔJSD) between **Priming** and Replicate conditions on stratified subgroups using **GPT-4.1-mini**. Darker blue signifies greater improvement in alignment (ΔJSD < 0), while darker red signifies greater worsening (ΔJSD > 0).

| | | Gay Marriage | Gun Regulation | Drug Addiction | Refugee Allowing | Health Insurance | Race Diversity | Income Inequality | Gender Role | Climate Change | Average |
|---|---|---|---|---|---|---|---|---|---|---|---|
| Race | White | -0.02 | 0.008 | 0.009 | 0.009 | 0.036 | 0.042 | 0.04 | 0.115 | 0.156 | 0.044 |
| | Black | 0.017 | 0.009 | 0.022 | 0.033 | 0.017 | 0.0 | 0.009 | 0.107 | 0.127 | 0.038 |
| | Asian | -0.021 | 0.003 | 0.0 | 0.013 | 0.03 | 0.039 | 0.013 | 0.066 | 0.096 | 0.027 |
| | Native American | 0.03 | -0.007 | 0.0 | 0.064 | -0.051 | 0.051 | 0.053 | 0.113 | 0.162 | 0.046 |
| | Hispanic | 0.0 | 0.001 | 0.005 | 0.019 | 0.044 | 0.053 | 0.027 | 0.102 | 0.135 | 0.043 |
| Discuss Politics | Like to discuss politics | -0.012 | 0.005 | 0.011 | 0.019 | 0.029 | 0.033 | 0.04 | 0.106 | 0.141 | 0.041 |
| | Never discuss politics | -0.003 | 0.006 | 0.01 | -0.009 | 0.006 | 0.05 | 0.024 | 0.077 | 0.216 | 0.042 |
| Ideology | Extremely liberal | 0.0 | 0.02 | 0.0 | 0.008 | 0.034 | 0.0 | 0.047 | 0.051 | 0.066 | 0.025 |
| | Liberal | 0.004 | -0.011 | 0.0 | 0.017 | 0.029 | 0.008 | 0.056 | 0.018 | 0.065 | 0.021 |
| | Slightly liberal | -0.013 | 0.007 | 0.0 | 0.032 | 0.056 | 0.0 | 0.083 | 0.051 | 0.043 | 0.029 |
| | Moderate | -0.015 | 0.009 | 0.0 | 0.032 | 0.054 | 0.016 | 0.061 | 0.065 | 0.122 | 0.038 |
| | Slightly conservative | 0.013 | 0.021 | 0.0 | 0.05 | 0.087 | 0.008 | 0.103 | 0.093 | 0.272 | 0.072 |
| | Conservative | -0.017 | 0.007 | 0.004 | 0.057 | 0.028 | 0.063 | 0.078 | 0.204 | 0.355 | 0.087 |
| | Extremely conservative | -0.075 | -0.021 | 0.069 | -0.022 | 0.051 | 0.226 | 0.026 | 0.26 | 0.395 | 0.101 |
| Party | Strong democrat | 0.003 | -0.005 | 0.0 | -0.004 | 0.017 | 0.006 | 0.013 | 0.028 | 0.093 | 0.017 |
| | Weak democrat | 0.02 | 0.007 | 0.0 | 0.016 | 0.052 | 0.016 | 0.046 | 0.052 | 0.088 | 0.033 |
| | Democrat-leaning indep. | 0.0 | -0.024 | 0.0 | -0.01 | 0.008 | 0.0 | 0.018 | 0.005 | 0.055 | 0.006 |
| | Independent | 0.014 | 0.042 | 0.0 | -0.014 | 0.047 | 0.027 | 0.024 | 0.073 | 0.199 | 0.046 |
| | Republican-leaning indep. | 0.017 | 0.013 | 0.0 | 0.029 | 0.095 | 0.047 | 0.15 | 0.098 | 0.239 | 0.076 |
| | Weak republican | 0.013 | 0.013 | 0.016 | 0.025 | 0.032 | 0.057 | 0.046 | 0.138 | 0.262 | 0.067 |
| | Strong republican | -0.078 | -0.055 | 0.029 | 0.021 | 0.026 | 0.099 | 0.052 | 0.225 | 0.337 | 0.073 |
| Church | Attend church | -0.006 | 0.008 | 0.005 | 0.02 | 0.04 | 0.016 | 0.04 | 0.123 | 0.169 | 0.046 |
| | Does not attend church | -0.022 | 0.004 | 0.013 | 0.005 | 0.021 | 0.054 | 0.023 | 0.102 | 0.125 | 0.036 |
| Gender | Man | -0.018 | 0.009 | 0.012 | 0.011 | 0.041 | 0.032 | 0.044 | 0.121 | 0.151 | 0.045 |
| | Woman | -0.008 | 0.003 | 0.007 | 0.015 | 0.022 | 0.043 | 0.025 | 0.103 | 0.141 | 0.039 |
| Political Interest | Very | -0.035 | 0.007 | 0.012 | -0.008 | 0.023 | 0.007 | 0.036 | 0.119 | 0.116 | 0.031 |
| | Somewhat | -0.015 | 0.002 | 0.009 | 0.021 | 0.021 | 0.029 | 0.035 | 0.092 | 0.117 | 0.035 |
| | Not very | 0.004 | 0.009 | 0.005 | 0.035 | 0.046 | 0.067 | 0.032 | 0.099 | 0.189 | 0.054 |
| | Not at all | 0.008 | 0.025 | 0.0 | 0.013 | 0.025 | 0.125 | 0.031 | 0.133 | 0.242 | 0.067 |

Table 21: Change in the absolute desirability gap (Δ|D|) between **Priming** and Replicate conditions on stratified subgroups using **GPT-4.1-mini**. Darker blue signifies a reduced gap (Δ|D| < 0), darker red a widened gap (Δ|D| > 0). Items follow the same order as Table 20.

| | | Gay Marriage | Gun Regulation | Drug Addiction | Refugee Allowing | Health Insurance | Race Diversity | Income Inequality | Gender Role | Climate Change | Average |
|---|---|---|---|---|---|---|---|---|---|---|---|
| Race | White | -0.008 | 0.034 | 0.005 | 0.032 | 0.081 | 0.015 | 0.083 | 0.006 | 0.151 | 0.044 |
| | Black | 0.035 | -0.001 | 0.01 | 0.04 | 0.052 | 0.0 | 0.025 | 0.012 | 0.092 | 0.029 |
| | Asian | -0.019 | 0.017 | 0.0 | 0.027 | 0.085 | 0.015 | 0.065 | 0.005 | 0.124 | 0.035 |
| | Native American | 0.037 | -0.088 | 0.0 | 0.064 | -0.075 | 0.019 | 0.096 | 0.008 | 0.12 | 0.02 |
| | Hispanic | 0.021 | -0.002 | 0.002 | 0.041 | 0.088 | 0.02 | 0.066 | 0.0 | 0.137 | 0.041 |
| Discuss Politics | Like to discuss politics | 0.007 | 0.035 | 0.005 | 0.048 | 0.08 | 0.011 | 0.093 | 0.005 | 0.145 | 0.048 |
| | Never discuss politics | 0.004 | -0.004 | 0.005 | -0.011 | -0.001 | 0.017 | 0.02 | 0.003 | 0.129 | 0.018 |
| Ideology | Extremely liberal | 0.0 | 0.096 | 0.0 | 0.011 | 0.043 | 0.0 | 0.057 | 0.083 | 0.035 | 0.036 |
| | Liberal | 0.004 | -0.056 | 0.0 | 0.014 | 0.038 | 0.003 | 0.089 | 0.006 | 0.066 | 0.018 |
| | Slightly liberal | -0.018 | -0.029 | 0.0 | 0.054 | 0.106 | 0.0 | 0.143 | 0.003 | 0.051 | 0.035 |
| | Moderate | -0.033 | 0.027 | 0.0 | 0.024 | 0.064 | 0.004 | 0.088 | 0.0 | 0.125 | 0.033 |
| | Slightly conservative | 0.04 | 0.036 | 0.0 | 0.044 | 0.115 | 0.002 | 0.134 | 0.0 | 0.154 | 0.058 |
| | Conservative | 0.024 | -0.007 | 0.001 | 0.082 | 0.064 | 0.023 | 0.09 | 0.0 | 0.251 | 0.059 |
| | Extremely conservative | 0.025 | -0.074 | 0.08 | -0.042 | 0.141 | 0.128 | 0.043 | 0.0 | 0.316 | 0.069 |
| Party | Strong democrat | 0.002 | -0.011 | 0.0 | -0.002 | 0.01 | 0.002 | 0.011 | 0.016 | 0.072 | 0.011 |
| | Weak democrat | 0.025 | 0.061 | 0.0 | 0.016 | 0.077 | 0.005 | 0.065 | 0.009 | 0.084 | 0.038 |
| | Democrat-leaning indep. | 0.0 | -0.042 | 0.0 | -0.005 | 0.004 | 0.0 | 0.015 | 0.002 | 0.055 | 0.003 |
| | Independent | 0.03 | 0.076 | 0.0 | -0.018 | 0.052 | 0.008 | 0.028 | 0.002 | 0.113 | 0.032 |
| | Republican-leaning indep. | 0.012 | -0.05 | 0.0 | 0.021 | 0.132 | 0.017 | 0.152 | 0.0 | 0.162 | 0.05 |
| | Weak republican | 0.023 | 0.061 | 0.009 | 0.032 | 0.036 | 0.02 | 0.046 | 0.002 | 0.168 | 0.044 |
| | Strong republican | -0.019 | -0.088 | 0.018 | 0.138 | 0.212 | 0.04 | 0.231 | 0.002 | 0.285 | 0.091 |
| Church | Attend church | 0.009 | 0.009 | 0.002 | 0.035 | 0.066 | 0.004 | 0.065 | 0.002 | 0.145 | 0.037 |
| | Does not attend church | -0.012 | 0.049 | 0.007 | 0.031 | 0.085 | 0.023 | 0.079 | 0.009 | 0.14 | 0.046 |
| Gender | Man | -0.012 | 0.017 | 0.006 | 0.033 | 0.102 | 0.011 | 0.101 | 0.004 | 0.143 | 0.045 |
| | Woman | 0.013 | 0.038 | 0.003 | 0.033 | 0.051 | 0.015 | 0.052 | 0.007 | 0.142 | 0.039 |
| Political Interest | Very | -0.024 | 0.048 | 0.008 | 0.021 | 0.087 | 0.002 | 0.096 | 0.009 | 0.099 | 0.039 |
| | Somewhat | -0.011 | 0.02 | 0.004 | 0.049 | 0.062 | 0.009 | 0.081 | 0.002 | 0.148 | 0.04 |
| | Not very | 0.013 | 0.031 | 0.001 | 0.042 | 0.083 | 0.029 | 0.053 | 0.001 | 0.172 | 0.047 |
| | Not at all | 0.071 | 0.022 | 0.0 | 0.017 | 0.047 | 0.058 | 0.041 | 0.008 | 0.183 | 0.05 |

Table 22: Difference in JS-divergence (ΔJSD) between **Preamble** and Replicate conditions on stratified subgroups using **GPT-4.1-mini**. Darker blue signifies greater improvement in alignment (ΔJSD < 0), while darker red signifies greater worsening (ΔJSD > 0).

| | | Race Diversity | Drug Addiction | Gun Regulation | Gay Marriage | Climate Change | Gender Role | Income Inequality | Health Insurance | Refugee Allowing | Average |
|---|---|---|---|---|---|---|---|---|---|---|---|
| Race | White | -0.014 | 0.009 | 0.011 | 0.007 | 0.023 | 0.045 | 0.047 | 0.129 | 0.132 | 0.043 |
| | Black | -0.008 | 0.022 | -0.004 | 0.025 | 0.021 | 0.04 | 0.026 | 0.071 | 0.177 | 0.041 |
| | Asian | 0.011 | 0.0 | 0.017 | 0.009 | 0.01 | 0.018 | -0.001 | 0.145 | 0.154 | 0.04 |
| | Native American | 0.004 | 0.0 | 0.005 | 0.001 | -0.013 | 0.034 | 0.017 | 0.067 | 0.151 | 0.03 |
| | Hispanic | 0.004 | 0.005 | 0.008 | 0.012 | 0.041 | 0.045 | 0.016 | 0.126 | 0.109 | 0.041 |
| Discuss Politics | Like to discuss politics | -0.007 | 0.011 | 0.011 | 0.012 | 0.022 | 0.041 | 0.043 | 0.13 | 0.134 | 0.044 |
| | Never discuss politics | -0.059 | 0.01 | 0.002 | 0.014 | 0.038 | 0.004 | -0.017 | 0.108 | 0.161 | 0.029 |
| Ideology | Extremely liberal | 0.0 | 0.0 | 0.035 | 0.0 | 0.046 | 0.017 | 0.051 | 0.043 | 0.031 | 0.025 |
| | Liberal | 0.003 | 0.0 | -0.006 | -0.001 | 0.031 | 0.014 | 0.042 | 0.054 | 0.037 | 0.019 |
| | Slightly liberal | -0.01 | 0.0 | -0.003 | 0.004 | -0.002 | 0.051 | 0.043 | 0.093 | 0.097 | 0.03 |
| | Moderate | -0.002 | 0.0 | 0.004 | 0.021 | 0.016 | 0.044 | 0.072 | 0.138 | 0.156 | 0.05 |
| | Slightly conservative | -0.012 | 0.0 | 0.013 | 0.036 | 0.081 | 0.041 | 0.098 | 0.242 | 0.197 | 0.077 |
| | Conservative | -0.04 | 0.004 | 0.013 | 0.016 | 0.053 | 0.043 | 0.096 | 0.253 | 0.259 | 0.077 |
| | Extremely conservative | -0.055 | 0.069 | 0.037 | -0.011 | 0.041 | 0.039 | 0.054 | 0.19 | 0.153 | 0.057 |
| Party | Strong democrat | 0.006 | 0.0 | 0.021 | 0.01 | 0.031 | 0.015 | 0.02 | 0.025 | 0.015 | 0.016 |
| | Weak democrat | -0.004 | 0.0 | 0.012 | 0.033 | 0.002 | 0.031 | 0.04 | 0.101 | 0.084 | 0.033 |
| | Democrat-leaning indep. | 0.0 | 0.0 | 0.019 | 0.0 | 0.005 | 0.005 | 0.04 | 0.021 | 0.014 | 0.012 |
| | Independent | -0.005 | 0.0 | 0.023 | 0.036 | 0.018 | 0.016 | 0.046 | 0.143 | 0.114 | 0.044 |
| | Republican-leaning indep. | -0.017 | 0.0 | -0.015 | 0.03 | 0.044 | 0.017 | 0.107 | 0.266 | 0.147 | 0.064 |
| | Weak republican | -0.018 | 0.016 | 0.014 | 0.034 | 0.06 | 0.062 | 0.089 | 0.201 | 0.184 | 0.071 |
| | Strong republican | -0.025 | 0.029 | -0.037 | -0.027 | 0.056 | 0.08 | 0.06 | 0.195 | 0.27 | 0.067 |
| Church | Attend church | -0.026 | 0.005 | 0.009 | 0.022 | 0.029 | 0.037 | 0.047 | 0.141 | 0.146 | 0.046 |
| | Does not attend church | -0.001 | 0.013 | 0.008 | -0.001 | 0.014 | 0.054 | 0.024 | 0.102 | 0.119 | 0.037 |
| Gender | Man | -0.031 | 0.012 | 0.012 | 0.006 | 0.014 | 0.036 | 0.04 | 0.115 | 0.13 | 0.037 |
| | Woman | 0.005 | 0.007 | 0.007 | 0.015 | 0.028 | 0.055 | 0.036 | 0.132 | 0.137 | 0.047 |
| Political Interest | Very | -0.018 | 0.012 | 0.011 | -0.002 | 0.021 | 0.033 | 0.038 | 0.096 | 0.077 | 0.03 |
| | Somewhat | -0.034 | 0.009 | 0.006 | 0.004 | 0.023 | 0.037 | 0.035 | 0.1 | 0.13 | 0.035 |
| | Not very | 0.021 | 0.005 | 0.016 | 0.019 | 0.017 | 0.031 | 0.054 | 0.195 | 0.203 | 0.062 |
| | Not at all | 0.049 | 0.0 | 0.034 | 0.042 | 0.038 | 0.097 | 0.064 | 0.21 | 0.218 | 0.083 |

Table 23: Change in the absolute desirability gap (Δ|D|) between **Preamble** and Replicate conditions on stratified subgroups using **GPT-4.1-mini**. Darker blue signifies a reduced gap (Δ|D| < 0), darker red a widened gap (Δ|D| > 0). Items follow the same order as Table 22.

| | | Race Diversity | Drug Addiction | Gun Regulation | Gay Marriage | Climate Change | Gender Role | Income Inequality | Health Insurance | Refugee Allowing | Average |
|---|---|---|---|---|---|---|---|---|---|---|---|
| Race | White | -0.007 | 0.005 | 0.068 | 0.048 | 0.068 | 0.004 | 0.083 | 0.189 | 0.149 | 0.067 |
| | Black | -0.002 | 0.01 | -0.059 | 0.045 | 0.014 | 0.01 | 0.06 | 0.169 | 0.154 | 0.045 |
| | Asian | 0.004 | 0.0 | -0.035 | 0.024 | 0.017 | -0.001 | 0.016 | 0.215 | 0.132 | 0.041 |
| | Native American | 0.002 | 0.0 | 0.014 | -0.011 | 0.078 | -0.002 | 0.011 | 0.058 | 0.118 | 0.03 |
| | Hispanic | 0.002 | 0.002 | 0.07 | 0.064 | 0.075 | 0.0 | 0.042 | 0.183 | 0.112 | 0.061 |
| Discuss Politics | Like to discuss politics | -0.003 | 0.005 | 0.079 | 0.057 | 0.069 | 0.003 | 0.083 | 0.201 | 0.15 | 0.072 |
| | Never discuss politics | -0.033 | 0.005 | 0.043 | 0.019 | 0.058 | 0.003 | -0.036 | 0.114 | 0.098 | 0.03 |
| Ideology | Extremely liberal | 0.0 | 0.0 | 0.127 | 0.0 | 0.031 | 0.056 | 0.058 | 0.047 | 0.023 | 0.038 |
| | Liberal | 0.001 | 0.0 | -0.035 | 0.001 | 0.048 | 0.005 | 0.076 | 0.053 | 0.023 | 0.019 |
| | Slightly liberal | -0.003 | 0.0 | -0.045 | 0.013 | -0.001 | 0.003 | 0.086 | 0.135 | 0.098 | 0.032 |
| | Moderate | -0.001 | 0.0 | 0.053 | 0.028 | 0.067 | 0.0 | 0.085 | 0.151 | 0.101 | 0.054 |
| | Slightly conservative | -0.004 | 0.0 | 0.07 | 0.084 | 0.082 | -0.002 | 0.098 | 0.245 | 0.13 | 0.078 |
| | Conservative | -0.02 | 0.001 | 0.053 | 0.102 | 0.082 | 0.0 | 0.086 | 0.343 | 0.304 | 0.106 |
| | Extremely conservative | -0.052 | 0.08 | 0.088 | 0.127 | 0.063 | 0.0 | 0.106 | 0.345 | 0.339 | 0.122 |
| Party | Strong democrat | 0.002 | 0.0 | 0.022 | 0.004 | 0.042 | 0.014 | 0.013 | 0.012 | 0.005 | 0.013 |
| | Weak democrat | -0.002 | 0.0 | 0.097 | 0.038 | 0.023 | 0.005 | 0.066 | 0.118 | 0.053 | 0.044 |
| | Democrat-leaning indep. | 0.0 | 0.0 | 0.022 | 0.0 | 0.042 | 0.002 | 0.03 | 0.008 | 0.004 | 0.012 |
| | Independent | -0.002 | 0.0 | 0.079 | 0.061 | 0.04 | -0.001 | 0.057 | 0.137 | 0.068 | 0.049 |
| | Republican-leaning indep. | -0.008 | 0.0 | 0.0 | 0.038 | 0.076 | 0.0 | 0.102 | 0.314 | 0.106 | 0.07 |
| | Weak republican | -0.009 | 0.009 | 0.113 | 0.076 | 0.089 | 0.002 | 0.097 | 0.226 | 0.152 | 0.084 |
| | Strong republican | -0.014 | 0.018 | -0.111 | 0.117 | 0.106 | 0.001 | 0.192 | 0.477 | 0.486 | 0.141 |
| Church | Attend church | -0.01 | 0.002 | 0.051 | 0.061 | 0.061 | 0.001 | 0.067 | 0.167 | 0.134 | 0.059 |
| | Does not attend church | 0.0 | 0.007 | 0.089 | 0.036 | 0.066 | 0.007 | 0.071 | 0.204 | 0.15 | 0.07 |
| Gender | Man | -0.015 | 0.006 | 0.071 | 0.038 | 0.059 | 0.002 | 0.079 | 0.195 | 0.15 | 0.065 |
| | Woman | 0.002 | 0.003 | 0.069 | 0.059 | 0.068 | 0.005 | 0.064 | 0.177 | 0.137 | 0.065 |
| Political Interest | Very | -0.006 | 0.008 | 0.08 | 0.048 | 0.056 | 0.006 | 0.077 | 0.179 | 0.117 | 0.063 |
| | Somewhat | -0.016 | 0.004 | 0.044 | 0.033 | 0.071 | 0.0 | 0.059 | 0.166 | 0.148 | 0.057 |
| | Not very | 0.011 | 0.001 | 0.095 | 0.043 | 0.066 | 0.0 | 0.072 | 0.232 | 0.156 | 0.075 |
| | Not at all | 0.029 | 0.0 | 0.137 | 0.099 | 0.056 | 0.008 | 0.111 | 0.235 | 0.202 | 0.097 |

### D.10 Confidence intervals for JS-divergence in ANES 2020

Tables 24, 25, and 26 report the mean JS-divergence and 95% bootstrap confidence intervals for ANES 2020 across three models.

Table 24: Mean JS divergence and 95% bootstrap confidence intervals between ANES 2020 and silicon distributions by item and condition, using Llama-8B. Intervals come from a paired bootstrap that resamples both silicon and ANES responses ($n = 10{,}000$) and are reported for description only.

| Question | Replicate | Reformulated | Reverse-Coded | Priming | Preamble |
|---|---|---|---|---|---|
| Climate Change | 0.3631 [0.3508, 0.3758] | 0.3389 [0.3273, 0.3513] | 0.1142 [0.1049, 0.1242] | 0.4140 [0.4022, 0.4261] | 0.3693 [0.3569, 0.3821] |
| Current Economy | 0.2566 [0.2449, 0.2687] | 0.1223 [0.1147, 0.1305] | N/A | 0.2958 [0.2837, 0.3086] | 0.2025 [0.1918, 0.2138] |
| Refugee Allowing | 0.2215 [0.2117, 0.2319] | 0.1948 [0.1861, 0.2036] | 0.2727 [0.2596, 0.2863] | 0.2039 [0.1947, 0.2136] | 0.1840 [0.1754, 0.1930] |
| Health Insurance | 0.2265 [0.2158, 0.2379] | 0.1749 [0.1656, 0.1846] | N/A | 0.2090 [0.1992, 0.2193] | 0.2133 [0.2026, 0.2248] |
| Income Inequality | 0.1447 [0.1337, 0.1561] | 0.1433 [0.1349, 0.1521] | 0.0621 [0.0545, 0.0702] | 0.1758 [0.1650, 0.1874] | 0.1305 [0.1203, 0.1417] |
| Race Diversity | 0.1219 [0.1117, 0.1325] | 0.1077 [0.0984, 0.1178] | 0.1694 [0.1586, 0.1808] | 0.0021 [0.0010, 0.0039] | 0.0246 [0.0201, 0.0299] |
| Drug Addiction | 0.1548 [0.1462, 0.1635] | 0.1187 [0.1100, 0.1278] | N/A | 0.1429 [0.1348, 0.1511] | 0.0999 [0.0915, 0.1085] |
| Gay Marriage | 0.1376 [0.1286, 0.1466] | 0.0293 [0.0243, 0.0349] | 0.1721 [0.1644, 0.1798] | 0.1702 [0.1622, 0.1785] | 0.1721 [0.1644, 0.1798] |
| Gun Regulation | 0.0909 [0.0818, 0.1003] | 0.0729 [0.0649, 0.0815] | N/A | 0.1923 [0.1802, 0.2049] | 0.1817 [0.1697, 0.1944] |
| Gender Role | 0.0832 [0.0750, 0.0919] | 0.1274 [0.1174, 0.1380] | 0.4378 [0.4215, 0.4542] | 0.1130 [0.1039, 0.1228] | 0.1383 [0.1279, 0.1490] |

Table 25: Mean JS divergence and 95% bootstrap confidence intervals between ANES 2020 and silicon distributions by item and condition, using Llama-70B. Intervals come from a paired bootstrap that resamples both silicon and ANES responses ($n = 10{,}000$) and are reported for description only.

| Question | Replicate | Reformulated | Reverse-Coded | Priming | Preamble |
|---|---|---|---|---|---|
| Climate Change | 0.1391 [0.1292, 0.1499] | 0.2168 [0.2047, 0.2293] | 0.0526 [0.0460, 0.0600] | 0.1776 [0.1664, 0.1899] | 0.1540 [0.1438, 0.1649] |
| Current Economy | 0.1497 [0.1408, 0.1595] | 0.2194 [0.2075, 0.2316] | N/A | 0.2007 [0.1898, 0.2121] | 0.1680 [0.1583, 0.1776] |
| Refugee Allowing | 0.1835 [0.1755, 0.1918] | 0.0455 [0.0394, 0.0525] | 0.1990 [0.1900, 0.2083] | 0.1912 [0.1815, 0.2010] | 0.1819 [0.1740, 0.1901] |
| Health Insurance | 0.2010 [0.1923, 0.2100] | 0.1801 [0.1720, 0.1885] | N/A | 0.1979 [0.1891, 0.2069] | 0.1835 [0.1755, 0.1918] |
| Income Inequality | 0.1463 [0.1390, 0.1539] | 0.1445 [0.1368, 0.1524] | 0.1301 [0.1219, 0.1388] | 0.1371 [0.1298, 0.1447] | 0.1607 [0.1526, 0.1691] |
| Race Diversity | 0.2045 [0.1945, 0.2146] | 0.2198 [0.2086, 0.2313] | 0.2599 [0.2501, 0.2700] | 0.1781 [0.1675, 0.1888] | 0.2167 [0.2077, 0.2260] |
| Drug Addiction | 0.1719 [0.1641, 0.1801] | 0.1215 [0.1129, 0.1307] | N/A | 0.1727 [0.1649, 0.1808] | 0.1727 [0.1649, 0.1808] |
| Gay Marriage | 0.0623 [0.0551, 0.0703] | 0.1436 [0.1325, 0.1554] | 0.1051 [0.0987, 0.1117] | 0.0592 [0.0520, 0.0671] | 0.0594 [0.0525, 0.0671] |
| Gun Regulation | 0.1556 [0.1444, 0.1673] | 0.1268 [0.1167, 0.1374] | N/A | 0.1591 [0.1480, 0.1709] | 0.1963 [0.1844, 0.2085] |
| Gender Role | 0.0073 [0.0050, 0.0103] | 0.1186 [0.1086, 0.1291] | 0.7206 [0.7064, 0.7356] | 0.1326 [0.1232, 0.1429] | 0.0107 [0.0078, 0.0143] |

Table 26: Mean JS divergence and 95% bootstrap confidence intervals between ANES 2020 and silicon distributions by item and condition, using GPT-4.1-mini. Intervals come from a paired bootstrap that resamples both silicon and ANES responses ($n = 10{,}000$) and are reported for description only.

| Question | Replicate | Reformulated | Reverse-Coded | Priming | Preamble |
|---|---|---|---|---|---|
| Climate Change | 0.0536 [0.0476, 0.0605] | 0.0213 [0.0170, 0.0263] | 0.3202 [0.3074, 0.3340] | 0.1996 [0.1890, 0.2108] | 0.0760 [0.0686, 0.0844] |
| Current Economy | 0.1958 [0.1850, 0.2071] | 0.1463 [0.1355, 0.1580] | N/A | 0.2637 [0.2533, 0.2753] | 0.1846 [0.1744, 0.1954] |
| Refugee Allowing | 0.0816 [0.0733, 0.0906] | 0.0192 [0.0153, 0.0239] | 0.0357 [0.0301, 0.0420] | 0.0936 [0.0848, 0.1034] | 0.2147 [0.2034, 0.2269] |
| Health Insurance | 0.0342 [0.0288, 0.0403] | 0.0577 [0.0507, 0.0653] | N/A | 0.0656 [0.0579, 0.0738] | 0.1579 [0.1473, 0.1694] |
| Income Inequality | 0.0371 [0.0313, 0.0435] | 0.0845 [0.0766, 0.0931] | 0.0048 [0.0029, 0.0073] | 0.0715 [0.0635, 0.0801] | 0.0752 [0.0671, 0.0840] |
| Race Diversity | 0.2273 [0.2157, 0.2390] | 0.1548 [0.1445, 0.1657] | 0.0787 [0.0705, 0.0877] | 0.2663 [0.2563, 0.2760] | 0.2167 [0.2049, 0.2282] |
| Drug Addiction | 0.1629 [0.1548, 0.1714] | 0.1392 [0.1315, 0.1473] | N/A | 0.1727 [0.1649, 0.1808] | 0.1727 [0.1649, 0.1808] |
| Gay Marriage | 0.0363 [0.0309, 0.0423] | 0.0294 [0.0244, 0.0350] | 0.0953 [0.0896, 0.1013] | 0.0237 [0.0192, 0.0288] | 0.0470 [0.0407, 0.0539] |
| Gun Regulation | 0.0018 [0.0007, 0.0034] | 0.0158 [0.0122, 0.0201] | N/A | 0.0085 [0.0059, 0.0117] | 0.0117 [0.0086, 0.0154] |
| Gender Role | 0.0696 [0.0619, 0.0777] | 0.0109 [0.0079, 0.0146] | 0.8459 [0.8330, 0.8591] | 0.1814 [0.1719, 0.1913] | 0.1156 [0.1063, 0.1251] |

### D.11 Confidence intervals for JS-divergence in ANES 2024

Table 27 reports the mean JS-divergence and 95% bootstrap confidence intervals for ANES 2024 using GPT-4.1-mini.

Table 27: Mean JS divergence and 95% bootstrap confidence intervals between **ANES 2024** and silicon distributions by item and condition, using GPT-4.1-mini. Intervals come from a paired bootstrap that resamples both silicon and ANES responses ($n = 10,000$) and are reported for description only.

| Question | Replicate | Reformulated |
|---|---|---|
| Climate Change | 0.0811 [0.0741, 0.0887] | 0.0297 [0.0248, 0.0353] |
| Health Insurance | 0.0230 [0.0184, 0.0280] | 0.0284 [0.0232, 0.0341] |
| Income Inequality | 0.0226 [0.0182, 0.0277] | 0.0962 [0.0877, 0.1049] |
| Race Diversity | 0.2700 [0.2574, 0.2832] | 0.1678 [0.1571, 0.1795] |
| Drug Addiction | 0.1229 [0.1156, 0.1306] | 0.1126 [0.1054, 0.1201] |
| Gay Marriage | 0.0214 [0.0171, 0.0263] | 0.0052 [0.0032, 0.0078] |
| Gun Regulation | 0.0009 [0.0003, 0.0022] | 0.0233 [0.0189, 0.0283] |
| Gender Role | 0.0256 [0.0212, 0.0305] | 0.0185 [0.0146, 0.0230] |

### D.12 Confidence intervals for JS-divergence in WVS Wave 7

Tables 28, 29, and 30 report the mean JS-divergence and 95% bootstrap confidence intervals for the three WVS Wave 7 countries using GPT-4.1-mini. Exact p-values for quantifying significance are in Table 31, Table 32 and Table 33.

Table 28: Mean JS-divergence and 95% bootstrap confidence intervals between WVS and silicon distributions by item and condition, using GPT-4.1-mini (**WVS Wave 7**, **Netherlands**). Intervals come from a paired bootstrap ($n = 10,000$) and are reported for description only.

| Question | Replicate | Reformulated |
|---|---|---|
| Environment vs Economy | 0.0114 [0.0067, 0.0174] | 0.0031 [0.0011, 0.0062] |
| Refugee Asylum | 0.0847 [0.0744, 0.0953] | 0.0743 [0.0637, 0.0854] |
| Government Responsibility | 0.0892 [0.0777, 0.1047] | 0.0725 [0.0616, 0.0871] |
| Income Equality | 0.0976 [0.0863, 0.1129] | 0.0693 [0.0586, 0.0841] |
| Immigration Impact | 0.1646 [0.1475, 0.1833] | 0.2129 [0.1930, 0.2351] |
| Gay Parenting | 0.0735 [0.0621, 0.0869] | 0.0715 [0.0608, 0.0843] |
| Maternal Employment | 0.0891 [0.0795, 0.0995] | 0.0573 [0.0483, 0.0681] |
| Immigration Policy | 0.2862 [0.2638, 0.3097] | 0.2092 [0.1898, 0.2295] |
| Government Surveillance | 0.3321 [0.3102, 0.3556] | 0.2516 [0.2299, 0.2739] |
| Strong Leader | 0.3594 [0.3379, 0.3818] | 0.2030 [0.1836, 0.2235] |

Table 29: Mean JS-divergence and 95% bootstrap confidence intervals between WVS and silicon distributions by item and condition, using GPT-4.1-mini (**WVS Wave 7**, **Germany**). Intervals come from a paired bootstrap ($n = 10,000$) and are reported for description only.

| Question | Replicate | Reformulated |
|---|---|---|
| Environment vs Economy | 0.0891 [0.0736, 0.1062] | 0.0717 [0.0579, 0.0871] |
| Refugee Asylum | 0.0528 [0.0434, 0.0634] | 0.0517 [0.0429, 0.0615] |
| Government Responsibility | 0.1295 [0.1125, 0.1514] | 0.1105 [0.0945, 0.1321] |
| Income Equality | 0.1943 [0.1750, 0.2190] | 0.1631 [0.1442, 0.1872] |
| Immigration Impact | 0.1475 [0.1284, 0.1691] | 0.2202 [0.1959, 0.2475] |
| Gay Parenting | 0.1131 [0.0972, 0.1315] | 0.1167 [0.1003, 0.1356] |
| Maternal Employment | 0.1640 [0.1500, 0.1791] | 0.1633 [0.1461, 0.1828] |
| Immigration Policy | 0.1227 [0.1063, 0.1407] | 0.1018 [0.0869, 0.1191] |
| Government Surveillance | 0.3513 [0.3234, 0.3809] | 0.2146 [0.1901, 0.2418] |
| Strong Leader | 0.2516 [0.2315, 0.2742] | 0.1610 [0.1421, 0.1827] |

Table 30: Mean JS-divergence and 95% bootstrap confidence intervals between WVS and silicon distributions by item and condition, using GPT-4.1-mini (**WVS Wave 7**, **Great Britain**). Intervals come from a paired bootstrap ($n = 10{,}000$) and are reported for description only.

| Question | Replicate | Reformulated |
|---|---|---|
| Environment vs Economy | 0.0432 [0.0345, 0.0531] | 0.0229 [0.0166, 0.0304] |
| Refugee Asylum | 0.2307 [0.2175, 0.2442] | 0.2214 [0.2078, 0.2352] |
| Government Responsibility | 0.1109 [0.0981, 0.1264] | 0.0852 [0.0738, 0.0994] |
| Income Equality | 0.1807 [0.1649, 0.1992] | 0.1414 [0.1273, 0.1587] |
| Immigration Impact | 0.1096 [0.0962, 0.1246] | 0.0775 [0.0662, 0.0907] |
| Gay Parenting | 0.0954 [0.0829, 0.1096] | 0.1014 [0.0891, 0.1155] |
| Maternal Employment | 0.1203 [0.1111, 0.1302] | 0.1091 [0.0977, 0.1214] |
| Immigration Policy | 0.0895 [0.0801, 0.0997] | 0.0791 [0.0707, 0.0880] |
| Government Surveillance | 0.3468 [0.3255, 0.3694] | 0.2446 [0.2251, 0.2651] |
| Strong Leader | 0.2664 [0.2496, 0.2848] | 0.1255 [0.1117, 0.1412] |

Table 31: Full paired-comparison statistics against the Replicate baseline, using GPT-4.1 **(WVS Wave 7, Netherlands)**. $\Delta$JSD and its 95% bootstrap CI are from the per-replicate difference distribution. $p$ is the two-sided bootstrap $p$-value and $p^{\mathrm{BH}}$ its Benjamini–Hochberg adjustment.

| Question | Comparison | $\Delta$JSD | 95% CI | $p$ | $p^{\mathrm{BH}}$ | Sig. |
|---|---|---|---|---|---|---|
| Environment vs Economy | Reformulated | $-0.0084$ | $[-0.0135, -0.0039]$ | 0.0002 | 0.0003 | ✓ |
| Refugee Asylum | Reformulated | $-0.0104$ | $[-0.0191, -0.0019]$ | 0.0278 | 0.0309 | ✓ |
| Government Responsibility | Reformulated | $-0.0167$ | $[-0.0305, -0.0031]$ | 0.0168 | 0.0210 | ✓ |
| Income Equality | Reformulated | $-0.0283$ | $[-0.0421, -0.0142]$ | 0.0002 | 0.0003 | ✓ |
| Immigration Impact | Reformulated | $0.0482$ | $[0.0288, 0.0683]$ | 0.0002 | 0.0003 | ✓ |
| Gay Parenting | Reformulated | $-0.0020$ | $[-0.0145, 0.0104]$ | 0.7493 | 0.7493 | – |
| Maternal Employment | Reformulated | $-0.0318$ | $[-0.0408, -0.0220]$ | 0.0002 | 0.0003 | ✓ |
| Immigration Policy | Reformulated | $-0.0770$ | $[-0.0974, -0.0561]$ | 0.0002 | 0.0003 | ✓ |
| Government Surveillance | Reformulated | $-0.0805$ | $[-0.1013, -0.0602]$ | 0.0002 | 0.0003 | ✓ |
| Strong Leader | Reformulated | $-0.1563$ | $[-0.1751, -0.1373]$ | 0.0002 | 0.0003 | ✓ |

Table 32: Full paired-comparison statistics against the Replicate baseline, using GPT-4.1 **(WVS Wave 7, Germany)**. $\Delta$JSD and its 95% bootstrap CI are from the per-replicate difference distribution. $p$ is the two-sided bootstrap $p$-value and $p^{\mathrm{BH}}$ its Benjamini–Hochberg adjustment.

| Question | Comparison | $\Delta$JSD | 95% CI | $p$ | $p^{\mathrm{BH}}$ | Sig. |
|---|---|---|---|---|---|---|
| Environment vs Economy | Reformulated | $-0.0174$ | $[-0.0325, -0.0026]$ | 0.0210 | 0.0350 | ✓ |
| Refugee Asylum | Reformulated | $-0.0010$ | $[-0.0079, 0.0056]$ | 0.7213 | 0.8015 | – |
| Government Responsibility | Reformulated | $-0.0189$ | $[-0.0390, 0.0011]$ | 0.0620 | 0.0886 | – |
| Income Equality | Reformulated | $-0.0312$ | $[-0.0524, -0.0100]$ | 0.0058 | 0.0132 | ✓ |
| Immigration Impact | Reformulated | $0.0727$ | $[0.0492, 0.0971]$ | 0.0002 | 0.0007 | ✓ |
| Gay Parenting | Reformulated | $0.0036$ | $[-0.0126, 0.0195]$ | 0.6723 | 0.8015 | – |
| Maternal Employment | Reformulated | $-0.0007$ | $[-0.0126, 0.0127]$ | 0.9813 | 0.9813 | – |
| Immigration Policy | Reformulated | $-0.0210$ | $[-0.0356, -0.0062]$ | 0.0066 | 0.0132 | ✓ |
| Government Surveillance | Reformulated | $-0.1368$ | $[-0.1634, -0.1113]$ | 0.0002 | 0.0007 | ✓ |
| Strong Leader | Reformulated | $-0.0907$ | $[-0.1084, -0.0730]$ | 0.0002 | 0.0007 | ✓ |

Table 33: Full paired-comparison statistics against the Replicate baseline, using GPT-4.1-mini **(WVS Wave 7, Great Britain)**. $\Delta$JSD and its 95% bootstrap CI are from the per-replicate difference distribution. $p$ is the two-sided bootstrap $p$-value and $p^{\text{BH}}$ its Benjamini–Hochberg adjustment.

| Question | Comparison | $\Delta$JSD | 95% CI | $p$ | $p^{\text{BH}}$ | Sig. |
|---|---|---|---|---|---|---|
| Environment vs Economy | Reformulated | $-0.0203$ | $[-0.0290, -0.0121]$ | 0.0002 | 0.0003 | ✓ |
| Refugee Asylum | Reformulated | $-0.0094$ | $[-0.0144, -0.0041]$ | 0.0014 | 0.0017 | ✓ |
| Government Responsibility | Reformulated | $-0.0257$ | $[-0.0396, -0.0125]$ | 0.0002 | 0.0003 | ✓ |
| Income Equality | Reformulated | $-0.0393$ | $[-0.0556, -0.0231]$ | 0.0002 | 0.0003 | ✓ |
| Immigration Impact | Reformulated | $-0.0321$ | $[-0.0454, -0.0188]$ | 0.0002 | 0.0003 | ✓ |
| Gay Parenting | Reformulated | $0.0061$ | $[-0.0072, 0.0192]$ | 0.3638 | 0.3638 | – |
| Maternal Employment | Reformulated | $-0.0113$ | $[-0.0197, -0.0020]$ | 0.0162 | 0.0180 | ✓ |
| Immigration Policy | Reformulated | $-0.0104$ | $[-0.0168, -0.0044]$ | 0.0010 | 0.0014 | ✓ |
| Government Surveillance | Reformulated | $-0.1022$ | $[-0.1227, -0.0815]$ | 0.0002 | 0.0003 | ✓ |
| Strong Leader | Reformulated | $-0.1409$ | $[-0.1553, -0.1263]$ | 0.0002 | 0.0003 | ✓ |

# E   Datasets and Prompts

## E.1   Selected demographic variables

Table 34: Selected demographic variables of ANES 2020 respondents. 'Code' indicates the variable names used in the ANES 2020 dataset.

| Variable | Code | Answer choices |
|---|---|---|
| Race | V201549x | 1: white, 2: black, 3: hispanic, 4: asian, 5: native american |
| Gender | V201600 | 1: man, 2: woman |
| Age | V201507x | <Open-ended> |
| Ideology | V201200 | 1: extremely liberal, 2: liberal, 3: slightly liberal, 4: moderate, 5: slightly conservative, 6: conservative, 7: extremely conservative |
| Party | V201231x | 1: a strong democrat, 2: a weak Democrat, 3: an independent who leans Democratic, 4: an independent, 5: an independent who leans Republican, 6: a weak Republican, 7: a strong Republican |
| Political interest | V202406 | 1: very, 2: somewhat, 3: not very, 4: not at all |
| Church attendance | V201452 | 1: attend church, 2: do not attend church |
| Political discuss | V202022 | 1: I like to discuss politics with my family and friends., 2: I never discuss politics with my family or friends. |

## E.2   All questions and prompt conditions of ANES 2020

**Race diversity V202371**

*Original*: Does the increasing number of people of many different races and ethnic groups in the United States make this country a better place to live, a worse place to live, or does it make no difference?

1. Better  2. Worse  3. Makes no difference

*Reformulated:* How would this respondent describe the impact of the increasing number of people of many different races and ethnic groups on the United States as a place to live?

1. Improves it as a place to live  2. Worsens it as a place to live  3. Makes no difference

*Reverse-coded:* Does limiting the number of people of many different races and ethnic groups in the United States make this country a better place to live, a worse place to live, or does it make no difference?

1. Better  2. Worse  3. Makes no difference

**Gender role V202287**

*Original*: Do you think it is better, worse, or makes no difference for the family as a whole if the man works outside the home and the woman takes care of the home and family?

1. Better  2. Worse  3. Makes no difference

*Reformulated:* How would this respondent assess whether it is better, worse, or makes no difference for the family as a whole if the man works outside the home and the woman takes care of the home and family?

1. Better  2. Worse  3. Makes no difference

*Reverse-coded:* Do you think it is better, worse, or makes no difference for the family as a whole if both the man and the woman share work outside the home and take care of the home and family

1. Better  2. Worse  3. Makes no difference

**Current Economy V201324**

*Original*: What do you think about the state of the economy these days in the United States?

Table 35: Selected demographic variables of WVS Wave 7 respondents. 'Code' indicates the variable names used in the WVS Wave 7 dataset.

| Variable | Code | Answer choices |
|---|---|---|
| Race | Q290 | Country-specific codes; not applicable for Germany.
*Netherlands:* 528001: white, 528002: black, 528003: South Asian, 528004: East Asian, 528005: Arabic or Central Asian, 528999: another ethnic group.
*Great Britain:* 826100: White British, 826101: Irish, 826102: Gypsy or Irish Traveller, 826103: any other White background, 826002: Black Caribbean, 826003: Black African, 826004: any other Black background, 826005: Indian, 826006: Pakistani, 826007: Bangladeshi, 826008: Chinese, 826107: any other Asian background, 826015: Arabic or Central Asian, 826016: mixed race, 826104 & 826105: mixed White and Black, 826106: mixed White and Asian, 826999: another ethnic background. |
| Gender | Q260 | 1: man, 2: woman |
| Age | Q262 | <Open-ended> |
| Education | Q275 | 0: early childhood education or no formal education, 1: primary education, 2: lower secondary education, 3: upper secondary education, 4: post-secondary non-tertiary education, 5: short-cycle tertiary education, 6: a bachelor's degree or equivalent, 7: a master's degree or equivalent, 8: a doctoral degree or equivalent |
| Income | Q288R | 1: low, 2: medium, 3: high |
| Ideology | Q240 | 1: on the far left, 2: on the left, 3: left of center, 4: slightly left of center, 5: at the center, 6: at the center, 7: slightly right of center, 8: right of center, 9: on the right, 10: on the far right |
| Political interest | Q199 | 1: very, 2: somewhat, 3: not very, 4: not at all |
| Religious attendance | Q171 | 1: attend religious services more than once a week, 2: attend religious services once a week, 3: attend religious services once a month, 4: attend religious services only on special holy days, 5: attend religious services once a year, 6: attend religious services less than once a year, 7: never attend religious services |
| Political discuss | Q200 | 1: I frequently discuss political matters with my friends, 2: I occasionally discuss political matters with my friends, 3: I never discuss political matters with my friends |

1. Very good  2. Good  3. Neither good nor bad  4. Bad  5. Very bad

*Reformulated:* How would this respondent describe the state of the economy these days in the United States?

1. Very strong  2. Strong  3. Neither strong nor weak  4. Weak  5. Very weak

*Reverse-coded:* Not applicable

**Drug addiction V202348**

*Original*: Do you think the federal government should be doing more about the opioid drug addiction issue, should be doing less, or is it currently doing the right amount?

1. Should be doing more  2. Should be doing less  3. Is doing the right amount

*Reformulated:* How would this respondent assess whether the federal government should be doing more about the opioid drug addiction issue, should be doing less, or is it currently doing the right amount?

1. Should be doing more  2. Should be doing less  3. Is doing the right amount

*Reverse-coded:* Not applicable

**Climate change V202332**

*Original*: How much, if at all, do you think climate change is currently affecting severe weather events or temperature patterns in the United States?

1. Not at all  2. A little  3. A moderate amount  4. A lot  5. A great deal

*Reformulated:* How would this respondent assess how much, if at all, climate change is currently affecting severe weather events or temperature patterns in the United States?

1. Not at all  2. A little  3. A moderate amount  4. A lot  5. A great deal

*Reverse-coded:* How much do you agree with the statement that climate change is unrelated to severe weather events or temperature patterns in the United States?

1. Not at all  2. A little  3. A moderate amount  4. A lot  5. A great deal

**Gay marriage V201416**

*Original*: Which comes closest to your view? You can just tell me the number of your choice.

1. Gay and lesbian couples should be allowed to legally marry.  2. Gay and lesbian couples should be allowed to form civil unions but not legally marry.  3. There should be no legal recognition of gay or lesbian couples' relationship.

*Reformulated:* Which comes closest to this respondent's view?

1. Gay and lesbian couples should be allowed to legally marry.  2. Gay and lesbian couples should be allowed to form civil unions but not legally marry.  3. There should be no legal recognition of gay or lesbian couples' relationship.

*Reverse-coded:* Which one disagrees the most with your view?

1. Gay and lesbian couples should be allowed to legally marry.  2. Gay and lesbian couples should be allowed to form civil unions but not legally marry.  3. There should be no legal recognition of gay or lesbian couples' relationship.

**Refugee allowing V202234**

*Original*: Do you favor, oppose, or neither favor nor oppose allowing refugees who are fleeing war, persecution, or natural disasters in other countries to come to live in the U.S.?

1. Favor  2. Oppose  3. Neither favor nor oppose

*Reformulated:* What is this respondent's position on whether refugees who are fleeing war, persecution, or natural disasters in other countries should be allowed to come to live in the U.S.?

1. Should be allowed  2. Should not be allowed  3. No clear position

*Reverse-coded:* Do you favor, oppose, or neither favor nor oppose prohibiting refugees who are fleeing war, persecution, or natural disasters in other countries from coming to live in the U.S.?

1. Favor  2. Oppose  3. Neither favor nor oppose

**Health insurance V202378**

*Original*: Do you favor an increase, decrease, or no change in government spending to help people pay for health insurance when people cannot pay for it all themselves?

1. Increase  2. Decrease  3. No change

*Reformulated:* How would this respondent assess if there should be an increase, decrease, or no change in government spending to help people pay for health insurance when people cannot pay for it all themselves?

1. Increase  2. Decrease  3. No change

*Reverse-coded:* Not applicable

**Gun regulation V202337**

*Original*: Do you think the federal government should make it more difficult for people to buy a gun than it is now, make it easier for people to buy a gun, or keep these rules about the same as they are now?

1. More difficult  2. Easier  3. Keep these rules about the same

*Reformulated:* What is this respondent's position on whether the federal government should make it more difficult for people to buy a gun than it is now, make it easier for people to buy a gun, or keep these rules about the same as they are now?

1. More difficult  2. Easier  3. Keep these rules about the same

*Reverse-coded:* Not applicable

### Income inequality V202257

*Original*: Do you favor, oppose, or neither favor nor oppose the government trying to reduce the difference in incomes between the richest and poorest households?

1. Favor  2. Oppose  3. Neither favor nor oppose

*Reformulated:* How would this respondent assess whether the government should be trying to reduce the difference in incomes between the richest and poorest households?

1. Should be trying  2. Should not be trying  3. Neither of these

*Reverse-coded:* Do you favor, oppose, or neither favor nor oppose the government to stop trying to reduce the difference in incomes between the richest and poorest households?

1. Favor  2. Oppose  3. Neither favor nor oppose

### E.3 All questions and prompt conditions of WVS Wave 7

### Gay Parenting Q36

*Original*: How would you feel about the following statement? Do you agree or disagree with it? Homosexual couples are as good parents as other couples.

1. Agree strongly 2. Agree 3. Neither agree nor disagree 4. Disagree 5. Disagree strongly

*Reformulated:* How would this respondent feel about the following statement? Does this respondent agree or disagree with it? Homosexual couples are as good parents as other couples.

1. Agree fully 2. Agree 3. Neither agree nor disagree 4. Disagree 5. Disagree fully

### Refugee Asylum Q125

*Original*: From your point of view, what have been the effects of immigration on the development of this country? For the following statement about the effects of immigration, please, tell me whether you agree or disagree with it. Gives asylum to political refugees who are persecuted elsewhere.

2. Agree 1. Hard to say 0. Disagree

*Reformulated:* For the following statement about the effects of immigration, does this respondent agree or disagree with it? Gives asylum to political refugees who are persecuted elsewhere.

2. Agree 1. Hard to say 0. Disagree

### Income Equality Q106

*Original*: Now I'd like you to tell me your view on an issue. How would you place your view on this scale? 1 means you agree completely with the statement on the left; 10 means you agree completely with the statement on the right; and if your views fall somewhere in between, you can choose any number in between.

1. Incomes should be made more equal 10. There should be greater incentives for individual effort

*Reformulated:* How would this respondent place their view on this scale? 1 means complete agreement with the statement on the left; 10 means complete agreement with the statement on the right; and if their views fall somewhere in between, any number in between is possible.

1. Incomes should be made more equal 10. There should be greater incentives for individual effort

### Maternal Employment Q28

*Original*: For the following statement I read out, can you tell me how strongly you agree or disagree with

it. Do you strongly agree, agree, disagree, or strongly disagree? When a mother works for pay, the children suffer.

1. Strongly agree 2. Agree 3. Disagree 4. Strongly disagree

*Reformulated:* For the following statement, how strongly does this respondent agree or disagree with it? Does this respondent fully agree, agree, disagree, or fully disagree? When a mother works for pay, the children suffer.

1. Fully agree 2. Agree 3. Disagree 4. Fully disagree

**Environment vs Economy Q111**

*Original*: Here are two statements people sometimes make when discussing the environment and economic growth. Which of them comes closer to your own point of view?

1. Protecting the environment should be given priority, even if it causes slower economic growth and some loss of jobs. 2. Economic growth and creating jobs should be the top priority, even if the environment suffers to some extent. 3. Other answer

*Reformulated:* Here are two statements people sometimes make when discussing the environment and economic growth. Which of them comes closer to this respondent's point of view?

1. Protecting the environment should be given priority, even if it causes slower economic growth and some loss of jobs. 2. Economic growth and creating jobs should be the top priority, even if the environment suffers to some extent. 3. Other answer

**Immigration Impact Q121**

*Original*: Now we would like to know your opinion about the people from other countries who come to live in your country - the immigrants. How would you evaluate the impact of these people on the development of your country?

5. Very good 4. Quite good 3. Neither good nor bad 2. Quite bad 1. Very bad

*Reformulated:* Now we would like to know this respondent's opinion about the people from other countries who come to live in their country - the immigrants. How would this respondent evaluate the impact of these people on the development of their country?

5. Highly beneficial 4. Beneficial 3. Neither beneficial nor detrimental 2. Detrimental 1. Highly detrimental

**Government Responsibility Q108**

*Original*: Now I'd like you to tell me your view on an issue. How would you place your view on this scale? 1 means you agree completely with the statement on the left; 10 means you agree completely with the statement on the right; and if your views fall somewhere in between, you can choose any number in between.

1. Government should take more responsibility to ensure that everyone is provided for 10. People should take more responsibility to provide for themselves

*Reformulated:* How would this respondent place their view on this scale? 1 means complete agreement with the statement on the left; 10 means complete agreement with the statement on the right; and if their views fall somewhere in between, any number in between is possible.

1. Government should take more responsibility to ensure that everyone is provided for 10. People should take more responsibility to provide for themselves

**Immigration Policy Q130**

*Original*: How about people from other countries coming here to work. Which one of the following do you think the government should do?

1. Let anyone come who wants to 2. Let people come as long as there are jobs available 3. Place strict limits on the number of foreigners who can come here 4. Prohibit people coming here from other countries

*Reformulated:* How about people from other countries coming here to work. Which one of the following does this respondent think the government should do?

1. Let anyone come who wants to 2. Let people come as long as there are jobs available 3. Place strict

limits on the number of foreigners who can come here 4. Prohibit people coming here from other countries

**Government Surveillance Q198**

*Original*: Do you think that the your country's government should or should not have the right to keep people under video surveillance in public areas?

1. Definitely should have the right 2. Probably should have the right 3. Probably should not have the right 4. Definitely should not have the right

*Reformulated:* Does this respondent think that the your country's government should or should not have the right to keep people under video surveillance in public areas?

1. Definitely should have the right 2. Probably should have the right 3. Probably should not have the right 4. Definitely should not have the right

**Strong Leader Q235**

*Original*: I'm going to describe a type of political system and ask what you think about it as a way of governing this country. Would you say it is a very good, fairly good, fairly bad or very bad way of governing this country? Having a strong leader who does not have to bother with parliament and elections.

1. Very good 2. Fairly good 3. Fairly bad 4. Very bad

*Reformulated:* I'm going to describe a type of political system and ask what this respondent thinks about it as a way of governing this country. Would this respondent say it is a highly beneficial, beneficial, detrimental, or highly detrimental way of governing this country? Having a strong leader who does not have to bother with parliament and elections.

1. Highly beneficial 2. Beneficial 3. Detrimental 4. Highly detrimental

### E.4 Example of prompts

We provide two prompt examples of GPT-4.1-mini simulations for user responses to the multiple-choice health insurance question (ANES 2020), utilizing randomly sampled demographic variables. These examples illustrate a comparison under the Replicate and Reformulated question conditions.

---

**Replicate Condition**

**System**
Today is November 3, 2020. Racially, I am white. I like to discuss politics with my family and friends. Ideologically, I am moderate. Politically, I am a strong Republican. I attend church. I am 36 years old. I am a man.

**User**
Question: *Do you favor* an increase, decrease, or no change in government spending to help people pay for health insurance when people can't pay for it all themselves? Answer choices: 1. Increase 2. Decrease 3. No change When answering, respond ONLY with a single number that corresponds to the option you choose. Do not include any additional text, punctuation or explanation. My answer is

**Output**
1

---

**Reformulated Condition**

**System**
Today is November 3, 2020. Racially, *the respondent* is white. The respondent likes to discuss politics with their family and friends. Ideologically, the respondent is slightly liberal. Politically, the respondent is an independent who leans Democratic. The respondent does not attend church. The respondent is 37 years old. The respondent is a woman. The respondent is somewhat interested in politics.

**User**
Question: *How would this respondent assess* if there should be an increase, decrease, or no change in government spending to help people pay for health insurance when people cannot pay for it all themselves? Answer choices: 1. Increase 2. Decrease 3. No change When answering, respond ONLY with a single number that corresponds to the option you choose. Do not include any additional text, punctuation or explanation. My answer is

**Output**
1

