# OpenReview forum: "Mitigating Social Desirability Bias in Random Silicon Sampling"
_TMLR — Under review for TMLR_

### Review · Reviewer_TmBn · 2026-06-28

**Summary Of Contributions:**

### **Summary**
This paper asks if small, minimal, and guided prompt edits can pull LLM-simulated survey responses closer to real human distributions on socially sensitive items, building directly on the random silicon sampling. Across three models and five prompt conditions, the authors found that neutral, third-person reformulation is the most consistent intervention, while priming, preamble, and reverse-coding give weak or unstable results.

### **Strengths**
Quite good experimental setup, covering three models, five conditions, demographic stratification, two ANES waves, a separate survey instrument, three non-U.S. populations, and a decoding-temperature sweep.

### **Weaknesses/Concerns**
1. The main results use greedy decoding for deterministic output to ensure maximum reproducibility (Section 3.2, Appendix E.1), so each unique demographic profile maps to exactly one answer. It undermines the sampling and distribution framing of the work, as the aggregate response distribution is then a deterministic function of the demographic marginals, not a sample from the model's opinion. Thus, change in JSD under reformulation therefore reflects a changed deterministic mapping, not necessarily reduced social desirability bias.
2. In this way, lower JSD does not necessarily establish reduced bias. When divergence drops, it is read as mitigation; and when it rises (for example Llama-70B) it is attributed to controversial baselines or inaccurate model beliefs (Section 4.1); but it is not a proper measurement for the problem. Without an independent measure of social desirability, the causal claim rests on a directional metric that can move for many reasons.
3. Reformulation seems like a combined intervention with semantic shifts. For several items the answer options and scales change meaning, not just framing: "good/bad" becomes "strong/weak" (Current Economy), and "very good to very bad" becomes "highly beneficial to highly detrimental" (Immigration Impact). The human distribution is defined over the original options, so the comparison may be across different constructs. The authors acknowledge the relabeling can corelate and confuse, but there is no human validation that the rewrites preserve meaning.
4. Uncertainty estimation is a bit weak. Significance is inferred from non-overlapping 95 percent bootstrap CIs, which is not a proper test, and there is no multiple-comparison correction across the large number of items, conditions, models, and subgroups. The bootstrap resamples only silicon responses, treating the ANES distribution as fixed and ignoring its sampling error. Under deterministic decoding, several CIs collapse to zero width (for example Refugee Asylum at [0.0926, 0.0926] in Table 8 and [0.2323, 0.2323] in Table 10), which signals complete mode collapse and shows the bootstrap is not capturing meaningful uncertainty for those items.
5. Inconsistent decoding across models weakens the evaluation results and claims. Llama-70B uses classification-style decoding (highest next-token probability over option tokens), while Llama-8B and GPT-4.1-mini generate text.
6. Non-response handling discards a relevant signal. Dropping and renormalizing "Don't know," "Refused," and "Inapplicable" removes refusal behavior, which itself correlates with topic sensitivity, the core idea the paper is standing on. The model is never offered an abstention option that humans had.
7. The social desirability labeling is subjective and central. The desirable, undesirable, and safe option labels are assigned by the authors and drive the entire social desirability bias interpretation, yet there is no inter-annotator agreement or external validation.
8. Table 3 shows reformulation worsening several 2024 items (Income Inequality +0.087, Health Insurance, Gun Regulation), so the robust across waves claim (Contribution 5) is weak and not properly supported by data.

**Additional Comments:**

1. Figures can be improved. It was very hard for me to understand and interpret the results.
2. Option treatment seems inconsistent within the same item across conditions: Refugee Allowing relabels options under reformulation but keeps the original Favor/Oppose labels under reverse-coding.
3. The first systematic evaluation (Contribution 1) novelty claim is strong since Salecha et al. already used reverse-coding and Sun et al. supplied the baseline. State the incremental contribution more precisely.
4. Several reported gains are small in absolute terms (for example, GPT-4.1-mini average JSD 0.103 to 0.079).
5. The "non-U.S. populations" claim covers only three culturally proximate Western European countries; so the tone should be downplayed a bit.
6. The "sincerity instructions activate evaluation-perception" hypothesis is shown from one citation only. Should make it stronger.
7. Add sample sizes and exact p-values to tables in place of CI-overlap asterisks.

**Audience:**

Yes

**Audience Explanation:**

Yes, I think some human-AI interaction researchers, computational social science and  human-centric NLP researchers will be interested in this.

**Broader Impact Concerns:**

No such concerns

**Claims And Evidence:**

Yes

**Claims Explanation:**

Yes, but ***weakly***.

The core elements are OK, but several contributions are overstating them. Some claims should be toned down or more evidence should be provided to support the claims properly. The weaknesses/concerns section has more details of such gaps.

**Requested Changes:**

1. Following W2, add at least one independent measure of social desirability rather than inferring mitigation from JSD alone. One option can be adding a desirability score per item, an entropy or mode-collapse statistic reported alongside JSD, or a held-out validation that distinguishes *moved toward humans* from *moved toward a plausible target*. Without this, I think, the central causal claim is now not falsifiable.
2. Following W3, the authors can run an ablation that changes framing only while holding option labels fixed for every item where reformulation changes answer-option text or scale (for example, Current Economy good/bad to strong/weak; Refugee Allowing Favor/Oppose to Should/Should not; Immigration Impact very good to highly beneficial). Or, tone down the claims and contributions accordingly.
3. For W4, replace CI-overlap inference with or include a proper paired test on the bootstrap difference distribution (test whether the per-replicate delta excludes zero). Use multiple-comparison correction across items, conditions, models, and subgroups. Can also incorporate ANES sampling error into the bootstrap rather than treating the human distribution as fixed. Then, adjust writings and claims accordingly.
4. For W1, please justify or remove the deterministic-decoding framing. It can be done by either re-running the main experiments with stochastic decoding so the aggregate is an actual sample or reframing the contribution as a deterministic demographic-to-answer mapping. May also revise the language about distributions and sampling throughout.
5. About W5, resolve the inconsistent decoding across models. Llama-70B uses argmax-over-option-tokens, while the other two generate text. Either standardize the procedure or run a control showing the Llama-70B polarization is a model property, not a decoding artifact, since a major interpretation claim depends on it.
6. Provide more details on  non-response handling, and how it is not affecting the main results or explains the correlations.
7. Connecting to W7, please provide inter-annotator agreement or an external grounding for these labels, as desirable/undesirable/safe assignment motivates/connects the entire social desirability bias interpretation.
8. Connecting to W8, please tone down or provide evidence to support the claim strongly.
9. Add a short analysis or a correlated-sampling comparison quantifying how much the independence assumption (Equation 1) affects the JSD baseline.

---

> ### Author Response · Authors · 2026-07-07
> **Response to Reviewer TmBn**
>
> We appreciate Reviewer TmBn for a careful reading and providing constructive feedback. We have provided a revised version of the manuscript and responded to all comments below. We look forward to your response and further discussion. (W=Weakness and R=Requested Changes)
>
> **W1/R4 (deterministic decoding)**. All main experiments were re-run under stochastic decoding and model parameters are in Appendix B.1 (Llama models T=0.6, top-p 0.9, top-k = 50; GPT-4.1-mini T=1.0). A greedy-vs-stochastic comparison (Appendix D.6) shows condition effects are nearly identical under both, so our earlier conclusions still hold, but stochastic decoding is now the primary setting throughout.
>
> **W2/R1 (JSD alone is not a measure of SDB)**. We appreciate the suggestion and we introduce a signed desirability gap D that scores each distribution by its mass toward the socially desirable pole. We claim SDB reduction only under a joint criterion (ΔJSD < 0 and Δ|D| < 0) and apply it throughout the results section (Sec. 4). The new metric confirms the reformulation finding, and exposes cases where divergence falls for reasons unrelated to desirability.
>
> **W3/R2 (reformulation changes constructs, not just framing)**. We agree and following the suggested alternative, we have toned down the claims. In the new version, we state explicitly that reformulation is a combined intervention in Introduction and Methodology. This was also stated in our Limitations.
>
> **W4/R3 (weak uncertainty estimation)**. Replaced as requested. Significance is now a two-sided bootstrap test on paired per-replicate differences. The bootstrap resamples the ANES side as well, sharing the human replicate across conditions to pair the deltas. We control FDR at q=0.05 with Benjamini–Hochberg across all item-condition comparisons per model and dataset. CIs (in Appendix) are used for description only.
>
> **W5/R5 (inconsistent decoding across models: Llama70B**). We standardized the generation procedure and all three models now generate text under stochastic decoding. The new results are all updated and Llama70B still follows the same polarized pattern as the old ones, indicating a model property rather than a decoding artifact.
>
> **W6/R6 (non-response)**. We provide a detailed section in Appendix D.8 (Non-response rates and their effects). At the data level, we found that differential non-response across conditioning variables is weak (max Cramér's V = 0.078, below the small-effect threshold 0.1). We also recompute divergences with non-response reintroduced as an explicit category. It preserves the ranking of group-item pairs (Spearman $\rho$ = 0.90). We also consider the model with an explicit abstention option as future work.
>
> **W7/R7 (desirability labels)**. Labels were assigned independently by authors and two external annotators; agreement was unanimous on every directional item, and items without a clear approval direction were reclassified as non-directional and excluded from the signed desirability gap D calculation. These labels do not reflect a universal moral standard. We interpret them as indicators of population-level social pressure rather than as claims about the values of every subgroup.
>
> **W8/R8 (cross-wave claim)**. We toned down as requested. Sec 4.3 now states that reformulation helps and fails on the same items in 2020 and 2024. We consider this as a consistency check rather than testing robustness under a large distribution shift.
>
> **R9 (correlated-sampling comparison)**. We explained our silicon sampling setting in footnote 2. In Sec. 3.1, we also explicitly stated that “we evaluate how closely the aggregated silicon distribution matched the *marginal* human distribution.” We did not add a correlated sampling study in this cycle and listed this as future work.
>
> Additional comments.
>
> (1) We updated figures considering both JSD and D metrics for better visualization.
>
> (2) Under reverse-coding, options intentionally retain the original labels. The relabelling under reformulation is part of the combined intervention acknowledged in Sec. 3.
>
> (3) We refined our contributions, preventing overclaims.
>
> (4) We agree that several improvements, while significant under the paired test, are modest in absolute magnitude.
>
> (5) We explicitly state "Western countries/populations” in addition to “outside the U.S.", also in limitations.
>
> (6) We weaken the hypothesis for the evaluation-perception explanation.
>
> (7) Human sample size (5441, ANES 2020) were reported in the draft. Bootstrap replicate count is 10000. Due to the large number of tables, we now report only exact p-values for WVS survey comparisons in Appendix D.12.

---

### Review · Reviewer_AfWb · 2026-07-17

**Summary Of Contributions:**

The paper looks at social desirability bias in silicon sampling, i.e. conditioning an LLM on demographic profiles and aggregating the answers to approximate a population's opinion distribution. Building on Sun et al. (2024), the authors ask whether small changes to prompt wording can bring the simulated distributions closer to real human ones on sensitive items.

The contributions are:
- a controlled benchmark re-running the Sun et al. pipeline on ANES 2020 across three instruction-tuned models (Llama-3.1-8B, Llama-3.1-70B, GPT-4.1-mini)
- a signed desirability gap D that captures the direction of the deviation, combined with JS-divergence into a "joint criterion" that only counts an intervention as mitigation if both improve
- a comparison of four prompt interventions (reformulation, reverse-coding, priming, preamble), with reformulation coming out best
- generalization checks across demographic strata, ANES 2020 vs 2024, decoding strategies, and WVS Wave 7

Strengths:
- the signed vs absolute gap distinction is effective, it is what shows that Llama-70B's small mean signed gap (D = 0.117) is cancellation between per-item deviations rather than agreement, which JS-divergence alone would not have caught
- the bootstrap resamples the human side as well instead of treating ANES as fixed, and multiple comparisons are FDR-controlled with an explicitly defined family
- the 2020 to 2024 comparison is informative in both directions, reformulation helps and fails on the same items in both waves
- the limitations and ethics sections are candid

Weaknesses:
- the desirability labels, which are the definitional basis of the new metric, are under-validated and confounded with political direction
- the motivation for the reformulation mitigation is not tested tested and equally plausible alternatives are not considered or discussed

**Audience:**

Yes

**Audience Explanation:**

Silicon sampling is getting adopted quickly in political science, marketing research and social simulation, often by people who take aggregate alignment with a human benchmark as evidence that the simulation is faithful. This paper is relevant to that audience in both directions.

The positive result is cheap to use, neutral third-person reformulation improves alignment on sensitive items, needs no model access or retraining, and holds across waves and decoding strategies.

I think the negative results are at least as useful, because they go against what a reasonable practitioner would guess. Adding a preamble promising anonymity and no judgment is standard in human survey work on sensitive topics, and the finding that it does not transfer, and makes things worse on two of three models, saves other groups from repeating an intuitive mistake. Same for the finding that telling a model to reason analytically increases response homogeneity, which matters because faithful simulation has to reproduce the variance of human attitudes and not just the central tendency.

The signed gap and the joint criterion are the part most likely to be reused elsewhere. The point that distributional distance alone cannot separate bias reduction from alignment improving for some unrelated reason applies to any targeted intervention evaluated against an aggregate benchmark. Llama-70B is a clean example of the failure mode and Income Inequality is a clean example of the criterion catching something the authors could otherwise have reported as a success.

**Broader Impact Concerns:**

The ethics section is more thoughtful than usual and I am not requesting anything beyond what is there.

I want to point out the "Risks of mitigation" paragraph as a strength. The authors say that a closer match to a population benchmark may reflect either grounded subgroup knowledge or just outputs shifted toward a plausible target, and that where the model lacks that knowledge, removing surface bias becomes actively misleading. That is the main risk of their own contribution, they identified it themselves, and they draw the right conclusion from it. I would encourage keeping it prominent, since it is the point most likely to get lost when the method gets picked up downstream, and the ethics section is where readers are least likely to look for it.

One thing connects to my main review. If the labels are really tracking political direction and not social approval, then mitigating SDB in practice means shifting simulated populations rightward relative to the model's default. A reader who accepts the SDB framing would read that shift as increased fidelity when it may be an ideological adjustment. Given that silicon sampling is intended for polling and policy-adjacent research, this is another reason for the construct validation above to appear in the main text.

**Claims And Evidence:**

Yes

**Claims Explanation:**

Overall the claims made in the paper are supported, however several aspects could be improved.

1) Social desirability bias is present and persistent at baseline
- Table 1 shows positive mean D for all three models, indicating that the tested models overly agree with what the authors labelled as social desirability options, and Figure 2b shows this is not driven by a few items
- I find the discussion of the labelling process and how to precisely define "social desirability" insufficient. The paper relies on agreement between the authors and two annotators external to the project, but no agreement statistic is reported, the annotators are not described, and the instructions are not given.
- More importantly, "social desirability" here seems equivalent to politically progressive responses. These are different explanations, one about normative approval under a perceived evaluative frame and one about an ideological prior from training and preference tuning, and they predict the same D
- Since D is what distinguishes this paper from prior work using divergence only, this ambiguity carries into everything downstream
- It would be helpful if the authors discuss this and for example cross reference their social desirability labels with independent political alignment labels like progressive, conservative etc to better contextualise their labelling procedure

2) The bias materialises differently across models
- The authors report an interesting and well supported finding that Llama-8B and GPT-4.1-mini concentrate on the desirable option, while Llama-70B is more polarized
- The evidence is direct, the gap between Llama-70B's signed (0.117) and absolute (0.178) values in Table 1, the per-item sign changes in Figure 2b, and the distributions in Figure 6 all agree
- This is also the clearest justification for introducing the signed/absolute distinction, the authors had a flattering headline number available and correctly diagnosed it as an artifact instead

3) Reformulation is an effective mitigation strategy compared to the others
- Overall the numbers support that reformulation effectively reduces SDB. Table 2 shows the largest improvements of any condition on two of three models, Figure 3a shows the two reductions co-occurring per item, Table 12 shows it is stable across decoding
- I find the explanation of why this specific mitigation should help questionable. The authors hypothesize that reformulation reduces the LLM's perception of being evaluated
- Evaluation awareness is to the best of my knowledge actively being researched in other fields of Machine Learning, and I do not see why the proposed reformulations should reduce that perception. A similarly possible hypothesis would be that the reformulation changes the perception of the LLM user as more neutral, reducing sycophantic tendencies of wanting to agree with the user. "What do you think" puts the model in the position of disclosing an opinion to someone with an inferable stance, "what would this respondent think" puts it in the position of predicting a third party's answer, and those two accounts fit the results equally well
- The authors already cite Ranaldi & Pucci (2023) on sycophancy in the limitations but do not connect it to their main result
- The confident mechanistic language in Section 4.1 also sits somewhat awkwardly next to the limitations section, which concedes that reformulation is a combined intervention over framing and option labels whose contributions cannot be separated
- A more nuanced discussion here would be helpful

4) Reformulation generalizes
- The ANES 2020 to 2024 shift shows temporal stability of the mitigation, and I find the way it does so more convincing than a bare positive replication, reformulation helps on the same four items in both waves and fails on the same items in both waves. Consistent failure replicating alongside consistent success is harder to get by chance
- The authors correctly note the demographic composition of the two waves differs only modestly, so this tests stability over time rather than robustness to a different population
- The authors only test surveys from countries culturally close to the US, therefore the generalization beyond that is unclear (the authors make this limitation clear however), as are the German confound and the fact that it holds only for a subset of items and one model


Overall, I think the limited discussion and clarity of the authors' definition of "social desirability" is the biggest issue, and additionally I think the authors should broaden their discussion of the underlying effect of reformulation (not just reduced evaluation awareness). Beyond that the authors sufficiently discuss existing limitations yet support the more interesting claims sufficiently.

**Requested Changes:**

All of these would strengthen the work, none of them is critical to my recommendation.

- Report the labelling procedure properly, including the number and background of annotators, the instructions given, whether labelling was blind to the hypotheses, how the non-directional classification was adjudicated, and an actual agreement statistic rather than a statement that agreement was reached
- Address the political confound in the main text. Cross reference the desirability labels against independently obtained political alignment labels and report where the two agree and where they come apart. The items where they come apart are the best available evidence for the SDB reading and should be highlighted. The WVS items may be more useful here than ANES, since Government Surveillance and Strong Leader do not map onto the US progressive/conservative axis in the same way. If the two labellings agree nearly everywhere, I think it is better to say so plainly and scope D accordingly than to leave it implicit
- Broaden the mechanism discussion. List the candidates explicitly (reduced evaluation awareness, reduced sycophancy from repositioning the model relative to the user, a shift from persona role-play to population-level prediction), say which ones the design can and cannot distinguish, and either soften Sections 3.3 and 4.1 or add evidence that separates them. Also reconcile Section 4.1 with the limitations section on the combined intervention
- Use the evidence you already have on this. Government Surveillance and Strong Leader in WVS keep identical answer options across conditions, so only the framing changes, and both improve significantly in all three countries. That is the cleanest evidence in the paper that framing does something on its own, and it currently goes unremarked
- The baseline uses different models and a different metric than Sun et al., so "re-implementation of the pipeline" would describe it better than "replication"

---

### Review · Reviewer_cQBA · 2026-07-19

**Summary Of Contributions:**

Running surveys with human participants can be expensive and/or impractical. “Silicon sampling” proposes to use LLMs to instead simulate human responses. The authors use the ANES dataset, which contains questions on sensitive topics such as gender roles, gun regulation, etc. Along with responses to the questions, the dataset includes the demographic attributes of each respondent. For silicon sampling, the LLM responses are conditioned on a particular set of demographic attributes included in the prompt. For example, “Racially, I am white. Ideologically, I am slightly conservative. Politically,...”. The question is how well the distribution of LLM responses approximates the distribution of human responses (when the LLM prompt demographics are derived from the respondents’ demographics).

The challenge the authors tackle is that LLMs are known to exhibit social desirability bias (SDB), where they tend to express views that are more “socially acceptable” than the human-expressed average. This effect threatens the validity of silicon sampling. The authors test whether this effect can be mitigated by four prompt engineering techniques. The most effective technique is “reformulation”, where the question is reworded in a more neutral third-person way. The authors test three LLMs (Llama 3.1 8B and 70B, GPT 4.1 mini), and reformulation significantly reduces SDB and improves distribution alignment between human and silicon for Llama 3.1 8B and GPT 4.1 mini, with a neutral-to-slightly-adverse effect for Llama 3.1 70B. The authors also provide more granular analysis by stratifying by question category, demographics, etc.

The authors also test temporal shift by comparing ANES 2020 and ANES 2024, and also test another dataset with European respondents. These experiments produce similar results, although they only include GPT 4.1 mini.

**Strengths**

1. The paper is clearly written and easy to read.
2. The experimental design generally seems sound, and the attention to statistical validity is particularly notable.
3. The claims in the paper are generally appropriate given the evidence.
4. The discussion of limitations is honest and helpful.

**Weaknesses**

1. I feel that the problem could be motivated better.
2. None of the prompt engineering methods are effective for all 3 LLMs. I think the authors *mostly* avoid overstating the effectiveness of these methods, though.
3. The 2020-vs-2024 and European experiments only use GPT 4.1 mini.
4. I think the stratified analysis has some issues (see below).

**Audience:**

Yes

**Audience Explanation:**

I think some individuals in TMLR’s audience would be interested in this paper. That said, I think better motivating the problem could increase the number of such individuals. For example, it seems to me like there are high-stakes questions like election polling where it’s probably important to survey humans directly, even if silicon sampling is relatively accurate. Then there are lower-stakes questions such as marketing where silicon sampling seems appropriate, but improving the efficiency of marketing research doesn’t seem particularly valuable to me. If the authors have particular applications in mind for this type of work, that could be helpful to include.

To be clear, I think improving the motivation is not necessary for acceptance, based on TMLR’s criteria. Also, my comments above are based on my personal values, which other readers may not share. This is just a suggestion that the authors are welcome to take or not.

**Broader Impact Concerns:**

I found the discussion of ethical considerations to be mostly sufficient, but I think there is one concern that could be discussed more explicitly. Specifically, I think there is a risk of overly relying on LLMs to judge our preferences for us which ultimately results in handing more and more decision-making power to these models. Even if the human and silicon samples produce the same answer distribution on some topics, I think there are second-order effects here that would concern me if silicon sampling were adopted widely (e.g., instead of running normal election polls, everyone decided to just do silicon sampling). The “Synthetic data” paragraph somewhat but not fully addresses this, in my opinion.

**Claims And Evidence:**

Yes

**Claims Explanation:**

Generally, yes. As mentioned above, the experimental design appears sound and I think the results and statistical analysis are sufficient to support most of the claims made by the authors. However, I do have three concerns, corresponding to Weaknesses 2–4.

W2: For me as a reader, I felt that the abstract and the intro slightly oversold the effectiveness of these methods. I was a bit surprised to find that even the best method only worked for 2 out of the 3 tested LLMs. I recognize that this is subjective, though.

W3: Why is only GPT 4.1 mini used for these secondary experiments?

W4: Table 3 appears to lack statistical significance analysis. My understanding is that the multiple comparisons correction is performed for 40 comparisons (10 topics X 4 prompt conditions), but this doesn’t cover the subgroup analysis in Table 3. What makes this a bit more confusing is that most effect sizes (not just for the stratified analysis) are reported without confidence intervals in the main body, even in cases where the authors did perform statistical testing. Including at least some CIs in the main body and clarifying the role of Table 3 would help.

**Requested Changes:**

**Main changes**
1.  I think that clarifying that the reformulation benefit only applied to 2 out of 3 LLMs would make the abstract and intro give a more accurate impression of the findings. However, like I said, this is subjective, and I don't insist on this change.
2. My inclination is to request that the Llama models also be included in the 2020-vs-2024 and European experiments, but I would also like to understand the authors’ rationale for originally not doing so.
3. I think reporting at least some CIs in the main body would be helpful.

**Minor changes/questions**

4. Why is ANES 2020 used for the primary results instead of using ANES 2024 or using both?
5. Looking at E.2, it seems like the answers labeled as “socially desirable” are mostly the same as the more liberal answers. Do you think this is accurate, and if so, what are the implications of this?
6. The citation for the Llama models appears to be formatted incorrectly.
7. I’m a bit confused about the reasoning for choosing GPT 4.1 mini over GPT 5 mini. Don’t they have comparable speed and cost?
8. “Limitatisons” is misspelled on pg 5
9. Are the demographic attributes provided in the user prompt or system prompt? Did you test if this change makes any difference?
10. Should equation 4 be D(X,c)?
11. It might be worth stating more prominently what fraction of items were non-directional. I had to search a bit to find that information.
12. For the 2020-vs-2024 and European results, it would be nice to have an overall results table that aggregates across the topics (similar to Table 2).

---

### Review · Reviewer_5iXd · 2026-07-22

**Summary Of Contributions:**

This paper studies social desirability bias in random silicon sampling, where LLMs are conditioned on demographic profiles and used to generate population-level survey responses.

The authors compare an original-question baseline against four prompt interventions: Reformulated, Reverse-coded, Priming, and Preamble. Experiments use ten ANES 2020 items and three instruction-tuned models: Llama-3.1-8B-Instruct, Llama-3.1-70B-Instruct, and GPT-4.1-mini.

The paper evaluates agreement with human responses using Jensen–Shannon divergence and introduces a signed desirability gap to measure the direction of disagreement. The main result is that the Reformulated condition improves average alignment for Llama-8B and GPT-4.1-mini, but not for Llama-70B. The other interventions show limited or inconsistent benefits. The paper also reports exploratory subgroup, temporal, and cross-country analyses.

**Audience:**

Yes

**Audience Explanation:**

The reliability of LLM-generated survey populations is relevant to researchers working on language models, computational social science, synthetic agents, polling, and human simulation.
The findings remain valuable if the paper is framed as a study of prompt-induced distributional sensitivity rather than as a validated general method for mitigating social desirability bias.

**Broader Impact Concerns:**

The Ethical Considerations section appropriately warns that silicon samples are model outputs rather than empirical human data and that they may reproduce stereotypes or harm marginalized groups.
However, phrases such as “more representative silicon samples” and “a practical path” may overstate the validation achieved by the experiments.

**Claims And Evidence:**

No

**Claims Explanation:**

The paper clearly shows that prompt wording can substantially change LLM response distributions. It also shows that the particular combined reformulation tested here moves some model–item combinations closer to the original human survey distribution.
However, the stronger claim that neutral third-person reformulation causally mitigates social desirability bias is not established for the following reasons.

First, human respondents answer the original survey questions, whereas LLMs in the Reformulated condition answer modified questions and, in some cases, modified response options. A lower divergence may therefore reflect a change in the measurement instrument rather than reduced bias. A human split-ballot experiment is needed to establish that the original and reformulated questions produce comparable human response distributions.

Second, reformulation changes multiple factors simultaneously: first-person versus third-person framing, evaluative wording, policy-focused wording, option labels, and the task itself. The task changes from role-playing as a demographic persona to predicting how another respondent would answer. The current design cannot determine which component causes the observed effect.

Third, the interpretation of the joint criterion is not fully correct. A reduction in the absolute desirability gap means that the model’s desirability score moves closer to the human score. It does not always mean that the model selects socially desirable answers less often, especially when the baseline signed gap is negative.

Fourth, inferential statistics are reported mainly for item-level changes in Jensen–Shannon divergence. The main text does not provide corresponding uncertainty estimates or tests for the desirability gap, the joint criterion, Table 2 averages, or model-by-condition interactions. Therefore, observed differences between models should not be described as statistically established model-specific effects.

Fifth, marginal agreement does not demonstrate demographic fidelity. A model could ignore the demographic profile and still approximate the overall human response distribution. No-demographics, shuffled-demographics, or demographic-ablation controls are needed.

Finally, explanations involving reduced evaluation pressure, monitoring awareness, internal population knowledge, or model-specific polarization are plausible hypotheses, but they are not directly tested. The temporal and WVS results also support only limited transfer for one model and selected items, not broad generalization.

**Requested Changes:**

1. Narrow the central claim or add a human split-ballot experiment.

The authors should test whether human respondents give comparable distributions under the original and reformulated questions. Without this evidence, the paper should state only that a combined reformulation improves agreement with the original human benchmark for selected models and items.

2. Decompose the Reformulated condition.

The effects of third-person framing, neutral wording, option relabeling, policy-focused wording, and respondent prediction should be tested separately. Multiple independent reformulations should also be used to reduce dependence on one hand-written prompt.

3. Revise and validate the desirability-gap analysis.

The joint criterion should be described as movement toward the human desirability score, not necessarily reduced selection of desirable answers. Confidence intervals or statistical tests should be reported for D, |D|, and changes in |D|. Sensitivity analyses should distinguish neutral from socially undesirable options.

4. Test interactions directly.

The paper should test model-by-condition, wave-by-condition, and country-by-condition interactions. Table 2 averages should include uncertainty, and items should be treated as an inferential unit if the authors wish to generalize across questions.

5. Verify demographic conditioning.

The study should include no-demographics, shuffled-demographics, and demographic-ablation controls, as well as comparisons of conditional human and LLM distributions. Otherwise, claims about representativeness or demographic fidelity should be removed.

6. Reframe or redesign the subgroup analysis.

Synthetic profiles should preserve the joint or conditional distribution of demographic attributes. Otherwise, subgroup results should be labeled exploratory, and the claim that the bias is structural rather than subgroup-specific should be removed.

7. Moderate generalization claims.

The ANES 2024 and WVS analyses use only GPT-4.1-mini and a limited set of items. The paper should describe these results as limited transfer evidence rather than broad temporal or cross-cultural generalization.

8. Clarify the estimand and data treatment.

The main text should state whether survey weights and complex survey design are used, how missing responses are handled, whether identical profiles are reused across conditions, and how many independent runs or random seeds are included.